# LiNo: Advancing Recursive Residual Decomposition of Linear and Nonlinear Patterns for Robust Time Series Forecasting

## Abstract

Forecasting models are pivotal in a data-driven world with vast volumes of time series data that appear as a compound of vast **Li**near and **No**nlinear patterns. Recent deep time series forecasting models struggle to utilize seasonal and trend decomposition to separate the entangled components. Such a strategy only explicitly extracts simple linear patterns like trends, leaving the other linear modes and vast unexplored nonlinear patterns to the residual. Their flawed linear and nonlinear feature extraction models and shallow-level decomposition limit their adaptation to the diverse patterns present in real-world scenarios. Given this, we innovate Recursive Residual Decomposition by introducing explicit extraction of both linear and nonlinear patterns. This deeper-level decomposition framework, which is named **LiNo**, captures linear patterns using a Li block which can be a moving average kernel, and models nonlinear patterns using a No block which can be a Transformer encoder. The extraction of these two patterns is performed alternatively and recursively. To achieve the full potential of LiNo, we develop the current simple linear pattern extractor to a general learnable autoregressive model, and design a novel No block that can handle all essential nonlinear patterns. Remarkably, the proposed LiNo achieves state-of-the-art on thirteen real-world benchmarks under univariate and multivariate forecasting scenarios. Experiments show that current forecasting models can deliver more robust and precise results through this advanced Recursive Residual Decomposition. We hope this work could offer insight into designing more effective forecasting models. Code is available at this anonymous repository: `https://anonymous.4open.science/r/LiNo-8225/`.

## 1 Introduction

Time series forecasting (TSF) is a long-established task (Lim & Zohren, 2021; Wang et al., 2024b), with a wide range of applications (Zhou et al., 2022a; Liu et al., 2022a; Piao et al., 2024). Notably, numerous deep learning methods have been employed to address the TSF problem, utilizing architectures such as Recurrent Neural Networks (RNNs) (Elman, 1990; Lin et al., 2023), Temporal Convolutional Networks (TCNs) (Donghao & Xue, 2024; Wu et al., 2023), Multilayer Perceptron (MLP) (Liu et al., 2023; Challu et al., 2023) and Transformers (Liu et al., 2022b; Kitaev et al., 2020). Recent advancements in the time series forecasting community have suggested that seasonal (nonlinear) and trend (linear) decomposition can enhance forecasting performance (Zhang et al., 2022b; Wu et al., 2021). This is typically performed once, with the trend component being extracted using methods such as the simple moving average kernel (MOV) (Zhou et al., 2022b; Wang et al., 2023), the exponential smoothing function (ESF) (Woo et al., 2022), or a learnable 1D convolution kernel (LD) (Yu et al., 2024). The seasonal component is then obtained by subtracting the trend part from the original time series.

Real-world time series data often exhibit more complex structures, combining multiple levels of linear and nonlinear characteristics (Stock & Watson, 1998; Dama & Sinoquet, 2021; Agrawal & Adhikari, 2013). This suggests that they are the result of the additive combination of various linear and nonlinear components, as illustrated in Figure 1. Simple seasonal and trend decomposition faces three main challenges when deployed in real-world time series data. Firstly, these methods rely on simple techniques (MOV, LD, ESF). MOV and LD employed a fixed window size, which

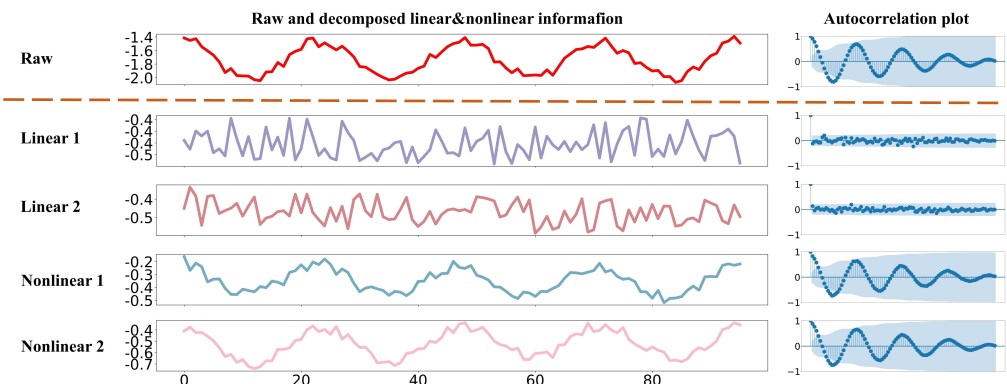

Figure 1: Example of the multi-level linear and nonlinear patterns in real-world time series. We take the ETTh2 dataset as an example and decompose a raw time series (**Raw**) into four signals through linear and nonlinear patterns decomposition. Linear 1&2 is the obtained linear patterns using the proposed Li block, and Nonlinear 1&2 is the obtained nonlinear patterns using No block. In other words, the raw time series (red) is the sum of the four signals below.

cannot fully extract all linear patterns, while the non-learnability of ESF limits its performance. They can only extract simple linear features such as trends, while other linear features like cyclic patterns and autoregression (Chatfield & Xing, 2019) remain underutilized. Secondly, we argue the obtained seasonal part is actually a residual consisting of all the unextracted linear and nonlinear information. Without further separating the nonlinear part from the residual severely hinders its extraction. Since it's extremely challenging for deep models to extract useful information from such mixtures (Bengio et al., 2013; Tishby & Zaslavsky, 2015). Another problem is the design of current nonlinear models, which mainly focus on one or two types of nonlinear patterns (temporal variations, frequency information, inter-series dependencies, etc). Such designs cannot satisfy our requirements in the real world. Thirdly, their shallow-level decomposition is incompatible with the multi-level characteristics of real-world time series. In contrast, numerous studies have shown that deeper-level decomposition can lead to better time series analysis (Huang et al., 1998; Rilling et al., 2003). Therefore, it is crucial to develop advanced decomposition methods capable of multi-granularity separation of various modes, and better linear and nonlinear pattern extractors to provide a more nuanced understanding of the signal's structure and potentially improve forecasting performance.

To address the challenges above, we propose adopting Recursive Residual Decomposition (RRD), a method used in Empirical Mode Decomposition (EMD) (Huang et al., 1998; Rilling et al., 2003), to decompose a time series into multiple patterns. This process is performed recursively. Each pattern is extracted based on the residual obtained by subtracting previously extracted patterns from the original signal, utilizing Intrinsic Mode Functions (IMF) to identify similar characteristics. Specifically, We can employ a Li block (MOV, LD, ESF, etc.) as the IMF for extracting linear patterns and a No block (Transformer, RNN, etc.) as the IMF for extracting nonlinear patterns. Then, the RRD is performed on a deeper level. We denote the proposed overall framework as **LiNo**. To fully realize RRD's potential, we propose an advanced **LiNo**, Specifically, we enhance the Li block to a general learnable autoregressive model with a full receptive field and propose a novel No block capable of modeling essential nonlinear features, such as temporal variations, period information, and inter-series dependencies in multivariate forecasting.

In summary, our contributions can be delineated as follows.

- We advance current shallow linear and nonlinear decomposition by innovating Recursive Residual Decomposition.

- Proposed No block demonstrates better nonlinear pattern extraction ability than current SOTA nonlinear models, such as iTransformer and TSMixer.

- LiNo consistently delivers top-tier performance in multivariate and univariate forecasting scenarios, demonstrating robust resilience against noise disturbances.

- The significant improvement by more nuanced and deeper linear and nonlinear decomposition provides insight for designing more effective and robust forecasting models.

## 2    RELATED WORK

**Advancement in recent time series forecasting.**    Time series forecasting is a critical area of research that finds applications in both industry and academia. With the powerful representation capability of neural networks, deep forecasting models have undergone a rapid development (Wang et al., 2024b; Lim & Zohren, 2021; Torres et al., 2021). Recent research endeavors have focused on segmenting the sequence into a series of patches (Nie et al., 2023; Zhang & Yan, 2023), better modeling the relationships between variables (Ng et al., 2022; Chen et al., 2024), the dynamic changes within a sequence (Wu et al., 2023; Du et al., 2023), or both (Yu et al., 2024; Liu et al., 2024a). Some works strive for more efficient forecasting solutions (Lin et al., 2024; Xu et al., 2024). Other efforts aim to revitalize existing architectures, such as RNN (Lin et al., 2023), Transformer (Liu et al., 2024b), TCN (Donghao & Xue, 2024), with new ideas, or to explore the potential of outstanding architectures from other domains, such as MLP-Mixer (Chen et al., 2023; Wang et al., 2024a), Mamba (Ahamed & Cheng, 2024; Wang et al., 2024c), Graph Neural Network (Yi et al., 2023; Shao et al., 2022), even Large Language Models (Jin et al., 2024; Liu et al., 2024c; Pan et al., 2024; Bian et al., 2024; Gruver et al., 2024), for application in time series forecasting. Notably, some efforts also begin to ponder the role of self-attention in time series forecasting (Ilbert et al., 2024).

**Importance of linear and nonlinear patterns.**    Deep time series models that are dedicated to model nonlinear patterns such as non-stationarity (Liu et al., 2022c), time-variant and time-invariant features (Liu et al., 2023), and frequency bias (Piao et al., 2024) have delivered outstanding performance in various domains. In contrast, recent advances have proved the importance of attention to linear patterns in time series (Toner & Darlow, 2024). For instance, DLinear (Zeng et al., 2023) and RLinear (Li et al., 2023a) achieve results comparable to, or even surpassing, many intricately designed nonlinear models in certain scenarios, using only simple linear layers. FITS (Xu et al., 2024) even achieves SOTA performance with merely $10k$ parameters. This suggests that balanced consideration of both linear and nonlinear patterns can be crucial for enhancing predictive performance in time series forecasting, which is unexplored in the current simple seasonal and trend decomposition.

**Decomposition based on Residual.**    **Autoformer** (Wu et al., 2021) first explored a rudimentary seasonal (**Nonlinear**) and trend (**Linear**) decomposition based on residual. The trend part is extracted using linear models like a moving average kernel, and the seasonal part is the residual of subtracting the input feature using the trend part. The earlier and more pioneering residual-based decomposition in deep learning time series was explored in **N-BEATS** (Oreshkin et al., 2020). N-BEATS is built using stacked residual blocks, where each block refines the forecast by predicting based on the residual errors of previous predictions. Instead of directly modeling the time series components (e.g., trend or seasonality), the model iteratively adjusts the residuals at each step.

Our LiNo develops the decomposition strategy in Autoformer by explicitly capturing both seasonal (**Nonlinear**) and trend (**Linear**) components and performing at a deeper level. Though LiNo and N-BEATS share some similarities, such as both employing the concept of Recursive Residual Decomposition (RRD), they are fundamentally different. First, N-BEATS does not explicitly capture linear and nonlinear separately. Second, as a univariate model, N-BEATS cannot handle more complex multivariate time series. Third, the role of RRD differs in each model: while N-BEATS uses RRD to refine the final prediction based on residual errors from previous predictions, LiNo employs RRD to capture more nuanced linear and nonlinear patterns. To empirically compare their differences, we designed a model based on the N-BEATS framework, where the internal components are identical to those of the proposed LiNo. In the subsequent experiments, we will denote this model as **Mu**. A more detailed comparison and analysis of LiNo and N-BEATS is left in Appendix F.

## 3    METHODOLOGY

### 3.1    PRELIMINARY

Given a multivariate time series $X \in \mathbb{R}^{C \times T}$ with a length of $T$ time steps, time series forecasting tasks are designed to predict its future $F$ time steps $\hat{Y} \in \mathbb{R}^{C \times F}$, where $C$ is the number of variate or channel ($C = 1$ in univariate case), and $T$ represents the look-back window. We aim to make $\hat{Y}$ close to the ground truth $Y \in \mathbb{R}^{C \times F}$.

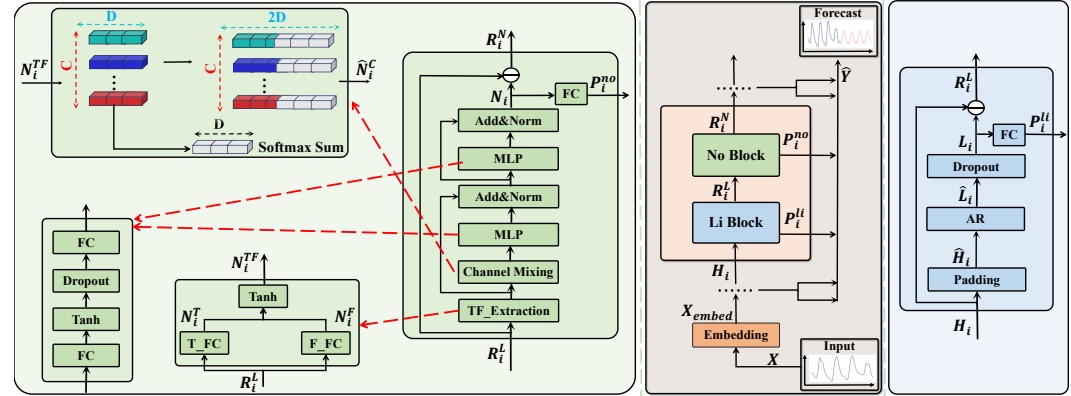

Figure 2: Framework of LiNo. Li block and No block extract patterns from the embedded input alternatively, in an RRD manner. The final prediction is aggregated from all blocks.

Without loss of generality and to simplify the analysis, we assume a real-world univariate time series $X$ consisting of $S$ linear patterns, $S$ nonlinear patterns, and a white noise $\varepsilon$. We denote $X = L_1 + N_1 + \cdots + L_S + N_S + \varepsilon$, where $L_i$ is a **L**inear signal, $N_i$ is a **No**nlinear signal. During the 'Recursive Residual Decomposition' (RRD) process of LiNo, if we denote the extracted linear and nonlinear patterns using different IMFs (such as MOV, LD, ESF for linear, Transformer, MLP-Mixer for nonlinear) at each time as $\hat{L}_i$ and $\hat{N}_i$, we have

$$R_1^L = X - \hat{L}_1, \; R_1^N = R_1^L - \hat{N}_1,$$
$$\cdots \tag{1}$$
$$R_S^L = R_{S-1}^N - \hat{L}_S, \; R_S^N = R_S^L - \hat{N}_S.$$

where $\lim_{i \to \infty} R_i^N = 0$. The modeling error will be $\delta = X - (\hat{L}_1 + \hat{N}_1 + \cdots + \hat{L}_n + \hat{N}_n) = (L_1 - \hat{L}_1) + (N_1 - \hat{N}_1) + (L_2 - \hat{L}_2) + (N_2 - \hat{N}_2) + \cdots + (L_S - \hat{L}_S) + (N_S - \hat{N}_S) + \varepsilon$. By alternating the extraction of linear and nonlinear features with subtracting the extracted features from the original input representation, we ensure previously extracted features do not affect the extraction of subsequent features, guaranteeing the independence of the resulting linear and nonlinear patterns. Moreover, $\lim_{i \to \infty} R_i^N = 0$ ensures that all valuable information in the sequence is fully extracted if the decomposition level is deep enough, preventing any loss of information. This design takes the existing shallow RRD to a deeper level, introducing the extraction of nonlinear patterns. Such a refined design ensures the model achieves more robust forecasting results. Notably, if set the RRD level to 1, then we get $Trend = IMF(X) = \hat{L}_1$, and $Seasonal = X - Trend = R_1^L$, where $IMF$ can be MOV, LD, or ESF, which is equivalent to the former seasonal and trend decomposition.

## 3.2 LiNo Pipeline

The structure of the LiNo framework is illustrated in Figure 2 (Middle). Initially, we extract the whole series embedding, which is then processed through $N$ LiNo blocks to forecast future values of the time series. In this subsection, we will explain the whole series embedding, Li block, and No block, step by step.

**Whole series embedding.**  Following iTransformer (Liu et al., 2024b), we first map time series data $X \in \mathbb{R}^{C \times T}$ from the original space to a new space to get $X_{embed} \in \mathbb{R}^{C \times D}$ using a simple linear projection, where $D$ denotes the dimension of the layer. Such a design can better preserve the unique patterns of each variate.

**Li block.**  The fixed window size of the MOV and learnable 1D convolution kernel (LD), and the non-learnability of the exponential smoothing function (ESF), prevent them from fully extracting all linear patterns. So we introduce a learnable autoregressive model (AR) with a full receptive field to replace them, where MOV, LD, and ESF are its subsets.

The structure of Li Block is shown in Figure 2 (right). Given its input feature $H_i \in \mathbb{R}^{C \times D}$ where $H_0 = X_{embed}$, we get $i$-th linear pattern $L_i \in \mathbb{R}^{C \times D}$ by:

$$
\begin{aligned}
\hat{L}_i[c,d] &= \phi_i[c,1] * H_i[c,1] + \phi_i[c,2] * H_i[c,2] \\
&\quad + \cdots + \phi_i[c,d] * H_i[c,d] + \beta_i[c], \\
L_i &= \mathbf{Dropout}(\hat{L}_i).
\end{aligned}
\tag{2}
$$

Here, $\phi_i \in \mathbb{R}^{C \times D}$ represents the autoregressive coefficients, $\beta_i \in \mathbb{R}^C$ denotes bias term. The whole process of extracting the linear part of the input feature can be easily deployed in a convolution fashion by setting the weight of the convolution kernel to $\phi_i$, and the weight of bias to $\beta_i$. Before applying the convolution (AR model), we pad the input feature $\mathbf{H}_i$ before convolution to make sure $H_i$ and $L_i$ have the same scale. The padded input feature $\hat{H}_i$ becomes,

$$
\hat{H}_i[:,t] = \begin{cases} H_i[:,t-D], & \text{for } t \geq D \\ 0, & \text{for } t < D \end{cases}.
\tag{3}
$$

We perform convolutions on each channel independently across the last dimension of $\hat{H}_i$. Following this, we apply dropout for generalization. The linear prediction $P_i^{li} \in \mathbb{R}^{C \times F}$ for this level is obtained by mapping the extracted linear component $L_i$. Subsequently, we pass $R_i^L = H_i - L_i$ to the next stage for nonlinear pattern modeling.

**No block.** Typical nonlinear time series characteristics include temporal variations, frequency information, inter-series dependencies, etc. While all of these factors are crucial for precise forecasting, current nonlinear models can only address one (Wu et al., 2023; Nie et al., 2023; Zhang et al., 2022a; Piao et al., 2024) or two (Liu et al., 2024b; Li et al., 2023b; Chen et al., 2023) of these aspects. Therefore, we designed a No block that simultaneously handles all these characteristics.

As in Figure 2 (left), given a feature $R_i^L \in \mathbb{R}^{C \times D}$, both temporal variation patterns and frequency information are accessible through linear projection. Hence, we start by applying a linear projection in the time domain to obtain temporal variation patterns $N_i^T \in \mathbb{R}^{C \times D}$ in a classical manner. Correspondingly, we apply a linear projection in the frequency domain, which is transformed from $R_i^L$ using the Fast Fourier Transform (FFT), following FITS (Xu et al., 2024). After that, the frequency domain representation is converted back, using the Inverse Fast Fourier Transform (IFFT) to obtain frequency information patterns $N_i^F \in \mathbb{R}^{C \times D}$. To leverage the complementary strengths of both domains, features extracted from both the time and frequency domains are fused and activated by: $N_i^{TF} = \mathbf{Tanh}(N_i^T + N_i^F)$.

To model inter-series dependencies, we first normalize $N_i^{TF}$ across the channel dimension using a softmax function, as in Figure 2 (upper left). We then compute a weighted mean by multiplying the softmax weights with $N_i^{TF}$ and summing over the channel dimension. This weighted mean is repeated to match and concatenate with $N_i^{TF}$. The concatenated result is passed through an MLP to obtain inter-series dependencies information $N_i^C$. We note that a recent work, **SOFTS** (Han et al., 2024), has adopted a similar approach to ours for modeling inter-series dependencies. A detailed comparative analysis is provided in Appendix G.

To integrate temporal variations, frequency information, and inter-series dependencies, we first apply Layer Normalization to the sum of $N_i^{TF}$ and $N_i^C$, resulting in $N_i^{TFC}$. A subsequent MLP is applied, and the result is added back to $N_i^{TFC}$, followed by a final Layer Normalization to produce the overall nonlinear pattern $N_i$. The nonlinear prediction $P_i^{no} \in \mathbb{R}^{C \times F}$ for this level is then obtained by mapping the extracted nonlinear part $N_i$. Finally, $R_i^N = R_i^L - N_i$ is passed to the next LiNo block, where $H_{i+1} = R_i^N$.

**RevIN and forecasting results.** We used RevIN (Kim et al., 2022) to counter the distribution problem following iTransformer(Liu et al., 2024b). The input first performs an Instance Normalization (Ulyanov, 2016) before being embedded. The final output is reversed using the Mean and Standard Deviation of Instance Normalization. The final prediction result is aggregated from multi-level by:

$$
\hat{Y} = \sum_{i=1}^{N} (P_i^{li}) + \sum_{i=1}^{N} (P_i^{no}).
\tag{4}
$$

Table 1: Multivariate forecasting results with prediction lengths $F \in \{12, 24, 36, 48\}$ for PEMS dataset while $F \in \{96, 192, 336, 720\}$ for others with fixed lookback window $T = 96$. Results are averaged from all prediction lengths. *Avg* means further averaged by subsets. Full result is left in Appendix E.1 due to space limit.

| Models | LiNo (Ours) | | iTransformer (2024b) | | RLinear (2023a) | | PatchTST (2023) | | TSMixer (2023) | | Crossformer (2023) | | TiDE (2023) | | TimesNet (2023) | | DLinear (2023) | | FEDformer (2022b) | | Autoformer (2021) | |
|---|---|---|---|---|---|---|---|---|---|---|---|---|---|---|---|---|---|---|---|---|---|---|
| Metric | MSE | MAE | MSE | MAE | MSE | MAE | MSE | MAE | MSE | MAE | MSE | MAE | MSE | MAE | MSE | MAE | MSE | MAE | MSE | MAE | MSE | MAE |
| ETT (Avg) | **0.368** | **0.387** | 0.383 | 0.399 | 0.380 | 0.392 | 0.381 | 0.397 | 0.388 | 0.402 | 0.685 | 0.578 | 0.482 | 0.470 | 0.391 | 0.404 | 0.442 | 0.444 | 0.408 | 0.428 | 0.465 | 0.459 |
| ECL | **0.164** | **0.260** | 0.178 | 0.270 | 0.219 | 0.298 | 0.205 | 0.290 | 0.186 | 0.287 | 0.244 | 0.334 | 0.251 | 0.344 | 0.192 | 0.295 | 0.212 | 0.300 | 0.214 | 0.327 | 0.227 | 0.338 |
| Exchange | **0.350** | **0.398** | 0.360 | 0.403 | 0.378 | 0.417 | 0.367 | 0.404 | 0.376 | 0.414 | 0.940 | 0.707 | 0.370 | 0.413 | 0.416 | 0.443 | 0.354 | 0.414 | 0.519 | 0.429 | 0.613 | 0.539 |
| Traffic | 0.465 | 0.296 | **0.428** | **0.282** | 0.626 | 0.378 | 0.481 | 0.304 | 0.522 | 0.357 | 0.550 | 0.304 | 0.760 | 0.473 | 0.620 | 0.336 | 0.625 | 0.383 | 0.610 | 0.376 | 0.628 | 0.379 |
| Weather | **0.241** | **0.270** | 0.258 | 0.279 | 0.272 | 0.291 | 0.259 | 0.281 | 0.256 | 0.279 | 0.259 | 0.315 | 0.271 | 0.320 | 0.259 | 0.287 | 0.265 | 0.317 | 0.309 | 0.360 | 0.338 | 0.382 |
| Solar-Energy | **0.230** | 0.270 | 0.233 | **0.262** | 0.369 | 0.356 | 0.270 | 0.307 | 0.260 | 0.297 | 0.641 | 0.639 | 0.347 | 0.417 | 0.301 | 0.319 | 0.330 | 0.401 | 0.291 | 0.381 | 0.885 | 0.711 |
| PEMS03 | **0.096** | **0.197** | 0.113 | 0.221 | 0.495 | 0.472 | 0.119 | 0.233 | 0.180 | 0.291 | 0.169 | 0.281 | 0.326 | 0.419 | 0.147 | 0.248 | 0.278 | 0.375 | 0.213 | 0.327 | 0.667 | 0.601 |
| PEMS04 | **0.098** | **0.203** | 0.111 | 0.221 | 0.526 | 0.491 | 0.103 | 0.215 | 0.195 | 0.307 | 0.209 | 0.314 | 0.353 | 0.437 | 0.129 | 0.241 | 0.295 | 0.388 | 0.231 | 0.337 | 0.610 | 0.590 |
| PEMS07 | **0.088** | **0.181** | 0.101 | 0.204 | 0.504 | 0.478 | 0.112 | 0.217 | 0.211 | 0.303 | 0.235 | 0.315 | 0.380 | 0.440 | 0.124 | 0.225 | 0.329 | 0.395 | 0.165 | 0.283 | 0.367 | 0.451 |
| PEMS08 | **0.138** | **0.217** | 0.150 | 0.226 | 0.529 | 0.487 | 0.165 | 0.261 | 0.280 | 0.321 | 0.268 | 0.307 | 0.441 | 0.464 | 0.193 | 0.271 | 0.379 | 0.416 | 0.286 | 0.358 | 0.814 | 0.659 |
| 1st Count | **9** | **8** | 1 | 2 | 0 | 0 | 0 | 0 | 0 | 0 | 0 | 0 | 0 | 0 | 0 | 0 | 0 | 0 | 0 | 0 | 0 | 0 |

## 4 EXPERIMENTS

**Datasets.** To thoroughly evaluate the performance of our proposed LiNo, we conduct extensive experiments on 13 widely used, real-world datasets including ETT (4 subsets) (Zhou et al., 2022a), Traffic, Exchange, Electricity(ECL), Weather (Wu et al., 2021), Solar-Energy(Solar) (Lai et al., 2018) and PEMS (4 subsets) (Liu et al., 2022a). Detailed descriptions of the datasets can be found in Appendix A. We select both univariate and multivariate time series forecasting tasks, ensuring a comprehensive assessment.

**Experimental setting.** All the experiments are conducted on a single NVIDIA GeForce RTX 4090 with 24G VRAM. The mean squared error (MSE) loss function is utilized for model optimization. We use the ADAM optimizer with an early stop parameter $patience = 6$. To foster reproducibility, we make our code, training scripts, and some visualization examples available in this **Anonymous Repository**[1]. Full implementation details and other information are in Appendix B.1.

### 4.1 MULTIVARIATE TIME SERIES FORECASTING RESULTS

**Compared methods and benchmarks.** We extensively compare the recent Linear-based or MLP-based methods, including DLinear (Zeng et al., 2023), TSMixer (Chen et al., 2023), TiDE (Das et al., 2023), RLinear (Li et al., 2023a). We also consider Transformer-based methods including FEDformer (Zhou et al., 2022b), Autoformer (Wu et al., 2021), PatchTST (Nie et al., 2023), Crossformer (Zhang & Yan, 2023), iTransformer (Liu et al., 2024b) and a CNN-based method TimesNet (Wu et al., 2023). These models represent the latest advancements in multivariate time series forecasting and encompass all mainstream prediction model types. The multivariate time series forecasting benchmarks follow the setting in iTransformer (Liu et al., 2024b). The lookback window is set to $T = 96$ for all datasets. We set the prediction horizon to $F \in \{12, 24, 48, 96\}$ for PEMS dataset and $F \in \{96, 192, 336, 720\}$ for others. Performance comparison among different methods is conducted based on two primary evaluation metrics: Mean Squared Error (MSE) and Mean Absolute Error (MAE). The results of TSMixer are reproduced following **Time Series Library** (Wang et al., 2024b) and other results are taken from iTransformer (Liu et al., 2024b).

**Result analysis.** As shown in Table 1, LiNo performed remarkably across 10 benchmark datasets. It achieved first place in **9** out of **10** datasets in MSE and **8** datasets in MAE, underscoring its leading position in multivariate time series forecasting tasks. LiNo successfully reduced the MSE metric by **3.41%** compared to the previous state-of-the-art method, **iTransformer**, across all 10 datasets. Notably, the PEMS datasets (*PEMS03:* 358 *variates, PEMS04:* 307 *variates, PEMS07:* 883 *variates, PEMS08:* 170 *variates*) and the ECL dataset (321 *variates*) present notorious challenges to

---
[1] https://anonymous.4open.science/r/LiNo-8225/

Table 2: Univariate forecasting results with prediction lengths $F \in \{96, 192, 336, 720\}$ and fixed lookback length $T = 96$ for all datasets. Results are averaged from all prediction lengths. Full result is left in Appendix E.2 due to space limit.

| Models | LiNo (Ours) | | MICN (2023) | | FEDformer (2022b) | | Autoformer (2021) | | Informer (2022a) | | LogTrans (2019) | |
|---|---|---|---|---|---|---|---|---|---|---|---|---|
| Metric | MSE | MAE | MSE | MAE | MSE | MAE | MSE | MAE | MSE | MAE | MSE | MAE |
| ETTm1 | **0.053** | **0.172** | 0.064 | 0.185 | 0.069 | 0.202 | 0.081 | 0.221 | 0.281 | 0.441 | 0.231 | 0.382 |
| ETTm2 | **0.118** | **0.255** | 0.131 | 0.266 | 0.119 | 0.262 | 0.130 | 0.271 | 0.175 | 0.320 | 0.130 | 0.277 |
| ETTh1 | **0.074** | **0.209** | 0.092 | 0.233 | 0.111 | 0.257 | 0.105 | 0.252 | 0.199 | 0.377 | 0.345 | 0.513 |
| ETTh2 | **0.180** | **0.332** | 0.252 | 0.390 | 0.206 | 0.350 | 0.218 | 0.364 | 0.243 | 0.400 | 0.252 | 0.408 |
| Traffic | **0.143** | **0.222** | 0.165 | 0.246 | 0.219 | 0.323 | 0.261 | 0.365 | 0.309 | 0.388 | 0.355 | 0.404 |
| Weather | **0.0016** | **0.030** | 0.0030 | 0.040 | 0.0055 | 0.058 | 0.0083 | 0.070 | 0.0033 | 0.044 | 0.0058 | 0.057 |
| 1st Count | **6** | **6** | 0 | 0 | 0 | 0 | 0 | 0 | 0 | 0 | 0 | 0 |

multivariate time series forecasting models due to their high dimensionality and complex nonlinearity. LiNo demonstrated its superiority in nonlinear pattern extraction by achieving a substantial relative decrease of **11.89%** in average MSE on the four PEMS-relevant benchmarks. On the ECL dataset, LiNo decreased the average MSE from **0.178** to **0.164**, representing a significant reduction of about **7.87%**.

Previous research suggests that simple linear models can outperform complex deep neural networks in certain scenarios (Zeng et al., 2023; Li et al., 2023a). For instance, in scenarios where the dataset displays clear nonlinear patterns, nonlinear models like iTransformer excel. However, on the ETT datasets (four subsets), **RLinear**, which consists of a linear layer combined with ReVIN (Kim et al., 2022), easily surpassed all previous sophisticated deep models. We argue that this is because most of these models focus solely on either linear or nonlinear patterns, neglecting the other, leading to inconsistent performance across different scenarios. In contrast, LiNo performs outstandingly across various scenarios, demonstrating the importance of a balanced approach to handling both linear and nonlinear patterns.

### 4.2 Univariate Time Series Forecasting Results

**Compared methods and benchmarks.** The models and results used for comparing univariate time series forecasting performance were collected from MICN (Wang et al., 2023), including MICN (Wang et al., 2023), FEDformer (Zhou et al., 2022b), Autoformer (Wu et al., 2021), Informer (Zhou et al., 2022a), LogTrans (Li et al., 2019). We follow the setting in MICN (Liu et al., 2024b) where the lookback window length is set to $T = 96$ and the prediction horizon to $F \in \{96, 192, 336, 720\}$ for all datasets.

**Result analysis.** Table 2 demonstrates the top-notch performance of LiNo in univariate time series forecasting tasks, achieving the best predictive results across all 6 datasets. On six datasets, LiNo reduced the MSE metric by **19.37%** and the MAE by **10.28%** compared to the previous SOTA method, MICN. Notably, on the Weather, ETTh2, and Traffic datasets, the MSE decreased by **47.11%**, **28.64%**, and **12.97%**, respectively, marking a significant improvement. The consistent superior advancement in both univariate and multivariate time series forecasting demonstrates the wide applicability of LiNo across various scenarios.

### 4.3 LiNo Analysis

**Ablation study on LiNo components.** To verify the effectiveness of LiNo components, we conducted ablation studies by removing components (w/o) on 7 multivariate time series forecasting benchmarks with a lookback window of $T = 96$ and prediction lengths $F \in \{96, 720\}$. The results are presented in Table 3. Every design in LiNo is crucial. Removing the No block results in significant performance degradation, with an MSE increase of up to **71.82%**. Similarly, the absence of the Li block leads to a **10.00%** rise in MSE, underscoring the importance of modeling both linear and nonlinear patterns. Further ablation of the No block reveals that temporal variation and frequency information extraction are essential. The absence of inter-series dependencies modeling results in a **15.91%** increase in MSE, highlighting its critical importance.

Table 3: Ablations on LiNo. 'w/o' means remove this design. 'TE' stands for temporal variations extraction, 'FE' strands for frequency information extraction, and 'CD' means the channel mixing step for inter-series dependencies modeling.

| Dataset | | LiNo | | w/o Li Block | | w/o No Block | | w/o TE | | w/o FE | | w/o CD | |
|---|---|---|---|---|---|---|---|---|---|---|---|---|---|
| Metric | | MSE | MAE | MSE | MAE | MSE | MAE | MSE | MAE | MSE | MAE | MSE | MAE |
| ECL | 96 | 0.138 | 0.233 | 0.149 | 0.242 | 0.186 | 0.267 | 0.143 | 0.241 | 0.140 | 0.237 | 0.154 | 0.248 |
| | 720 | 0.191 | 0.290 | 0.210 | 0.300 | 0.245 | 0.320 | 0.197 | 0.293 | 0.196 | 0.294 | 0.223 | 0.311 |
| Weather | 96 | 0.154 | 0.199 | 0.211 | 0.300 | 0.162 | 0.206 | 0.158 | 0.204 | 0.159 | 0.204 | 0.164 | 0.209 |
| | 720 | 0.343 | 0.342 | 0.355 | 0.348 | 0.342 | 0.338 | 0.347 | 0.344 | 0.345 | 0.343 | 0.347 | 0.345 |
| ETTm2 | 96 | 0.171 | 0.254 | 0.177 | 0.260 | 0.180 | 0.261 | 0.173 | 0.256 | 0.173 | 0.255 | 0.175 | 0.257 |
| | 720 | 0.395 | 0.393 | 0.399 | 0.396 | 0.409 | 0.400 | 0.399 | 0.396 | 0.397 | 0.395 | 0.391 | 0.392 |
| ETTh2 | 96 | 0.292 | 0.340 | 0.295 | 0.344 | 0.301 | 0.348 | 0.294 | 0.342 | 0.294 | 0.341 | 0.291 | 0.340 |
| | 720 | 0.422 | 0.441 | 0.416 | 0.436 | 0.429 | 0.446 | 0.424 | 0.442 | 0.426 | 0.443 | 0.424 | 0.442 |
| PEMS04 | 12 | 0.069 | 0.169 | 0.083 | 0.186 | 0.121 | 0.232 | 0.070 | 0.173 | 0.070 | 0.171 | 0.083 | 0.186 |
| | 96 | 0.137 | 0.247 | 0.278 | 0.359 | 1.016 | 0.762 | 0.149 | 0.261 | 0.146 | 0.259 | 0.296 | 0.349 |
| PEMS08 | 12 | 0.070 | 0.166 | 0.076 | 0.176 | 0.119 | 0.230 | 0.075 | 0.180 | 0.074 | 0.176 | 0.085 | 0.189 |
| | 96 | 0.247 | 0.283 | 0.359 | 0.359 | 1.075 | 0.771 | 0.258 | 0.299 | 0.253 | 0.292 | 0.431 | 0.384 |
| Solar-Energy | 96 | 0.200 | 0.250 | 0.238 | 0.310 | 0.338 | 0.258 | 0.203 | 0.257 | 0.204 | 0.258 | 0.226 | 0.271 |
| | 720 | 0.250 | 0.283 | 0.251 | 0.395 | 0.369 | 0.292 | 0.260 | 0.296 | 0.267 | 0.297 | 0.277 | 0.297 |
| avg-promote | | 0 | 0 | -10.00% | -6.48% | -71.82% | -35.97% | -2.27% | -2.52% | -2.27% | -1.80% | -15.91% | -8.27% |

Table 4: Impact of the number of LiNo blocks (layers) $N$ on the model's performance. The task is **input-96-predict-96** for PEMS04&08, and **input-96-predict-720** for others. We set $N \in \{1, 2, 3, 4\}$. The best results are bold in red.

| Number of LiNo blocks | | ETTh1 | | ETTh2 | | ECL | | Weather | | Solar | | PEMS04 | | PEMS08 | |
|---|---|---|---|---|---|---|---|---|---|---|---|---|---|---|---|
| Value \Metric | | MSE | MAE | MSE | MAE | MSE | MAE | MSE | MAE | MSE | MAE | MSE | MAE | MSE | MAE |
| N | 1 | 0.472 | 0.466 | 0.427 | 0.446 | 0.197 | 0.295 | 0.346 | 0.344 | 0.268 | 0.299 | 0.152 | 0.262 | 0.267 | 0.308 |
| | 2 | 0.471 | 0.466 | **0.421** | **0.440** | **0.191** | **0.290** | **0.343** | **0.342** | **0.250** | **0.283** | 0.145 | 0.256 | **0.247** | **0.283** |
| | 3 | **0.459** | **0.456** | 0.423 | 0.442 | 0.192 | 0.292 | 0.346 | 0.344 | 0.255 | 0.285 | 0.143 | 0.250 | 0.258 | 0.296 |
| | 4 | 0.468 | 0.463 | 0.424 | 0.442 | 0.194 | 0.293 | 0.347 | 0.345 | 0.257 | 0.286 | **0.137** | **0.247** | 0.258 | 0.296 |

Table 5: Further ablation study on the No Block. 'Temporal' means only extracting temporal variations (without introducing nonlinearity), 'Frequency' strands for frequency information extraction (without introducing nonlinearity), and 'TF' means these two patterns are captured simultaneously. The nonlinearity of the No Block is introduced by adding **ReLU** or **Tanh** activation function.

| Models | | Temporal | | Frequency | | TF | | ReLU | | Tanh | |
|---|---|---|---|---|---|---|---|---|---|---|---|
| Metric | | MSE | MAE | MSE | MAE | MSE | MAE | MSE | MAE | MSE | MAE |
| ETTh1 | 96 | 0.385 | 0.398 | 0.383 | 0.393 | 0.383 | **0.392** | 0.384 | 0.398 | **0.375** | 0.394 |
| | 720 | 0.487 | 0.475 | 0.482 | 0.469 | 0.478 | 0.467 | 0.488 | 0.477 | **0.464** | **0.458** |
| ETTm1 | 96 | 0.341 | 0.369 | 0.342 | 0.369 | 0.338 | 0.366 | 0.328 | 0.365 | **0.322** | **0.359** |
| | 720 | 0.481 | 0.447 | 0.479 | 0.445 | 0.474 | **0.439** | 0.478 | 0.445 | **0.465** | 0.442 |
| ECL | 96 | 0.186 | 0.268 | 0.187 | 0.267 | 0.183 | 0.263 | 0.154 | 0.248 | **0.150** | **0.243** |
| | 720 | 0.245 | 0.320 | 0.245 | 0.320 | 0.241 | 0.317 | 0.223 | 0.311 | **0.221** | **0.308** |
| Weather | 96 | 0.165 | 0.209 | 0.165 | 0.208 | **0.162** | **0.206** | 0.162 | 0.208 | 0.163 | 0.209 |
| | 720 | 0.344 | 0.344 | 0.345 | 0.343 | **0.342** | **0.338** | 0.345 | 0.345 | 0.343 | 0.339 |

**Further ablation on No Block.** Considering that in real-world scenarios, nonlinear temporal variations and frequency information are more prevalent than inter-series dependencies in most cases, we conducted further ablation studies. Specifically, without considering inter-series dependencies, we simplified the No block to retain only the Temporal&Frequency Extraction component (**TF_Extraction**) shown in Figure 2 (left), and performed additional ablation experiments.

Combining Temporal and Frequency projections enhances expressiveness, improving the model's ability to learn complex patterns. **Tanh** provides a smooth, symmetric activation function that better captures the complex dependencies in both time and frequency domains, therefore delivering a better performance than the more commonly used **ReLU**, as in Table 5.

**Sensitivity to the number of LiNo blocks.** We investigate the impact of the number of LiNo blocks (layers) $N$ on the model's performance, as shown in Table 4. The best forecasting results for each dataset are generally achieved when $N > 1$, indicating the necessity of deeper RRD. The variation in optimal $N$ across datasets suggests that LiNo can flexibly adapt to different RRD requirements.

Table 6: Ablation study of different No block choice and different model design.

(a) Ablation study of different No block choice. '→ iTransformer' means replacing the proposed No block with iTransformer. Same to '→ TSMixer'. The input sequence length is set to $T = 96$ for all tasks.

| Models | | Ori LiNo | | → iTransformer | | → TSMixer | |
|---|---|---|---|---|---|---|---|
| Metric | | MSE | MAE | MSE | MAE | MSE | MAE |
| ETTm2 | 96 | **0.171** | **0.254** | 0.174 | 0.259 | 0.173 | 0.256 |
| | 192 | **0.237** | **0.298** | 0.238 | 0.299 | 0.239 | 0.301 |
| | 336 | **0.296** | **0.336** | 0.302 | 0.340 | 0.304 | 0.343 |
| | 720 | **0.395** | **0.393** | 0.402 | 0.398 | 0.406 | 0.402 |
| | Avg | **0.275** | **0.320** | 0.279 | 0.324 | 0.281 | 0.326 |
| ECL | 96 | **0.138** | **0.233** | 0.141 | 0.239 | 0.145 | 0.249 |
| | 192 | **0.155** | **0.250** | 0.163 | 0.254 | 0.164 | 0.263 |
| | 336 | **0.171** | **0.267** | 0.175 | 0.271 | 0.187 | 0.285 |
| | 720 | **0.191** | **0.290** | 0.201 | 0.295 | 0.228 | 0.320 |
| | Avg | **0.164** | **0.260** | 0.170 | 0.265 | 0.181 | 0.279 |
| Weather | 96 | **0.154** | **0.199** | 0.156 | 0.201 | 0.157 | 0.203 |
| | 192 | **0.205** | **0.248** | 0.206 | 0.248 | 0.206 | 0.250 |
| | 336 | **0.262** | **0.290** | 0.264 | 0.291 | 0.267 | 0.295 |
| | 720 | **0.343** | **0.342** | 0.345 | 0.343 | 0.349 | 0.347 |
| | Avg | **0.241** | **0.270** | 0.243 | 0.271 | 0.245 | 0.274 |

(b) To compare the performance of different forecasting model designs, we choose **iTransformer** as the backbone and sequentially employ 'Raw', 'Mu', and **LiNo**. The input sequence length is set to $T = 96$.

| Models | | LiNo | | Mu | | Raw | |
|---|---|---|---|---|---|---|---|
| Metric | | MSE | MAE | MSE | MAE | MSE | MAE |
| ETTm2 | 96 | **0.174** | **0.259** | 0.179 | 0.264 | 0.184 | 0.268 |
| | 192 | **0.238** | **0.299** | 0.243 | 0.304 | 0.247 | 0.307 |
| | 336 | **0.302** | **0.340** | 0.307 | 0.345 | 0.311 | 0.348 |
| | 720 | **0.402** | **0.398** | 0.406 | 0.400 | 0.408 | 0.402 |
| | Promote | **-2.96%** | **-2.19%** | -1.30% | -0.91% | 0 | 0 |
| ECL | 96 | **0.141** | **0.239** | 0.154 | 0.246 | 0.153 | 0.245 |
| | 192 | **0.163** | **0.254** | 0.167 | 0.259 | 0.166 | 0.257 |
| | 336 | **0.175** | **0.271** | 0.183 | 0.276 | 0.183 | 0.275 |
| | 720 | **0.201** | **0.295** | 0.220 | 0.309 | 0.224 | 0.310 |
| | Promote | **-6.34%** | **-2.58%** | -0.28% | 0.28% | 0 | 0 |
| Weather | 96 | **0.156** | **0.201** | 0.174 | 0.214 | 0.178 | 0.217 |
| | 192 | **0.206** | **0.248** | 0.223 | 0.257 | 0.224 | 0.258 |
| | 336 | **0.264** | **0.291** | 0.278 | 0.298 | 0.281 | 0.299 |
| | 720 | **0.345** | **0.343** | 0.355 | 0.350 | 0.358 | 0.352 |
| | Promote | **-6.72%** | **-3.82%** | -1.06% | -0.62% | 0 | 0 |

**Superiority of the proposed No block.** Extracting nonlinear patterns, such as inter-series dependencies, temporal variations, and frequency information, is crucial for accurate predictions. To demonstrate the competence of the proposed No block, we sequentially replaced it with two renowned nonlinear time series forecasting models: **iTransformer** and **TSMixer**. As shown in Table 6 (a), our proposed No block consistently outperforms other nonlinear pattern extractors. It delivers superior performance across ETTm2 (7 variates), Weather (21 variates), and ECL (321 variates), showcasing its remarkable ability to extract nonlinear patterns.

Further analysis of LiNo is provided in Appendix C.

## 4.4 Analysis of Different Forecasting Model Designs

**Forecasting performance comparison.** We use **iTransformer**, a leading transformer-based time series forecasting model, as the backbone. We evaluate three model designs: 'Raw' (classical design), 'Mu' (recursive splitting of representations for prediction, design used in N-BEATS (Oreshkin et al., 2020)), and 'LiNo' (our proposed framework with further recursive splitting into linear and nonlinear patterns). The forecasting performance in Table 6 (b) demonstrates the effectiveness of the LiNo framework. Compared to 'Raw', 'Mu' reduces the MSE on ETTm2, ECL, and Weather by 1.3%, 0.28%, and 1.06%, respectively. LiNo further reduces the MSE by 2.96%, 6.34%, and 6.72%. These results indicate that while 'Mu' improves forecasting performance, it remains suboptimal due to the entanglement of linear and nonlinear predictions. LiNo effectively separates these patterns, achieving more accurate results.

**Noise robustness.** To investigate the robustness of different forecasting model designs to noise, we conducted experiments using **iTransformer** as the backbone. Given an input multivariate time series signal $X \in \mathbb{R}^{B \times T \times N}$, we added Gaussian noise to obtain: $\hat{X} = X + \alpha \cdot \text{noise}$, where $\alpha \in \{0\%, 25\%, 50\%, 75\%, 100\%\}$ is the noise intensity coefficient, and noise $\in \mathbb{R}^{B \times T \times N}$ is Gaussian noise with mean 0 and standard deviation 1. The noisy input $\hat{X}$ was used during training. A more robust forecasting model will be less affected by this noise.

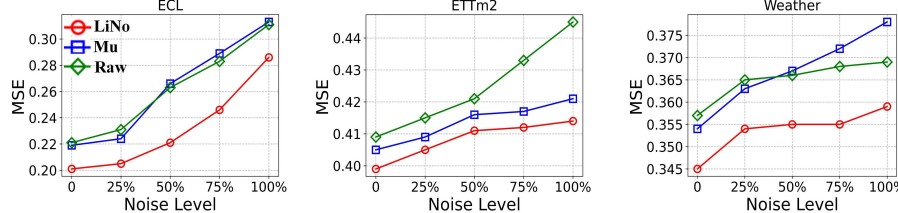

Figure 3: Multivariate forecasting performance of three different model designs using **iTransformer** as backbone under different noise levels across datasets of ECL, ETTm2, and Weather.

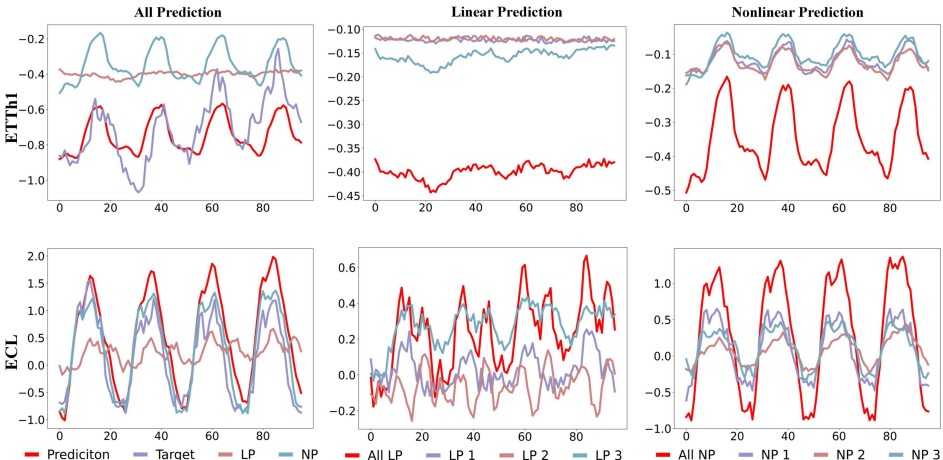

Figure 4: Visualization of multivariate forecasting result (***last channel/variate***) of the proposed **LiNo** on ETTh1 and ECL datasets. We set the number of LiNo blocks (layers) to $N = 3$. The task is multivariate time series forecasting with input $T = 96$ and target $F = 96$. 'LP' denotes Linear prediction, and 'NP' stands for Nonlinear prediction. LP $i$ or NP $i$ ($i \in \{1, 2, 3\}$) is the linear or nonlinear prediction of $i$-th layer (level).

Our LiNo design consistently outperforms the 'Mu' and 'Raw' models across various noise levels, demonstrating superior robustness and reliability in forecasting, as shown in Figure 3. This result supports our hypothesis that separating linear and nonlinear patterns enhances model robustness.

**Visualization of linear and nonlinear predictions.** We visualize LiNo's forecasting results on the ETTh1 and ECL datasets in Figure 4. The Li block primarily captures linear patterns such as long-term trends, while the No block effectively captures nonlinear signals like fluctuations and seasonality. The forecasting results for ECL and ETTh1 reveal three distinct linear modes and three different nonlinear patterns, enhancing the interpretability of time series forecasting and aiding in understanding the underlying data dynamics. Additional visualization results of LiNo can be found in Appendix D.

## 5 CONCLUSION

The commonly used seasonal and trend decomposition (STD) in previous methods still rely on flawed linear pattern extractors. Their lack of separating the nonlinear component from residual and shallow-level decomposition severely hinders its modeling capability. This work advances them by incorporating a more general linear extraction model and introducing a novel and powerful nonlinear extraction model into RRD. By performing RRD at a deeper and more nuanced level, we achieve a more refined decomposition, leading to more accurate and robust forecasting results. The proposed No block excels in capturing nonlinear features. Experiments across multiple benchmarks demonstrated LiNo's superior performance in both univariate and multivariate forecasting tasks, offering improved accuracy and stability. These findings could offer opportunities to design more robust and precise forecasting models.

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

# A   DATASETS DESCRIPTION

Table 7: Detailed dataset descriptions. *Channels* denotes the number of channels in each dataset. *Dataset Split* denotes the total number of time points in (Train, Validation, Test) split respectively. *Prediction Length* denotes the future time points to be predicted, and four prediction settings are included in each dataset. *Granularity* denotes the sampling interval of time points.

| Dataset | Channels | Prediction Length | Dataset Split | Granularity | Domain |
|---------|----------|-------------------|---------------|-------------|--------|
| ETTh1, ETTh2 | 7 | {96, 192, 336, 720} | (8545, 2881, 2881) | Hourly | Electricity |
| ETTm1, ETTm2 | 7 | {96, 192, 336, 720} | (34465, 11521, 11521) | 15min | Electricity |
| Exchange | 8 | {96, 192, 336, 720} | (5120, 665, 1422) | Daily | Economy |
| Weather | 21 | {96, 192, 336, 720} | (36792, 5271, 10540) | 10min | Weather |
| ECL | 321 | {96, 192, 336, 720} | (18317, 2633, 5261) | Hourly | Electricity |
| Traffic | 862 | {96, 192, 336, 720} | (12185, 1757, 3509) | Hourly | Transportation |
| Solar-Energy | 137 | {96, 192, 336, 720} | (36601, 5161, 10417) | 10min | Energy |
| PEMS03 | 358 | {12, 24, 48, 96} | (15617,5135,5135) | 5min | Transportation |
| PEMS04 | 307 | {12, 24, 48, 96} | (10172,3375,281) | 5min | Transportation |
| PEMS07 | 883 | {12, 24, 48, 96} | (16911,5622,468) | 5min | Transportation |
| PEMS08 | 170 | {12, 24, 48, 96} | (10690,3548,265) | 5min | Transportation |

We elaborate on the datasets employed in this study with the following details.

1. **ETT (Electricity Transformer Temperature)** Zhou et al. (2022a) [2] comprises two hourly-level datasets (ETTh) and two 15-minute-level datasets (ETTm). Each dataset contains seven oil and load features of electricity transformers from July 2016 to July 2018.

2. **Exchange** (Wu et al., 2021) [3] collects the panel data of daily exchange rates from 8 countries from 1990 to 2016.

3. **Traffic** (Wu et al., 2021) [4] describes the road occupancy rates. It contains the hourly data recorded by the sensors of San Francisco freeways from 2015 to 2016.

4. **Electricity** (Wu et al., 2021) [5] collects the hourly electricity consumption of 321 clients from 2012 to 2014.

5. **Weather** (Wu et al., 2021) [6] includes 21 indicators of weather, such as air temperature, and humidity. Its data is recorded every 10 min for 2020 in Germany.

6. **Solar-Energy** Lai et al. (2018) [7] records the solar power production of 137 PV plants in 2006, which is sampled every 10 minutes.

7. **PEMS** (Liu et al., 2022a) [8] contains public traffic network data in California collected by 5-minute windows.

We follow the same data processing and train-validation-test set split protocol used in iTransformer (Liu et al., 2024b), where the train, validation, and test datasets are strictly divided according to chronological order to make sure there are no data leakage issues. We fix the length of the lookback series as $T = 96$ for all datasets, and the prediction length $F \in \{12, 24, 48, 96\}$ for PEMS datasets, and $F \in \{96, 192, 336, 720\}$ for others. Other details of these datasets is concluded in Table 7.

---

[2] https://github.com/zhouhaoyi/ETDataset
[3] https://github.com/thuml/iTransformer
[4] http://pems.dot.ca.gov
[5] https://archive.ics.uci.edu/ml/datasets/ElectricityLoadDiagrams20112014
[6] https://www.bgc-jena.mpg.de/wetter/
[7] https://github.com/thuml/iTransformer
[8] https://pems.dot.ca.gov/

## B  IMPLEMENT DETAILS

### B.1  EXPERIMENT DETAILS

To foster reproducibility, we make our code, training scripts, and some visualization examples available in this **Anonymous Repository**[9]. All the experiments are conducted on a single NVIDIA GeForce RTX 4090 with 24G VRAM. The mean squared error (MSE) loss function is utilized for model optimization. We use the ADAM optimizer with an early stop parameter $patience = 6$. We explore the number of LiNo blocks $N \in \{1, 2, 3, 4\}$, dropout ratio $dp \in \{0.0, 0.2, 0.5\}$, and the dimension of layers $dim \in \{256, 512\}$. The $learning\ rate \in \{1e-3, 1e-4, 1e-5\}$ and $batch\ size \in \{32, 64, 128, 256\}$ are adjusted based on the size and dimensionality of the dataset, as well as the specific conditions of our experimental setup. All the compared multivariate forecasting baseline models that we reproduced are implemented based on the benchmark of **Time series Lab** (Wang et al., 2024b) Repository [10], which is fairly built on the configurations provided by each model's original paper or official code.

Performance comparison among different methods is conducted based on two primary evaluation metrics: Mean Squared Error (MSE) and Mean Absolute Error (MAE). The formula is below: **Mean Squared Error (MSE)**:

$$\text{MSE} = \frac{1}{F} \sum_{i=1}^{F} (\mathbf{Y}_i - \hat{\mathbf{Y}}_i)^2. \tag{5}$$

**Mean Absolute Error (MAE)**:

$$\text{MAE} = \frac{1}{F} \sum_{i=1}^{F} |\mathbf{Y}_i - \hat{\mathbf{Y}}_i|. \tag{6}$$

where $\mathbf{Y}, \hat{\mathbf{Y}} \in \mathbb{R}^{F \times C}$ are the ground truth and prediction results of the future with $F$ time points and $C$ channels. $\mathbf{Y}_i$ denotes the $i$-th future time point.

## C  FURTHER MODEL ANALYSIS

### C.1  MODEL ROBUSTNESS

Table 8: Error Bar ($Mean \pm Std$) of LiNo's multivariate forecasting result on ETTh2, ETTm2, Weather, PEMS04, PEMS08. We set lookback window $T = 96$, and prediction length $F \in \{12, 24, 48, 96\}$ for PEMS04 and PEMS08, and $F \in \{96, 192, 336, 720\}$ for others. Mean and standard deviation were obtained on 5 runs with different random seeds.

| Models | ETTh2 | | ETTm2 | | Weather | | PEMS04 | | PEMS08 | |
|---|---|---|---|---|---|---|---|---|---|---|
| Metric | MSE | MAE | MSE | MAE | MSE | MAE | MSE | MAE | MSE | MAE |
| 96/12 | 0.293 ± 0.0017 | 0.341 ± 0.0012 | 0.172 ± 0.0004 | 0.254 ± 0.0005 | 0.156 ± 0.0013 | 0.201 ± 0.0013 | 0.069 ± 0.0002 | 0.168 ± 0.0009 | 0.071 ± 0.0004 | 0.169 ± 0.0026 |
| 192/24 | 0.377 ± 0.0015 | 0.392 ± 0.0008 | 0.238 ± 0.0006 | 0.298 ± 0.0005 | 0.206 ± 0.0021 | 0.248 ± 0.0019 | 0.081 ± 0.0015 | 0.186 ± 0.0031 | 0.094 ± 0.0011 | 0.191 ± 0.0027 |
| 336/48 | 0.417 ± 0.0009 | 0.426 ± 0.0005 | 0.297 ± 0.0005 | 0.336 ± 0.0004 | 0.265 ± 0.0020 | 0.291 ± 0.0017 | 0.104 ± 0.0027 | 0.214 ± 0.0036 | 0.139 ± 0.0036 | 0.227 ± 0.0064 |
| 720/96 | 0.422 ± 0.0007 | 0.440 ± 0.0007 | 0.395 ± 0.0011 | 0.393 ± 0.0009 | 0.343 ± 0.0016 | 0.342 ± 0.0012 | 0.139 ± 0.0021 | 0.247 ± 0.0031 | 0.254 ± 0.0079 | 0.296 ± 0.0091 |

We provide LiNo's Error Bar ($Mean \pm Std$) on several representative datasets in Table 8. LiNo demonstrated a relatively lower Error Bar across results with different random seeds, indicating consistent performance, high stability, and considerable generalization ability.

### C.2  MODEL EFFICIENCY

We evaluated the **parameter count**, and the **inference time** of four cutting-edge transformer-based multivariate time series forecasting models: **iTransformer**, **Crossformer**, **PatchTST**, and **FED-former**. Results can be found in Table 9. Although LiNo introduces slightly more trainable parameters compared to iTransformer well-known for its simple and efficient design, its inference speed and prediction performance are significantly superior to iTransformer.

---

[9] https://anonymous.4open.science/r/LiNo-8225/
[10] https://github.com/thuml/Time-Series-Library

Table 9: Model efficiency analysis. We evaluated the **parameter count**, and the **inference time** (average of 5 runs on a single NVIDIA 4090 24GB GPU) with $batch\_size = 1$ on **ETTh1** and **ECL** dataset. We set the dimension of layer $dim \in \{256, 512\}$, and the number of network layers $N = 2$. The task is **input-96-forecast-720**. **\*** means 'former.' **Para** means 'Parameter count(M).' **Time** means 'inference time(ms).'

| Datasets/Models | dim | LiNo | | PatchTST | | Cross* | | iTrans* | | FED* | |
|---|---|---|---|---|---|---|---|---|---|---|---|
| | | Param | Time | Para | Time | Para | Time | Para | Time | Para | Time |
| ETTh1 | 256 | 1.59 | **143.81** | 3.27 | 251.00 | 8.19 | 399.00 | **1.27** | 177.67 | 3.43 | 303.556 |
| | 512 | 4.82 | **145.68** | 8.64 | 266.66 | 32.11 | 445.74 | **4.63** | 190.92 | 13.68 | 345.736 |
| Electricity | 256 | 1.75 | **151.33** | 3.27 | 322.53 | 13.66 | 432.40 | **1.27** | 192.12 | 4.24 | 347.634 |
| | 512 | 5.14 | **152.24** | 8.64 | 411.96 | 43.04 | 507.54 | **4.63** | 249.60 | 15.29 | 398.599 |

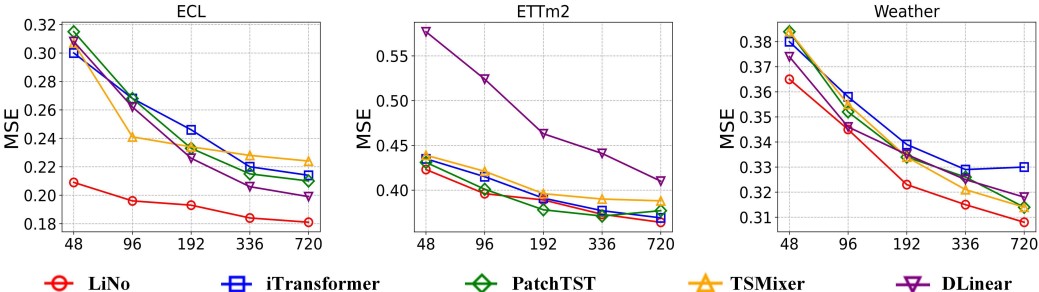

Figure 5: Multivariate forecasting performance improves with the increase of lookback window $T \in \{48, 96, 192, 336, 720\}$ and a fixed prediction length $F = 720$. Notably, LiNo consistently and stably enhances its forecasting performance as the lookback window size increases.

## C.3 INCREASING LOOKBACK LENGTH

It is generally expected that increasing the input length will enhance forecasting performance by incorporating more information (Zeng et al., 2023). This improvement is typically observed in linear forecasts, supported by statistical methods (Liu et al., 2023) that utilize extended historical data. Figure 5 evaluates the performance of LiNo and other prestigious baselines. LiNo effectively leverages longer lookback windows, showing a positive correlation between predictive performance and input length. It significantly outperforms other baselines on the Weather and ECL datasets and achieves comparable results on the ETTm2 dataset.

## C.4 COMPARE WITH EXTENSIVE BASELINES

After careful consideration, we additionally selected SegRNN (Lin et al., 2023) as an RNN-based baseline and considered another well-known work, TimeMixer (Wang et al., 2024a), which is also based on trend (linear) and seasonal (nonlinear) decomposition.

We noticed that **Official Implementation Repository of SegRNN** (Lin et al., 2023) [11] added their experimental results with the input length of $T = 96$. Therefore, we directly referenced those results.

For TimeMixer, we still utilized the well-known **Time series Lab** (Wang et al., 2024b) Repository [12], authored by the same team as TimeMixer. We re-ran the models using this updated implementation.

LiNo still consistently outperforms overall, achieving leading results in **15/28 cases for MSE** and **19/28 cases for MAE**, as illustrated in Table 10.

## C.5 INFLUENCE OF LOSS FUNCTION ON PROPOSED METHOD

The experimental results in Table 11 show the performance of LiNo and SegRNN (Lin et al., 2023) trained with MAE or MSE, respectively. When taking the average of the two loss functions (MSE and MAE), our LiNo still significantly outperforms SegRNN. Notably, the results obtained using

---

[11] https://github.com/lss-1138/SegRNN
[12] https://github.com/thuml/Time-Series-Library

Table 10: Multivariate forecasting results comparison between LiNo, TimeMixer, and SegRNN models with input length fixed to $T = 96$. The **bold** values indicate the best performance for each dataset.

| Models | | LiNo | | TimeMixer | | SegRNN | |
|---|---|---|---|---|---|---|---|
| Metric | | MSE | MAE | MSE | MAE | MSE | MAE |
| ETTm1 | 96 | **0.322** | **0.361** | 0.328 | 0.367 | 0.330 | 0.369 |
| | 192 | **0.365** | **0.383** | 0.369 | 0.389 | 0.369 | 0.392 |
| | 336 | 0.401 | **0.408** | 0.404 | 0.411 | **0.399** | 0.412 |
| | 720 | 0.469 | 0.447 | 0.473 | 0.451 | **0.454** | **0.443** |
| ETTm2 | 96 | **0.171** | **0.254** | 0.176 | 0.259 | 0.173 | 0.255 |
| | 192 | **0.237** | **0.298** | 0.242 | 0.303 | 0.237 | 0.298 |
| | 336 | **0.296** | **0.336** | 0.303 | 0.339 | 0.296 | 0.336 |
| | 720 | 0.395 | **0.393** | 0.396 | 0.399 | **0.389** | 0.407 |
| ETTh1 | 96 | 0.378 | 0.395 | 0.384 | 0.400 | **0.368** | 0.395 |
| | 192 | 0.423 | 0.423 | 0.437 | 0.429 | **0.408** | 0.419 |
| | 336 | 0.455 | **0.438** | 0.472 | 0.446 | **0.444** | 0.440 |
| | 720 | 0.459 | **0.456** | 0.508 | 0.489 | **0.446** | 0.457 |
| ETTh2 | 96 | 0.292 | 0.340 | 0.297 | 0.348 | **0.278** | 0.335 |
| | 192 | 0.375 | 0.391 | 0.369 | 0.392 | **0.359** | 0.389 |
| | 336 | **0.418** | 0.426 | 0.427 | 0.435 | 0.421 | 0.436 |
| | 720 | **0.422** | **0.441** | 0.442 | 0.461 | 0.432 | 0.455 |
| ECL | 96 | **0.138** | **0.233** | 0.153 | 0.244 | 0.151 | 0.245 |
| | 192 | **0.155** | **0.250** | 0.168 | 0.259 | 0.164 | 0.258 |
| | 336 | **0.171** | **0.267** | 0.185 | 0.275 | 0.180 | 0.277 |
| | 720 | **0.191** | **0.290** | 0.227 | 0.312 | 0.218 | 0.313 |
| Traffic | 96 | 0.429 | 0.276 | 0.473 | 0.287 | **0.419** | **0.269** |
| | 192 | 0.450 | 0.289 | 0.486 | 0.294 | **0.434** | **0.276** |
| | 336 | 0.468 | 0.297 | 0.488 | 0.298 | **0.450** | **0.284** |
| | 720 | 0.514 | 0.320 | 0.536 | 0.314 | **0.483** | **0.302** |
| Weather | 96 | **0.154** | **0.199** | 0.162 | 0.208 | 0.165 | 0.227 |
| | 192 | **0.205** | **0.248** | 0.208 | 0.252 | 0.211 | 0.273 |
| | 336 | **0.262** | **0.290** | 0.263 | 0.293 | 0.270 | 0.318 |
| | 720 | **0.343** | **0.342** | 0.345 | 0.345 | 0.357 | 0.376 |
| **1st Count** | | **15** | **19** | 0 | 0 | 13 | 9 |

Table 11: Multivariate forecasting results comparison of experimental results between LiNo (**Ours**) and SegRNN (Lin et al., 2023) with input length fixed to $T = 96$, using MSE or MAE as loss functions. The average of both loss functions is denoted as **AVG**.

| Models | | LiNo (MSE) | | LiNo (MAE) | | SegRNN (MSE) | | SegRNN (MAE) | | LiNo (AVG) | | SegRNN (AVG) | |
|---|---|---|---|---|---|---|---|---|---|---|---|---|---|
| Metric | | MSE | MAE | MSE | MAE | MSE | MAE | MSE | MAE | MSE | MAE | MSE | MAE |
| ETTm1 | 96 | 0.322 | 0.361 | 0.310 | 0.339 | 0.342 | 0.379 | 0.330 | 0.369 | 0.316 | 0.350 | 0.336 | 0.374 |
| | 192 | 0.365 | 0.383 | 0.363 | 0.368 | 0.383 | 0.402 | 0.369 | 0.392 | 0.364 | 0.376 | 0.376 | 0.397 |
| | 336 | 0.401 | 0.408 | 0.396 | 0.388 | 0.407 | 0.420 | 0.399 | 0.412 | 0.399 | 0.398 | 0.403 | 0.416 |
| | 720 | 0.469 | 0.447 | 0.451 | 0.437 | 0.471 | 0.455 | 0.454 | 0.443 | 0.460 | 0.442 | 0.463 | 0.449 |
| ETTm2 | 96 | 0.171 | 0.254 | 0.170 | 0.248 | 0.176 | 0.259 | 0.173 | 0.255 | 0.171 | 0.251 | 0.175 | 0.257 |
| | 192 | 0.237 | 0.298 | 0.233 | 0.291 | 0.241 | 0.305 | 0.237 | 0.298 | 0.235 | 0.295 | 0.239 | 0.302 |
| | 336 | 0.296 | 0.336 | 0.291 | 0.329 | 0.301 | 0.346 | 0.296 | 0.336 | 0.294 | 0.333 | 0.299 | 0.341 |
| | 720 | 0.395 | 0.393 | 0.386 | 0.392 | 0.425 | 0.436 | 0.389 | 0.407 | 0.391 | 0.393 | 0.407 | 0.422 |
| ETTh1 | 96 | 0.378 | 0.395 | 0.369 | 0.388 | 0.385 | 0.411 | 0.368 | 0.395 | 0.374 | 0.392 | 0.377 | 0.403 |
| | 192 | 0.423 | 0.423 | 0.419 | 0.417 | 0.434 | 0.441 | 0.408 | 0.419 | 0.421 | 0.420 | 0.421 | 0.430 |
| | 336 | 0.455 | 0.438 | 0.449 | 0.436 | 0.462 | 0.463 | 0.444 | 0.440 | 0.452 | 0.437 | 0.453 | 0.452 |
| | 720 | 0.459 | 0.456 | 0.453 | 0.451 | 0.497 | 0.488 | 0.446 | 0.457 | 0.456 | 0.454 | 0.472 | 0.473 |
| ETTh2 | 96 | 0.292 | 0.340 | 0.281 | 0.333 | 0.284 | 0.340 | 0.278 | 0.335 | 0.287 | 0.337 | 0.281 | 0.338 |
| | 192 | 0.375 | 0.391 | 0.366 | 0.382 | 0.375 | 0.396 | 0.359 | 0.389 | 0.371 | 0.387 | 0.367 | 0.393 |
| | 336 | 0.418 | 0.426 | 0.408 | 0.421 | 0.425 | 0.437 | 0.421 | 0.436 | 0.413 | 0.424 | 0.423 | 0.437 |
| | 720 | 0.422 | 0.441 | 0.409 | 0.429 | 0.431 | 0.454 | 0.432 | 0.455 | 0.416 | 0.435 | 0.432 | 0.455 |
| Weather | 96 | 0.154 | 0.199 | 0.150 | 0.187 | 0.165 | 0.230 | 0.165 | 0.227 | 0.152 | 0.193 | 0.165 | 0.229 |
| | 192 | 0.205 | 0.248 | 0.199 | 0.236 | 0.213 | 0.277 | 0.211 | 0.273 | 0.202 | 0.242 | 0.212 | 0.275 |
| | 336 | 0.262 | 0.290 | 0.256 | 0.282 | 0.274 | 0.323 | 0.270 | 0.318 | 0.259 | 0.286 | 0.272 | 0.321 |
| | 720 | 0.343 | 0.342 | 0.338 | 0.334 | 0.354 | 0.372 | 0.357 | 0.376 | 0.341 | 0.338 | 0.356 | 0.374 |
| Avg | | 0.342 | 0.363 | 0.335 | 0.354 | 0.352 | 0.382 | 0.340 | 0.372 | 0.338 | 0.359 | 0.346 | 0.377 |

MAE are almost always superior to those obtained with MSE. And the difference is not subtle; it is a significant and unmistakable discrepancy. **Therefore, comparing baselines trained with MAE to those trained with MSE would certainly lead to an unfair comparison!**

## D VISUALIZATION

### D.1 VISUALIZATION OF LINO'S FORECASTING RESULTS

To provide insights that help readers better understand the working mechanism of LiNo and to intuitively grasp the effects of advanced 'Recursive Residual Decomposition' (RRD), we present visualizations of LiNo's forecasting results (***Last channel/variate***) across 13 multivariate forecasting benchmarks. We set the input length $T = 96$, prediction length $F = 96$, and number of LiNo blocks (layers) $N = 3$. 'LP' denotes Linear prediction, and 'NP' stands for Nonlinear prediction. LP $i$ or NP $i$ ($i \in \{1, 2, 3\}$) is the linear or nonlinear prediction of $i$-th layer (level). Results are in Figure 6– 18.

### D.2 VISUALIZATION OF WEIGHT OF LI BLOCKS AND NO BLOCKS

We present visualizations of the weight of Li blocks and No blocks obtained across 13 multivariate forecasting benchmarks to help better understand the proposed LiNo. We set the input length $T = 96$, prediction length $F = 96$, and number of LiNo blocks (layers) $N = 3$. Results are in Figure 19– 31.

The method used for getting the weight follows the approach outlined in **Analysis of linear model** (Toner & Darlow, 2024). Plotting the learned matrices as in Figure 19 requires us to first convert each trained model into the form $f(\vec{x}) = A\vec{x} + \vec{b}$. To do this we note that $f(\vec{0}) = A\vec{0} + \vec{b} = \vec{b}$. Thus, the bias can be found by passing the zero vector into the trained model. We can determine $A$ in a similar manner. Let $\vec{e_i}$ denote the $i$-th coordinate vector, that is $\vec{e_i}$ is the vector which is 1 at position $i$ and zero elsewhere. Then $f(\vec{e_i}) = A\vec{e_i} + \vec{b} = A_{.,i} + \vec{b}$ where $A_{.,i}$ is the $i$-th column of $A$. Hence, given that we have already computed the bias term, we may derive $A$ simply by passing through each coordinate vector $\vec{e_i}$ and subtracting $\vec{b}$. Then, we repeat this process separately to each Li block and No block to get their weight, since they each output a forecasting result.

Take the ETTh1 dataset as an example, we observe in Figure 7 that each block (Li or No block) produces significantly different weights. This indicates that each block focuses on different patterns. The three No blocks all generate weights with noticeable periodicity, while the three Li blocks' weights are more concentrated on the most recent points in the input series. These interesting findings help us better understand how neural networks behave when extracting features from time series.

## E FULL RESULTS

### E.1 FULL RESULTS OF MULTIVARIATE FORECASTING BENCHMARK

The full multivariate forecasting results are provided in Table 15 and Table 16 due to the space limitation of the main text. The proposed model achieves comprehensive state-of-the-art in real-world multivariate time series forecasting applications.

### E.2 FULL RESULTS OF UNIVARIATE FORECASTING BENCHMARK

Table 17 provides the full univariate forecasting results to save space in the main text. LiNo surpasses previous state-of-the-art MICN (Wang et al., 2023) by a large, earning its prominent place in univariate time series forecasting tasks.

## F COMPARISON BETWEEN LINO AND N-BEATS

### F.1 THEORY ANALYSIS OF N-BEATS AND LINO

To better assist the reader in understanding the similarities and differences between our work and N-BEATS (Oreshkin et al., 2020), and to help clarify our contributions, we have provided the following analysis.

Here are some essential definitions:

1. The linear feature extractor is denoted as $Li\_Block$, the extracted pattern $L_i$, and linear prediction $P_i^{li}$.

2. The nonlinear feature extractor is denoted as $No\_Block$, the extracted pattern $N_i$, and nonlinear prediction $P_i^{no}$.

3. The projection to get the prediction is denoted as $FC$

4. We assume the input to the current layer in deep neural network as $H_i$ and the next layer $H_{i+1}$, where $H_0 = X\_embed = XW + b$, $X$ is the input raw time series

5. Prediction of current layer $P_i$

6. Final prediction $\hat{Y}$

Then, N-BEATS can be summarized as:

$$N_i = No\_Block(H_i),$$
$$P_i = FC(N_i),$$
$$H_{i+1} = H_i - N_i,$$
$$\hat{Y} = \sum_{i=1}^{N} P^i. \tag{7}$$

Our LiNo can be formulated as:

$$L_i = Li\_Block(H_i),$$
$$P_i^{li} = FC(L_i),$$
$$R_i^L = H_i - L_i,$$
$$N_i = No\_Block(R_i^L),$$
$$P_i^{no} = FC(N_i),$$
$$R_i^N = R_i^L - N_i,$$
$$H_{i+1} = R_i^N,$$
$$P^i = P_i^{li} + P_i^{no},$$
$$\hat{Y} = \sum_{i=1}^{N} P^i. \tag{8}$$

Below are some differences:

1. LiNo introduces a two-stage extraction process: linear ($Li\_Block$) and nonlinear ($No\_Block$), while N-BEATS only uses a one-stage extraction and without separation of linear and nonlinear extraction.

2. LiNo computes predictions from both linear and nonlinear components, while N-BEATS only uses the nonlinear output.

3. LiNo refines intermediate representations after each extraction stage, whereas N-BEATS directly modifies hidden states.

4. LiNo combines both linear and nonlinear predictions, while N-BEATS only obtains nonlinear predictions.

## F.2 EXPERIMENT RESULTS ANALYSIS OF DIFFERENT MODEL DESIGN

We designed the following experiments to validate *the superiority of LiNo over N-BEATS-style design* under a **Univariate Scenario**, to exclude any interference from channel-independent or channel-dependent information.

The learnable autoregressive model (AR) mentioned in the paper is employed as the linear feature extractor ($Li\_Block$).

The Temporal + Frequency projection, combined with the Tanh activation function, is adopted as the nonlinear feature extractor ($No\_Block$).

To ensure a fair comparison with the **N-BEATS-style design** (since completely removing the $Li\_Block$ may result in potential loss of parameters), the $No\_Block$ used in N-BEATS is actually a combination of $Li\_Block$ and $No\_Block$, which is still nonlinear:

$$
\begin{aligned}
L_i &= Li\_Block(H_i), \\
N_i &= No\_Block(L_i), \\
P_i &= FC(N_i), \\
H_{i+1} &= H_i - N_i, \\
\hat{Y} &= \sum_{i=1}^{N} P^i.
\end{aligned}
\tag{9}
$$

We also include two more designs to investigate the specific effectiveness of LiNo.

**RAW.** The input features pass through the Li Block and No Block sequentially, but no feature decomposition is performed. The model's final output features are directly used for prediction (Traditional Design).

$$
\begin{aligned}
L_i &= Li\_Block(H_i), \\
N_i &= No\_Block(L_i), \\
H_{i+1} &= N_i, \\
\hat{Y} &= FC(H_N).
\end{aligned}
\tag{10}
$$

**LN.** The input features pass through the Li Block and No Block sequentially, with each block generating its own prediction based on the features extracted, but no residual decomposition is applied (Common Linear-Nonlinear Decomposition Design).

$$
\begin{aligned}
L_i &= Li\_Block(H_i), \\
P_i^{li} &= FC(L_i), \\
N_i &= No\_Block(L_i), \\
P_i^{no} &= FC(N_i), \\
H_{i+1} &= N_i, \\
\hat{Y} &= \sum_{i=1}^{N} (P_i^{li} + P_i^{no}).
\end{aligned}
\tag{11}
$$

Experimental results in Table 12 demonstrate the superiority of LiNo over the N-BEATS-style design. Although both N-BEATS and LN show slight improvements over RAW, ***our LiNo design significantly outperforms the other designs***. This demonstrates:

1. The effectiveness of explicit linear and nonlinear modeling.
2. The effectiveness of RRD for representation decomposition.
3. The potential for further improvement by combining both approaches (LiNo).

### F.3 COMPARISON BETWEEN LINO AND ORIGINAL N-BEATS

We also include the comparison of LiNo and the original version of N-BEATS. As in Table 13, LiNo significantly outperforms N-BEATS (Oreshkin et al., 2020) and its successor, N-HiTS (Challu et al., 2023), across the ETTm2, ECL, and Weather datasets. This demonstrates that, compared to using Recursive Residual Decomposition (RRD) to refine the final prediction based on residual errors from previous predictions (as proposed in N-BEATS), employing RRD to capture more nuanced linear and nonlinear patterns (as in this work) is a better choice.

## G COMPARISON BETWEEN PROPOSED NO BLOCK AND SOFTS

We noticed that there is a recent work that is so close to our channel mixing technique, which is called **SOFTS** (NeurIPS 2024). Our channel mixing technique and **SOFTS** both stem from the idea

of learning channel dependence while maintaining channel independence. Despite that, here are some major differences between LiNo and SOFTS:

- LiNo uses the weighted sum of all channels with weight generated by the softmax function, while SOFTS uses a stochastic pooling technique to obtain the global token, which is more time-consuming.
- We directly perform softmax to input feature, while SOFTS first go through a Feedforward Network, which is redundant.

Experiment results in Table 14 successfully demonstrate the superiority of the proposed No block over SOFTS.

Table 12: Ablations on overall design under **Univariate Scenario** with input length fixed to $T = 96$. 'RAW' refers to the raw model. 'LN' stands for separately modeling linear and nonlinear patterns, but no decomposition is applied. 'N-BEATS' refers to the N-BEATS-style model where RRD is employed but without separating the extraction and prediction of linear and nonlinear. 'LiNo' is the proposed framework, where both RRD and separation of linear and nonlinear patterns. The **bold** values indicate the best performance.

| Models | | RAW | | LN | | N-BEATS | | LiNo | |
|---|---|---|---|---|---|---|---|---|---|
| Metric | | MSE | MAE | MSE | MAE | MSE | MAE | MSE | MAE |
| ETTh2 | 96 | 0.133 | 0.282 | 0.128 | 0.274 | 0.126 | 0.271 | **0.126** | **0.271** |
| | 192 | 0.183 | 0.335 | 0.177 | 0.327 | 0.179 | 0.328 | **0.174** | **0.324** |
| | 336 | 0.216 | 0.372 | 0.213 | 0.367 | 0.213 | 0.368 | **0.209** | **0.364** |
| | 720 | 0.227 | 0.384 | 0.229 | 0.385 | 0.226 | 0.381 | **0.223** | **0.379** |
| ETTm2 | 96 | 0.067 | 0.186 | 0.068 | 0.188 | 0.067 | 0.187 | **0.066** | **0.186** |
| | 192 | 0.102 | 0.239 | 0.102 | 0.237 | 0.102 | 0.238 | **0.099** | **0.234** |
| | 336 | 0.132 | 0.276 | 0.133 | 0.274 | 0.132 | 0.276 | **0.130** | **0.272** |
| | 720 | 0.185 | 0.334 | 0.185 | 0.334 | **0.185** | **0.333** | 0.187 | 0.335 |
| Traffic | 96 | 0.167 | 0.247 | 0.163 | 0.255 | 0.168 | 0.253 | **0.159** | **0.248** |
| | 192 | 0.164 | 0.245 | 0.158 | 0.244 | 0.159 | 0.244 | **0.155** | **0.241** |
| | 336 | 0.163 | 0.247 | 0.158 | 0.245 | 0.156 | 0.242 | **0.153** | **0.237** |
| | 720 | 0.183 | 0.263 | 0.176 | 0.261 | 0.177 | 0.262 | **0.174** | **0.260** |
| Avg | | | 0.160 | 0.284 | 0.158 | 0.283 | 0.158 | 0.282 | **0.155** | **0.279** |

Table 13: Multivariate forecasting results with prediction lengths $F \in \{96, 192, 336, 720\}$ and fixed lookback winodw $T = 96$ for all datasets. The Results of N-BEATS and N-HiTS are taken from (Challu et al., 2023). The **bold** values indicate the best performance.

| Models | | LiNo | | N-HiTS | | N-BEATS | |
|---|---|---|---|---|---|---|---|
| Metric | | MSE | MAE | MSE | MAE | MSE | MAE |
| ETTm2 | 96 | **0.171** | **0.254** | 0.176 | 0.255 | 0.184 | 0.263 |
| | 192 | **0.237** | **0.298** | 0.245 | 0.305 | 0.273 | 0.337 |
| | 336 | **0.296** | **0.336** | 0.295 | 0.346 | 0.309 | 0.355 |
| | 720 | **0.395** | **0.393** | 0.401 | 0.413 | 0.411 | 0.425 |
| | Promote | -6.63% | -7.17% | -5.10% | -4.42% | 0 | 0 |
| ECL | 96 | **0.138** | **0.233** | 0.147 | 0.249 | 0.145 | 0.247 |
| | 192 | **0.155** | **0.250** | 0.167 | 0.269 | 0.180 | 0.283 |
| | 336 | **0.171** | **0.267** | 0.186 | 0.290 | 0.200 | 0.308 |
| | 720 | **0.191** | **0.290** | 0.243 | 0.340 | 0.266 | 0.362 |
| | Promote | -17.19% | -13.33% | -6.07% | -4.33% | 0 | 0 |
| Weather | 96 | **0.154** | **0.199** | 0.158 | 0.195 | 0.167 | 0.203 |
| | 192 | **0.205** | **0.248** | 0.211 | 0.247 | 0.229 | 0.261 |
| | 336 | **0.262** | **0.290** | 0.274 | 0.300 | 0.287 | 0.304 |
| | 720 | **0.343** | **0.342** | 0.351 | 0.353 | 0.368 | 0.359 |
| | Promote | -8.28% | -4.26% | -5.42% | -2.84% | 0 | 0 |

Table 14: Multivariate forecasting performance comparison between SOFTS and the proposed No Block with input length fixed to $T = 96$. The **bold** values indicate the best performance. The result of SOFTS is directly referenced from the original paper (Han et al., 2024).

| Models | | ETTh1 | | ETTh2 | | ETTm1 | | ETTm2 | | Weather | |
|---|---|---|---|---|---|---|---|---|---|---|---|
| Metric | | MSE | MAE | MSE | MAE | MSE | MAE | MSE | MAE | MSE | MAE |
| SOFTS | 96 | 0.381 | 0.399 | 0.297 | 0.347 | **0.325** | **0.361** | 0.180 | 0.261 | **0.166** | **0.208** |
| | 192 | 0.435 | 0.431 | 0.373 | 0.394 | 0.375 | 0.389 | 0.246 | 0.306 | 0.217 | 0.253 |
| | 336 | 0.480 | 0.452 | **0.410** | 0.426 | **0.405** | 0.412 | 0.319 | 0.352 | 0.282 | 0.300 |
| | 720 | 0.499 | 0.488 | **0.411** | **0.433** | 0.466 | 0.447 | 0.405 | 0.401 | 0.356 | 0.351 |
| No Block | 96 | **0.377** | **0.394** | **0.293** | **0.342** | 0.333 | 0.365 | **0.175** | **0.257** | 0.172 | 0.211 |
| | 192 | **0.423** | **0.420** | **0.372** | **0.390** | 0.369 | 0.385 | **0.241** | **0.300** | **0.214** | **0.249** |
| | 336 | **0.457** | **0.440** | 0.414 | **0.425** | 0.407 | **0.407** | 0.302 | 0.340 | **0.276** | **0.296** |
| | 720 | **0.454** | **0.457** | 0.417 | 0.438 | 0.470 | **0.440** | 0.399 | 0.396 | **0.353** | **0.347** |

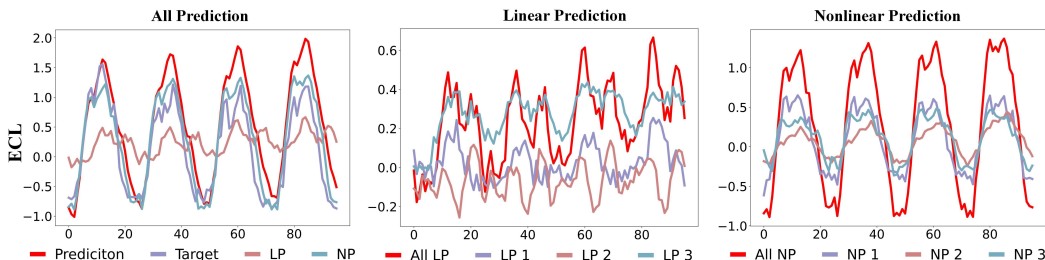

Figure 6: Visualization of LiNo's multivariate forecasting result on ECL dataset. 'LP' denotes Linear prediction, and 'NP' stands for Nonlinear prediction. LP $i$ or NP $i$ ($i \in \{1, 2, 3\}$) is the linear or nonlinear prediction of $i$-th layer (level). Same to followed figures.

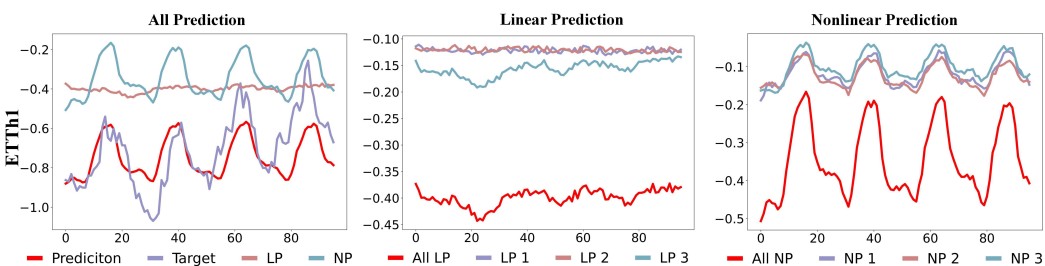

Figure 7: Visualization of LiNo's multivariate forecasting result on ETTh1 dataset.

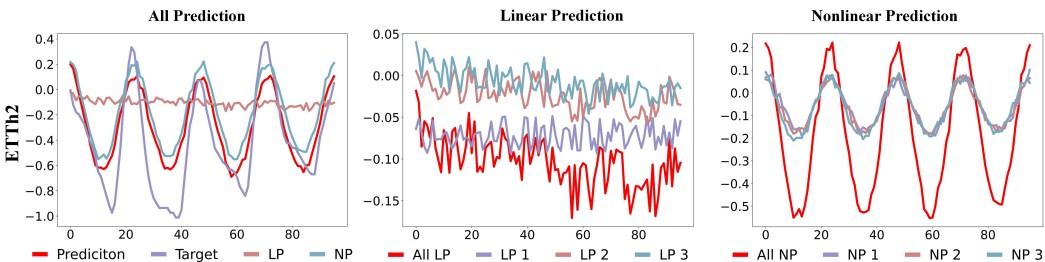

Figure 8: Visualization of LiNo's multivariate forecasting result on ETTh2 dataset.

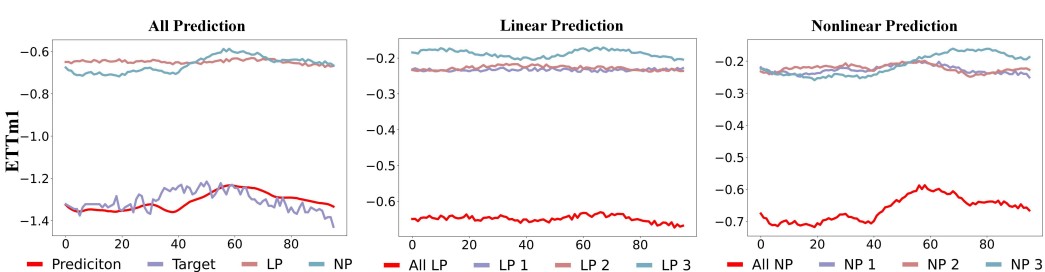

Figure 9: Visualization of LiNo's multivariate forecasting result on ETTm1 dataset.

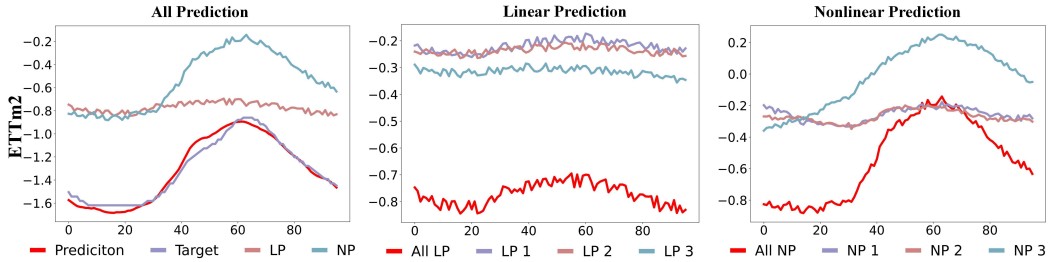

Figure 10: Visualization of LiNo's multivariate forecasting result on ETTm2 dataset.

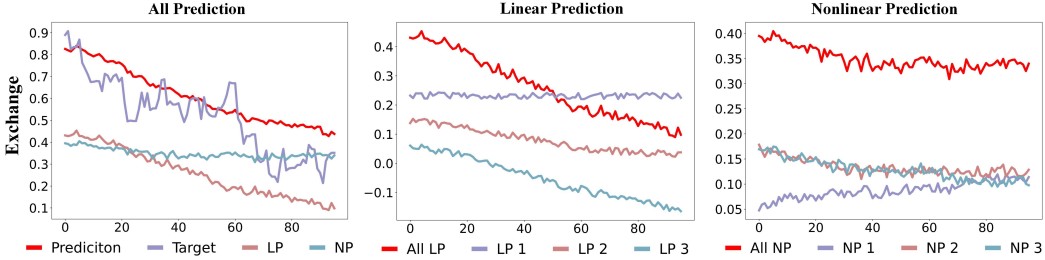

Figure 11: Visualization of LiNo's multivariate forecasting result on Exchange dataset.

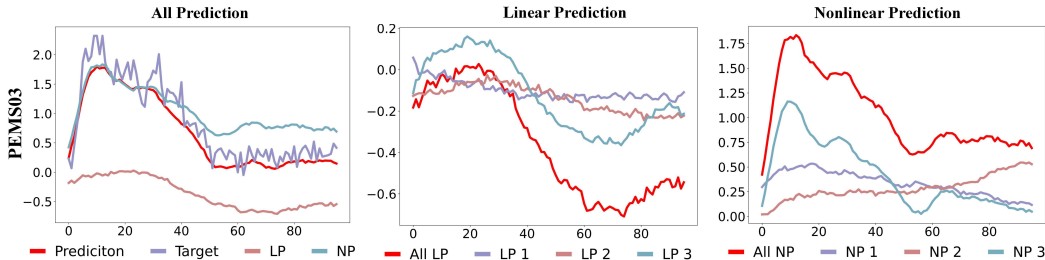

Figure 12: Visualization of LiNo's multivariate forecasting result on PEMS03 dataset.

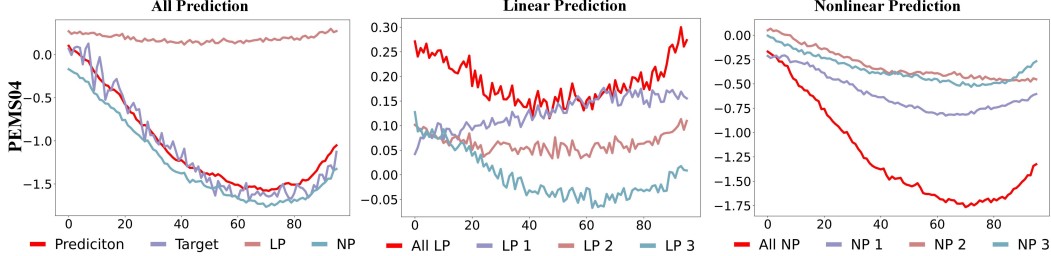

Figure 13: Visualization of LiNo's multivariate forecasting result on PEMS04 dataset.

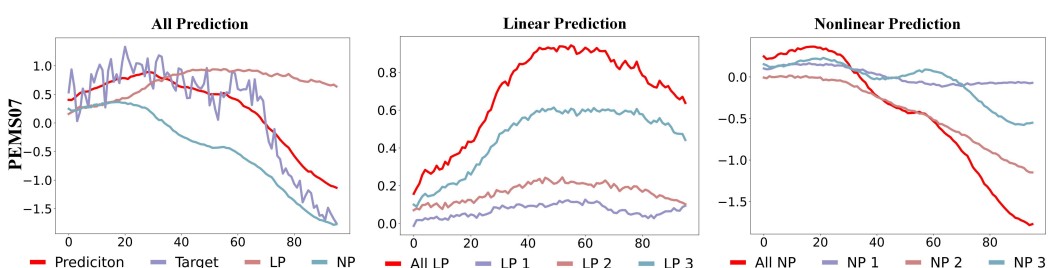

Figure 14: Visualization of LiNo's multivariate forecasting result on PEMS07 dataset.

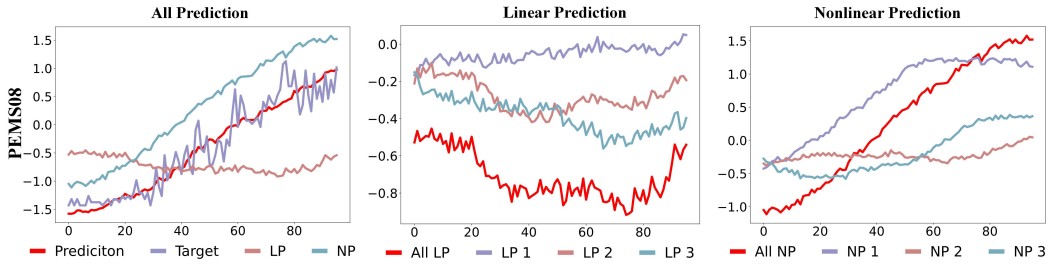

Figure 15: Visualization of LiNo's multivariate forecasting result on PEMS08 dataset.

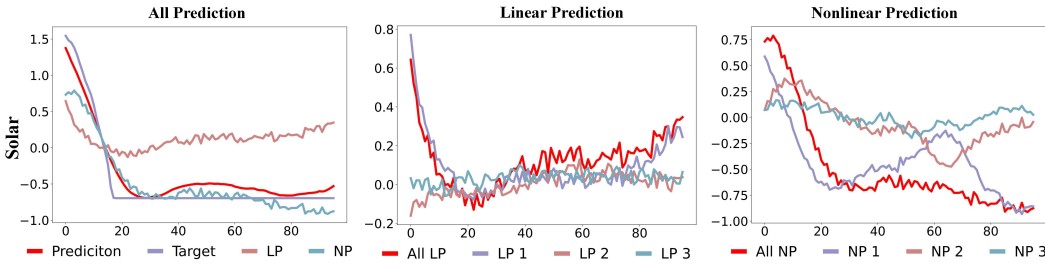

Figure 16: Visualization of LiNo's multivariate forecasting result on Solar dataset.

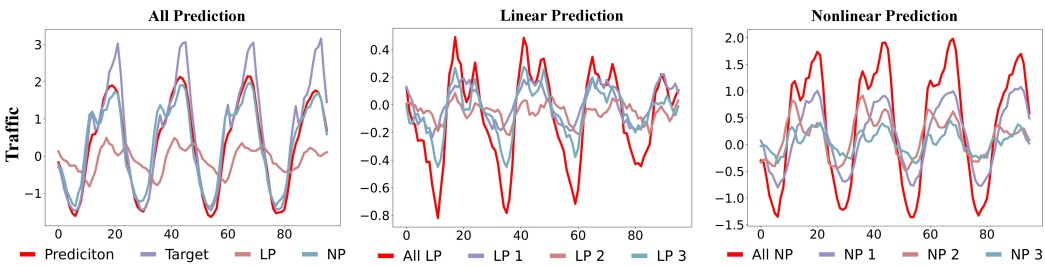

Figure 17: Visualization of LiNo's multivariate forecasting result on Traffic dataset.

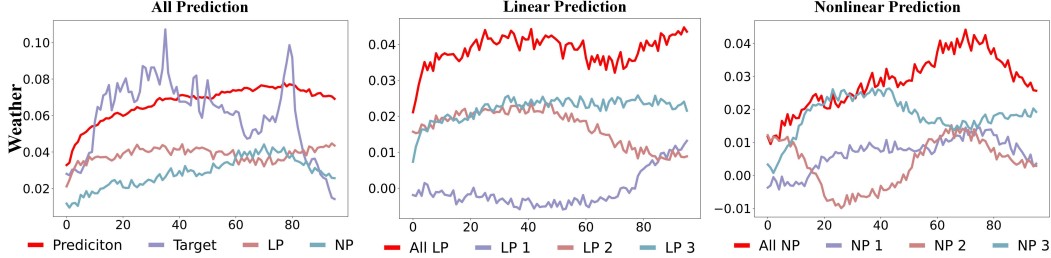

Figure 18: Visualization of LiNo's multivariate forecasting result on Weather dataset.

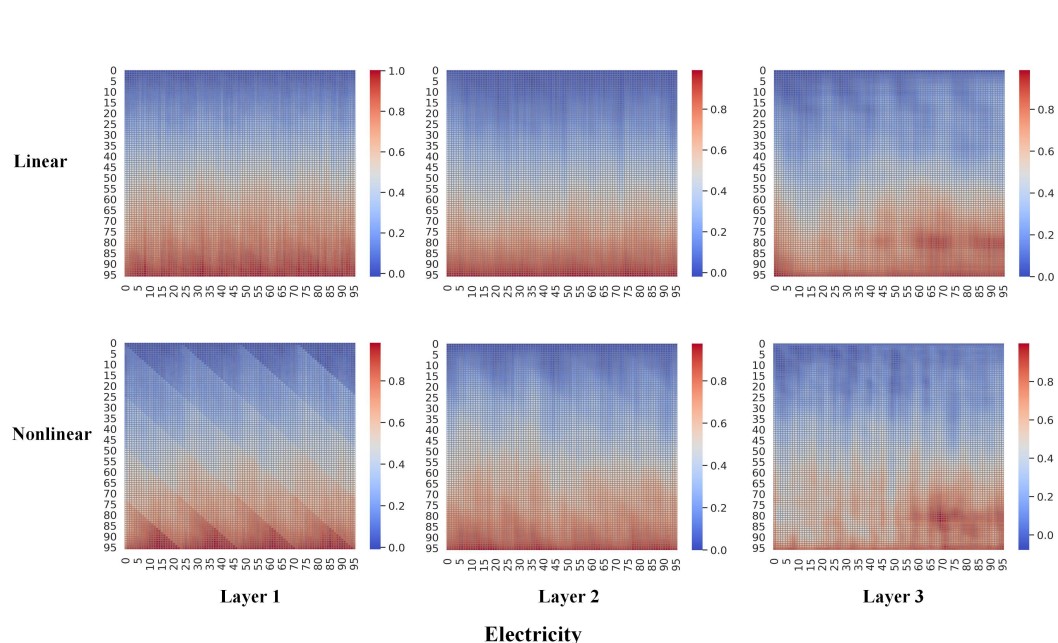

Figure 19: Visualization of LiNo's weight on ECL dataset.

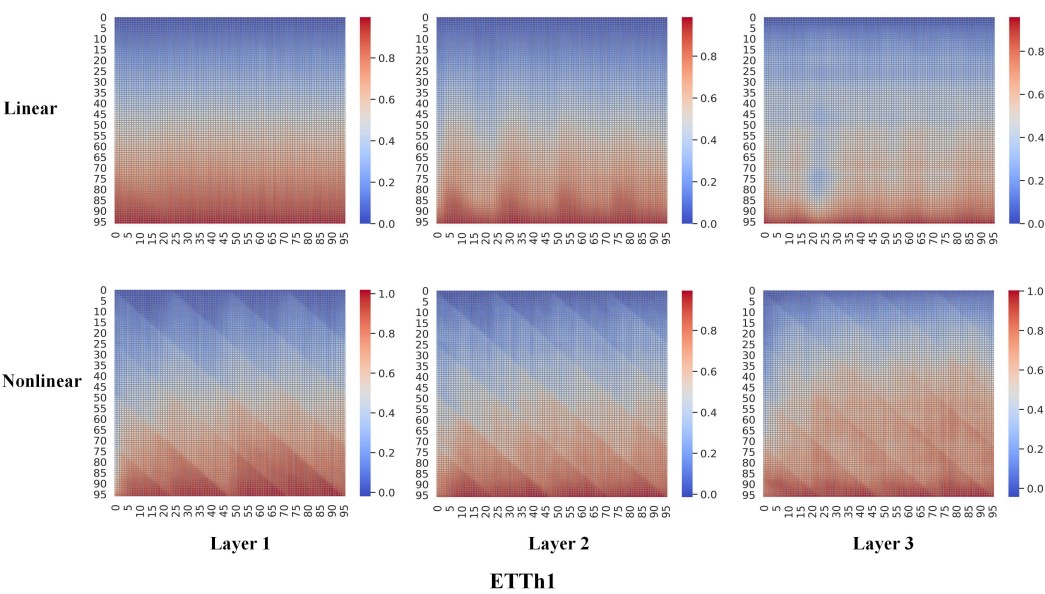

Figure 20: Visualization of LiNo's weight on ETTh1 dataset.

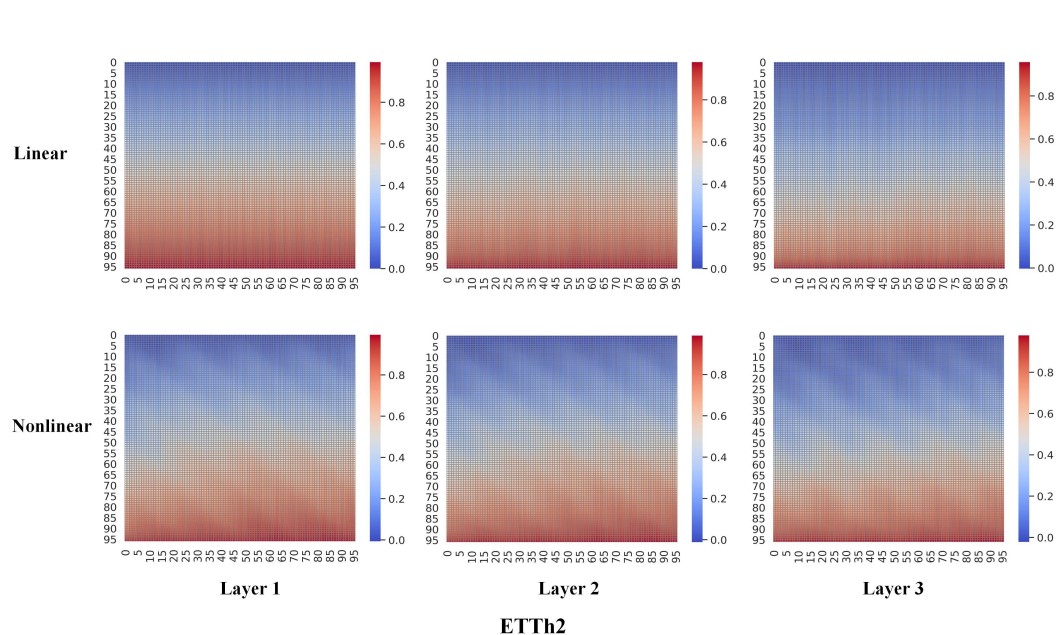

Figure 21: Visualization of LiNo's weight on ETTh2 dataset.

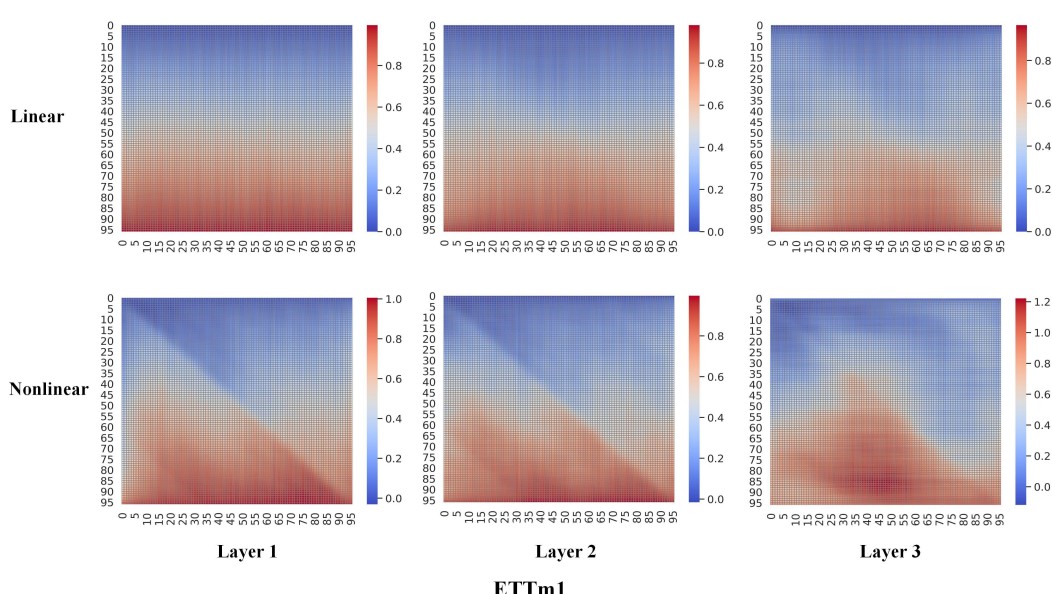

Figure 22: Visualization of LiNo's weight on ETTm1 dataset.

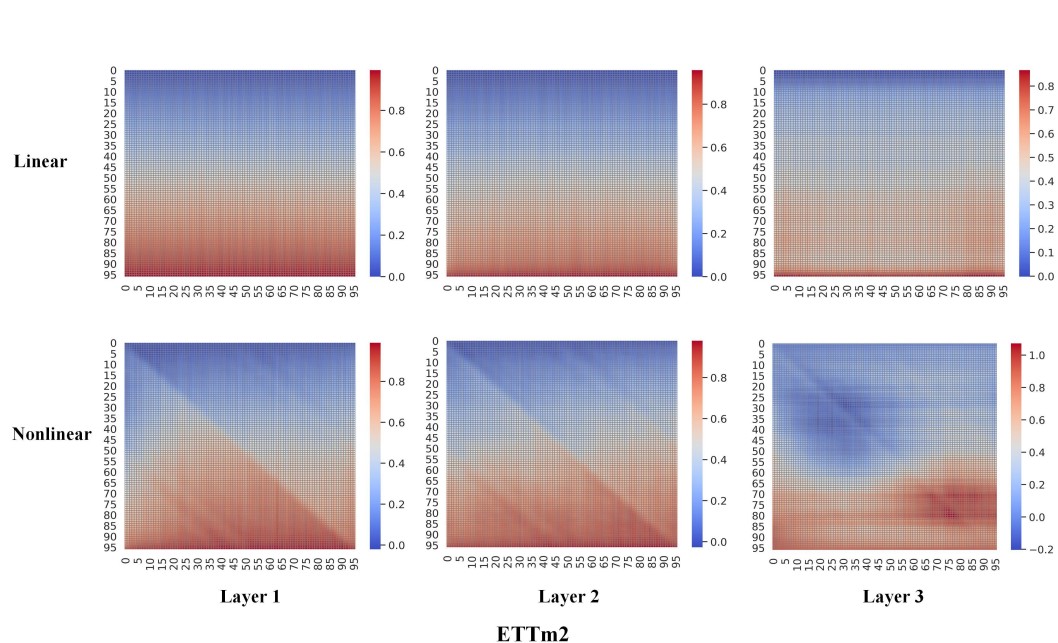

Figure 23: Visualization of LiNo's weight on ETTm2 dataset.

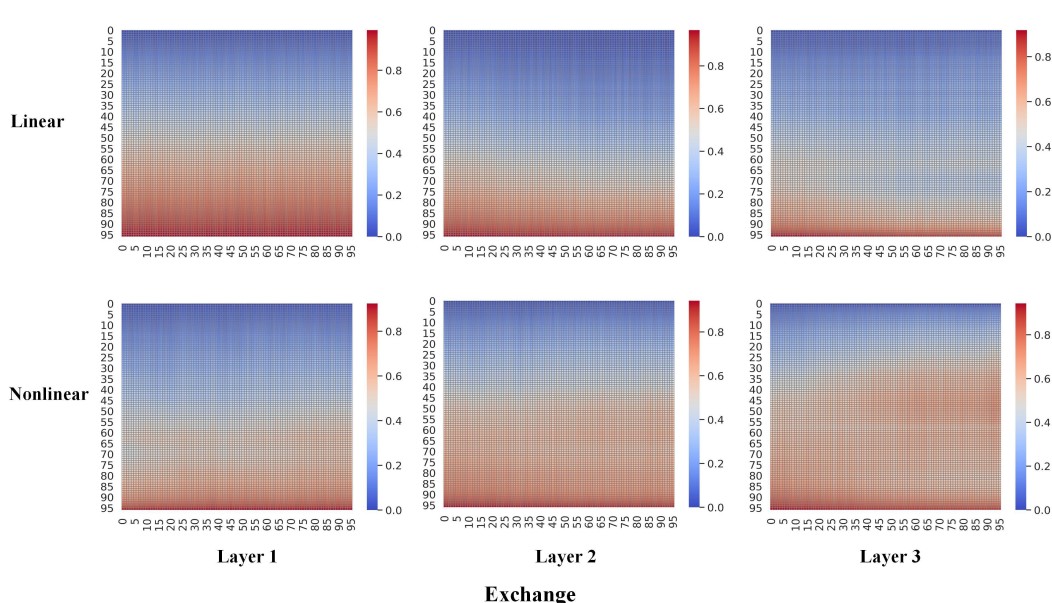

Figure 24: Visualization of LiNo's weight on Exchange dataset.

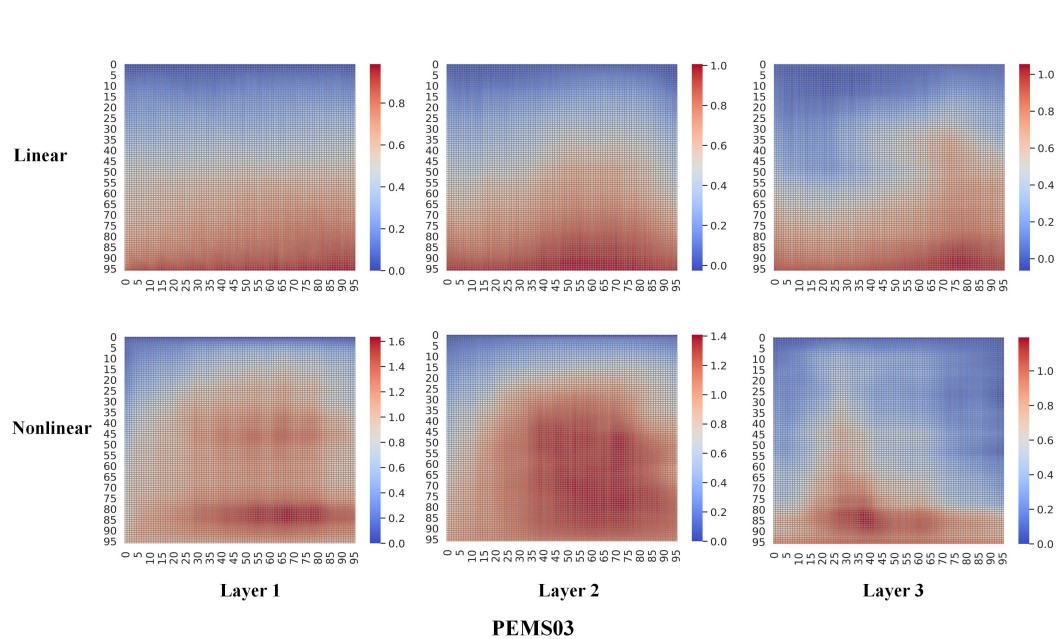

Figure 25: Visualization of LiNo's weight on PEMS03 dataset.

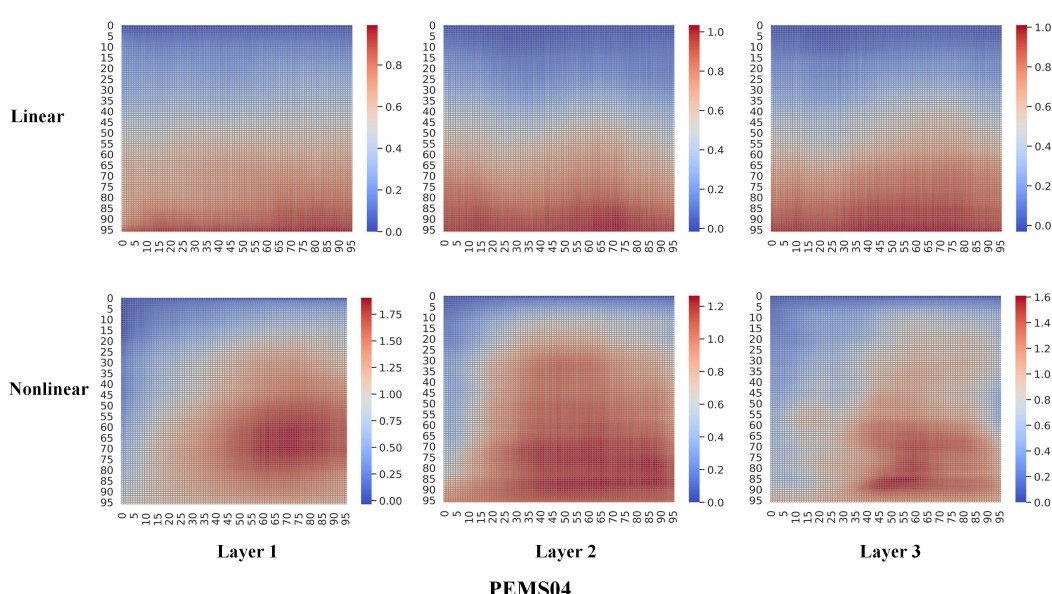

Figure 26: Visualization of LiNo's weight on PEMS04 dataset.

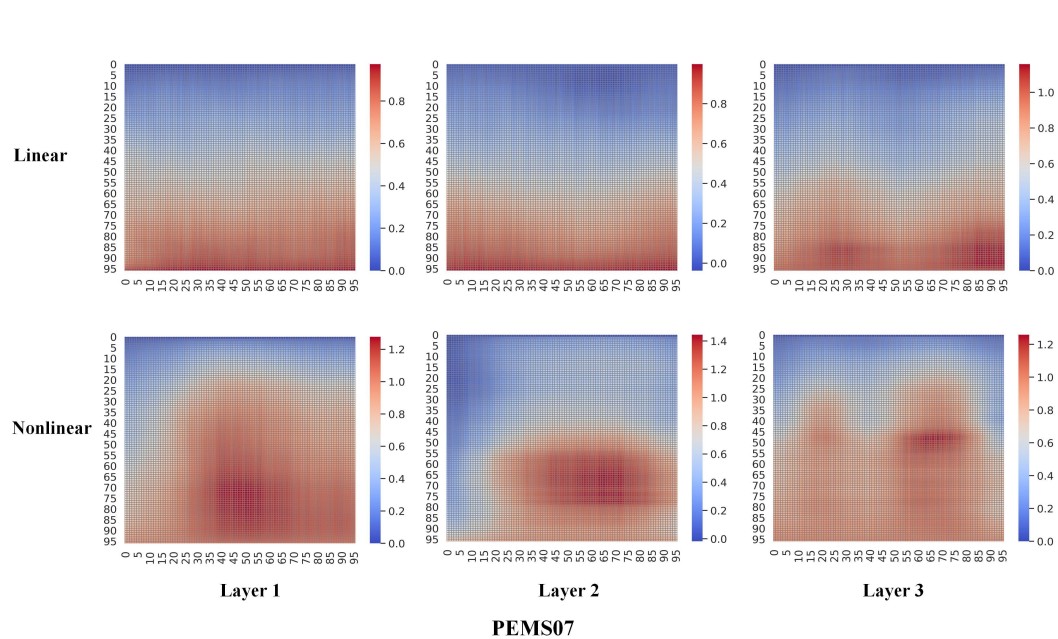

Figure 27: Visualization of LiNo's weight on PEMS07 dataset.

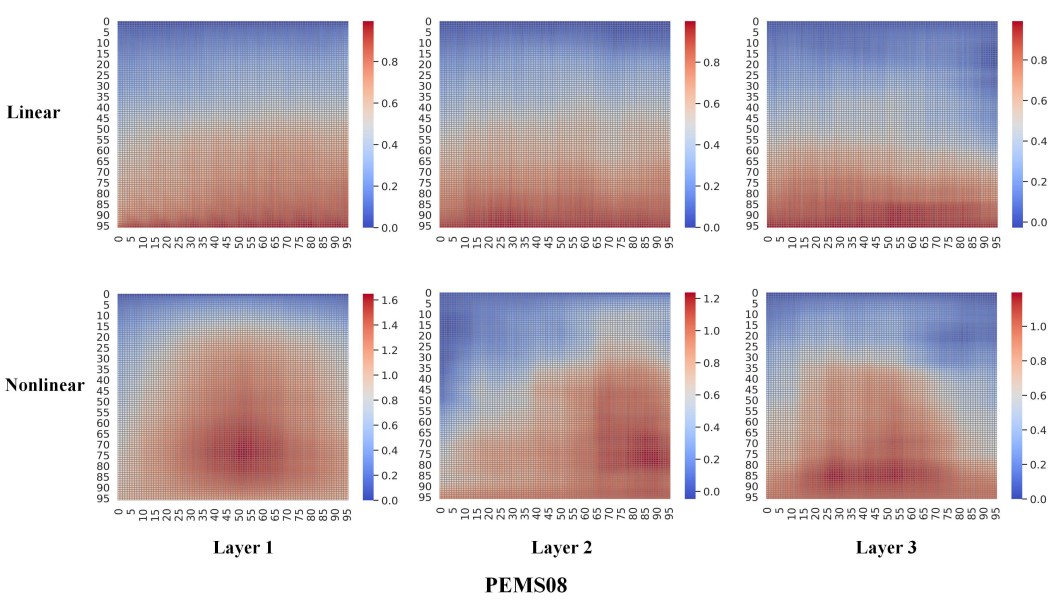

Figure 28: Visualization of LiNo's weight on PEMS08 dataset.

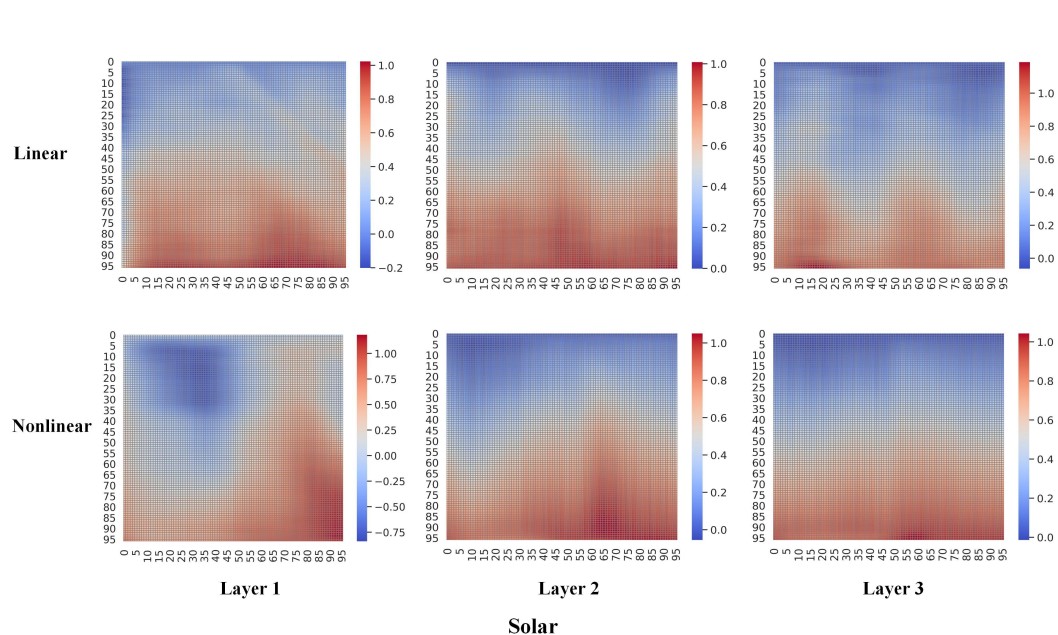

Figure 29: Visualization of LiNo's weight on Solar dataset.

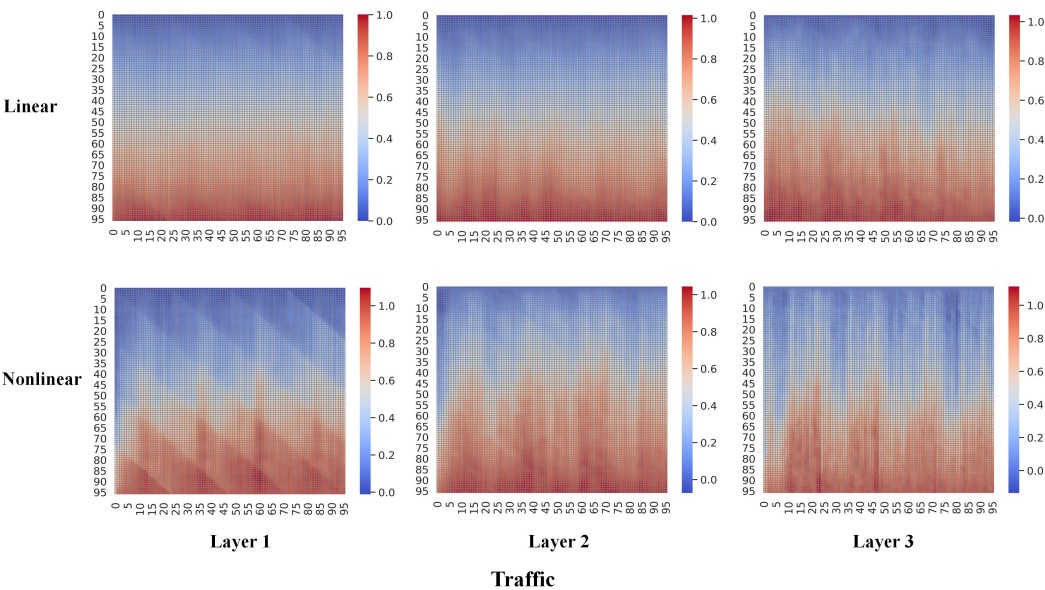

Figure 30: Visualization of LiNo's weight on Traffic dataset.

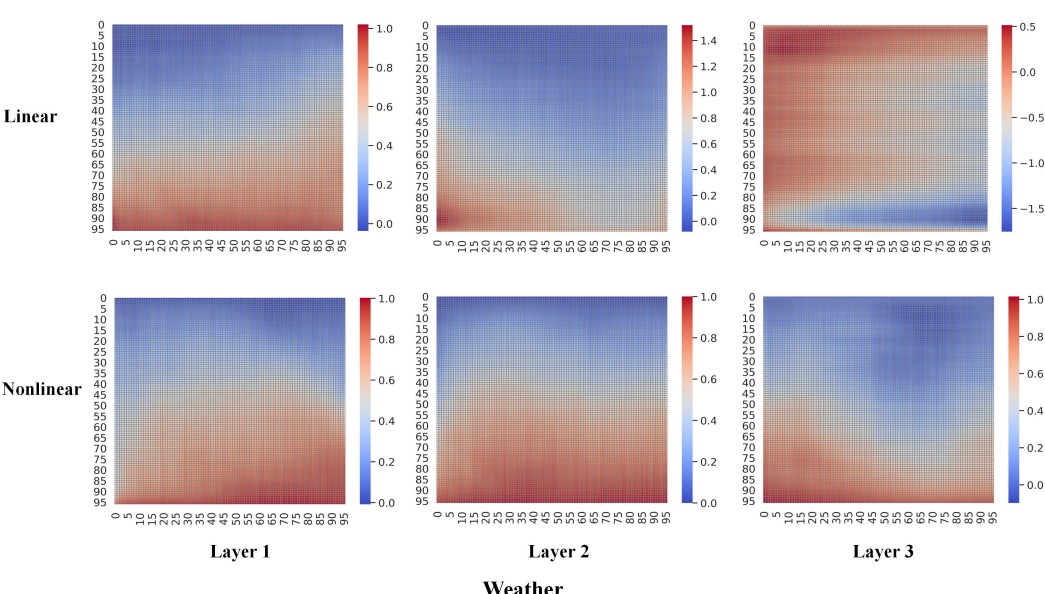

Figure 31: Visualization of LiNo's weight on Weather dataset.

Table 15: Full results of the long-term forecasting task. The input sequence length is set to $T = 96$ for all baselines. *Avg* means the average results from all four prediction lengths.

| Models | | LiNo (Ours) | | iTransformer (2024b) | | Rlinear (2023a) | | TSMixer (2023) | | PatchTST (2023) | | Crossformer (2023) | | TiDE (2023) | | TimesNet (2023) | | DLinear (2023) | | FEDformer (2022b) | | Autoformer (2021) | |
|---|---|---|---|---|---|---|---|---|---|---|---|---|---|---|---|---|---|---|---|---|---|---|---|
| Metric | | MSE | MAE | MSE | MAE | MSE | MAE | MSE | MAE | MSE | MAE | MSE | MAE | MSE | MAE | MSE | MAE | MSE | MAE | MSE | MAE | MSE | MAE |
| ETTm1 | 96 | **0.322** | **0.361** | 0.334 | 0.368 | 0.355 | 0.376 | 0.323 | 0.363 | 0.329 | 0.367 | 0.404 | 0.426 | 0.364 | 0.387 | 0.338 | 0.375 | 0.345 | 0.372 | 0.379 | 0.419 | 0.505 | 0.475 |
| | 192 | **0.365** | **0.383** | 0.377 | 0.391 | 0.391 | 0.392 | 0.376 | 0.392 | 0.367 | 0.385 | 0.450 | 0.451 | 0.398 | 0.404 | 0.374 | 0.387 | 0.380 | 0.389 | 0.426 | 0.441 | 0.553 | 0.496 |
| | 336 | 0.401 | **0.408** | 0.426 | 0.420 | 0.424 | 0.415 | 0.407 | 0.413 | **0.399** | 0.410 | 0.532 | 0.515 | 0.428 | 0.425 | 0.410 | 0.411 | 0.413 | 0.413 | 0.445 | 0.459 | 0.621 | 0.537 |
| | 720 | 0.469 | 0.447 | 0.491 | 0.459 | 0.487 | 0.450 | 0.485 | 0.459 | **0.454** | **0.439** | 0.666 | 0.589 | 0.487 | 0.461 | 0.478 | 0.450 | 0.474 | 0.453 | 0.543 | 0.490 | 0.671 | 0.561 |
| | Avg | 0.389 | **0.400** | 0.407 | 0.410 | 0.414 | 0.407 | 0.398 | 0.407 | **0.387** | 0.400 | 0.513 | 0.496 | 0.419 | 0.419 | 0.400 | 0.406 | 0.403 | 0.407 | 0.448 | 0.452 | 0.588 | 0.517 |
| ETTm2 | 96 | **0.171** | **0.254** | 0.180 | 0.264 | 0.182 | 0.265 | 0.182 | 0.266 | 0.175 | 0.259 | 0.287 | 0.366 | 0.207 | 0.305 | 0.187 | 0.267 | 0.193 | 0.292 | 0.203 | 0.287 | 0.255 | 0.339 |
| | 192 | **0.237** | **0.298** | 0.250 | 0.309 | 0.246 | 0.304 | 0.249 | 0.309 | 0.241 | 0.302 | 0.414 | 0.492 | 0.290 | 0.364 | 0.249 | 0.309 | 0.284 | 0.362 | 0.269 | 0.328 | 0.281 | 0.340 |
| | 336 | **0.296** | **0.336** | 0.311 | 0.348 | 0.307 | 0.342 | 0.309 | 0.347 | 0.305 | 0.343 | 0.597 | 0.542 | 0.377 | 0.422 | 0.321 | 0.351 | 0.369 | 0.427 | 0.325 | 0.366 | 0.339 | 0.372 |
| | 720 | **0.395** | **0.393** | 0.412 | 0.407 | 0.407 | 0.398 | 0.416 | 0.408 | 0.402 | 0.400 | 1.730 | 1.042 | 0.588 | 0.524 | 0.408 | 0.403 | 0.554 | 0.522 | 0.421 | 0.415 | 0.433 | 0.432 |
| | Avg | **0.275** | **0.320** | 0.288 | 0.332 | 0.286 | 0.327 | 0.289 | 0.333 | 0.281 | 0.326 | 0.757 | 0.610 | 0.358 | 0.404 | 0.291 | 0.333 | 0.350 | 0.401 | 0.305 | 0.349 | 0.327 | 0.371 |
| ETTh1 | 96 | 0.378 | **0.395** | 0.386 | 0.405 | 0.386 | 0.395 | 0.401 | 0.412 | 0.414 | 0.419 | 0.423 | 0.448 | 0.479 | 0.464 | 0.384 | 0.402 | 0.386 | 0.400 | **0.376** | 0.419 | 0.449 | 0.459 |
| | 192 | 0.423 | **0.423** | 0.441 | 0.436 | 0.437 | 0.424 | 0.452 | 0.442 | 0.460 | 0.445 | 0.471 | 0.474 | 0.525 | 0.492 | 0.436 | 0.429 | 0.437 | 0.432 | **0.420** | 0.448 | 0.500 | 0.482 |
| | 336 | **0.455** | **0.438** | 0.487 | 0.458 | 0.479 | 0.446 | 0.492 | 0.463 | 0.501 | 0.466 | 0.570 | 0.546 | 0.565 | 0.515 | 0.491 | 0.469 | 0.481 | 0.459 | 0.459 | 0.465 | 0.521 | 0.496 |
| | 720 | **0.459** | **0.456** | 0.503 | 0.491 | 0.481 | 0.470 | 0.507 | 0.490 | 0.500 | 0.488 | 0.653 | 0.621 | 0.594 | 0.558 | 0.521 | 0.500 | 0.519 | 0.516 | 0.506 | 0.507 | 0.514 | 0.512 |
| | Avg | **0.429** | **0.428** | 0.454 | 0.447 | 0.446 | 0.434 | 0.463 | 0.452 | 0.469 | 0.454 | 0.529 | 0.522 | 0.541 | 0.507 | 0.458 | 0.450 | 0.456 | 0.452 | 0.440 | 0.460 | 0.496 | 0.487 |
| ETTh2 | 96 | 0.292 | 0.340 | 0.297 | 0.349 | **0.288** | **0.338** | 0.319 | 0.361 | 0.302 | 0.348 | 0.745 | 0.584 | 0.400 | 0.440 | 0.340 | 0.374 | 0.333 | 0.387 | 0.358 | 0.397 | 0.346 | 0.388 |
| | 192 | 0.375 | 0.391 | 0.380 | 0.400 | **0.374** | **0.390** | 0.402 | 0.410 | 0.388 | 0.400 | 0.877 | 0.656 | 0.528 | 0.509 | 0.402 | 0.414 | 0.477 | 0.476 | 0.429 | 0.439 | 0.456 | 0.452 |
| | 336 | 0.418 | 0.426 | 0.428 | 0.432 | **0.415** | 0.426 | 0.444 | 0.446 | 0.426 | 0.433 | 1.043 | 0.731 | 0.643 | 0.571 | 0.452 | 0.452 | 0.594 | 0.541 | 0.496 | 0.487 | 0.482 | 0.486 |
| | 720 | 0.422 | 0.441 | 0.427 | 0.445 | **0.420** | **0.440** | 0.441 | 0.450 | 0.431 | 0.446 | 1.104 | 0.763 | 0.874 | 0.679 | 0.462 | 0.468 | 0.831 | 0.657 | 0.463 | 0.474 | 0.515 | 0.511 |
| | Avg | 0.377 | 0.400 | 0.383 | 0.407 | **0.374** | **0.398** | 0.401 | 0.417 | 0.387 | 0.407 | 0.942 | 0.684 | 0.611 | 0.550 | 0.414 | 0.427 | 0.559 | 0.515 | 0.437 | 0.449 | 0.450 | 0.459 |
| ECL | 96 | **0.138** | **0.233** | 0.148 | 0.240 | 0.201 | 0.281 | 0.157 | 0.260 | 0.181 | 0.270 | 0.219 | 0.314 | 0.237 | 0.329 | 0.168 | 0.272 | 0.197 | 0.282 | 0.193 | 0.308 | 0.201 | 0.317 |
| | 192 | **0.155** | **0.250** | 0.162 | 0.253 | 0.201 | 0.283 | 0.173 | 0.274 | 0.188 | 0.274 | 0.231 | 0.322 | 0.236 | 0.330 | 0.184 | 0.289 | 0.196 | 0.285 | 0.201 | 0.315 | 0.222 | 0.334 |
| | 336 | **0.171** | **0.267** | 0.178 | 0.269 | 0.215 | 0.298 | 0.192 | 0.295 | 0.204 | 0.293 | 0.246 | 0.337 | 0.249 | 0.344 | 0.198 | 0.300 | 0.209 | 0.301 | 0.214 | 0.329 | 0.231 | 0.338 |
| | 720 | **0.191** | **0.290** | 0.225 | 0.317 | 0.257 | 0.331 | 0.223 | 0.318 | 0.246 | 0.324 | 0.280 | 0.363 | 0.284 | 0.373 | 0.220 | 0.320 | 0.245 | 0.333 | 0.246 | 0.355 | 0.254 | 0.361 |
| | Avg | **0.164** | **0.260** | 0.178 | 0.270 | 0.219 | 0.298 | 0.186 | 0.287 | 0.205 | 0.290 | 0.244 | 0.334 | 0.251 | 0.344 | 0.192 | 0.295 | 0.212 | 0.300 | 0.214 | 0.327 | 0.227 | 0.338 |
| Exchange | 96 | **0.084** | **0.203** | 0.086 | 0.206 | 0.093 | 0.217 | 0.089 | 0.211 | 0.088 | 0.205 | 0.256 | 0.367 | 0.094 | 0.218 | 0.107 | 0.234 | 0.088 | 0.218 | 0.148 | 0.278 | 0.197 | 0.323 |
| | 192 | **0.176** | **0.298** | 0.177 | 0.299 | 0.184 | 0.307 | 0.177 | 0.302 | 0.176 | 0.299 | 0.470 | 0.509 | 0.184 | 0.307 | 0.226 | 0.344 | 0.176 | 0.315 | 0.271 | 0.315 | 0.300 | 0.369 |
| | 336 | 0.316 | **0.409** | 0.331 | 0.417 | 0.351 | 0.432 | 0.327 | 0.415 | **0.301** | **0.397** | 1.268 | 0.883 | 0.349 | 0.431 | 0.367 | 0.448 | 0.313 | 0.427 | 0.460 | 0.427 | 0.509 | 0.524 |
| | 720 | **0.823** | **0.682** | 0.847 | 0.691 | 0.886 | 0.714 | 0.912 | 0.727 | 0.901 | 0.714 | 1.767 | 1.068 | 0.852 | 0.698 | 0.964 | 0.746 | 0.839 | 0.695 | 1.195 | 0.695 | 1.447 | 0.941 |
| | Avg | **0.350** | **0.398** | 0.360 | 0.403 | 0.378 | 0.417 | 0.376 | 0.414 | 0.367 | 0.404 | 0.940 | 0.707 | 0.370 | 0.413 | 0.416 | 0.443 | 0.354 | 0.414 | 0.519 | 0.429 | 0.613 | 0.539 |
| Traffic | 96 | 0.429 | 0.276 | **0.395** | **0.268** | 0.649 | 0.389 | 0.493 | 0.336 | 0.462 | 0.295 | 0.522 | 0.290 | 0.805 | 0.493 | 0.593 | 0.321 | 0.650 | 0.396 | 0.587 | 0.366 | 0.613 | 0.388 |
| | 192 | 0.450 | 0.289 | **0.417** | **0.276** | 0.601 | 0.366 | 0.497 | 0.351 | 0.466 | 0.296 | 0.530 | 0.293 | 0.756 | 0.474 | 0.617 | 0.336 | 0.598 | 0.370 | 0.604 | 0.373 | 0.616 | 0.382 |
| | 336 | 0.468 | 0.297 | **0.433** | **0.283** | 0.609 | 0.369 | 0.528 | 0.361 | 0.482 | 0.304 | 0.558 | 0.305 | 0.762 | 0.477 | 0.629 | 0.336 | 0.605 | 0.373 | 0.621 | 0.383 | 0.622 | 0.337 |
| | 720 | 0.514 | 0.320 | **0.467** | **0.302** | 0.647 | 0.387 | 0.569 | 0.380 | 0.514 | 0.322 | 0.589 | 0.328 | 0.719 | 0.449 | 0.640 | 0.350 | 0.645 | 0.394 | 0.626 | 0.382 | 0.660 | 0.408 |
| | Avg | 0.465 | 0.296 | **0.428** | **0.282** | 0.626 | 0.378 | 0.522 | 0.357 | 0.481 | 0.304 | 0.550 | 0.304 | 0.760 | 0.473 | 0.620 | 0.336 | 0.625 | 0.383 | 0.610 | 0.376 | 0.628 | 0.379 |
| Weather | 96 | **0.154** | **0.199** | 0.174 | 0.214 | 0.192 | 0.232 | 0.166 | 0.210 | 0.177 | 0.218 | 0.158 | 0.230 | 0.202 | 0.261 | 0.172 | 0.220 | 0.196 | 0.255 | 0.217 | 0.296 | 0.266 | 0.336 |
| | 192 | **0.205** | **0.248** | 0.221 | 0.254 | 0.240 | 0.271 | 0.215 | 0.256 | 0.225 | 0.259 | 0.206 | 0.277 | 0.242 | 0.298 | 0.219 | 0.261 | 0.237 | 0.296 | 0.276 | 0.336 | 0.307 | 0.367 |
| | 336 | **0.262** | **0.290** | 0.278 | 0.296 | 0.292 | 0.307 | 0.287 | 0.300 | 0.278 | 0.297 | 0.272 | 0.335 | 0.287 | 0.335 | 0.280 | 0.306 | 0.283 | 0.335 | 0.339 | 0.380 | 0.359 | 0.395 |
| | 720 | **0.343** | **0.342** | 0.358 | 0.347 | 0.364 | 0.353 | 0.355 | 0.348 | 0.354 | 0.348 | 0.398 | 0.418 | 0.351 | 0.386 | 0.365 | 0.359 | 0.345 | 0.381 | 0.403 | 0.428 | 0.419 | 0.428 |
| | Avg | **0.241** | **0.270** | 0.258 | 0.278 | 0.272 | 0.291 | 0.256 | 0.279 | 0.259 | 0.281 | 0.259 | 0.315 | 0.271 | 0.320 | 0.259 | 0.287 | 0.265 | 0.317 | 0.309 | 0.360 | 0.338 | 0.382 |
| Solar-Energy | 96 | **0.200** | 0.250 | 0.203 | **0.237** | 0.322 | 0.339 | 0.221 | 0.275 | 0.234 | 0.286 | 0.310 | 0.331 | 0.312 | 0.399 | 0.250 | 0.292 | 0.290 | 0.378 | 0.242 | 0.342 | 0.884 | 0.711 |
| | 192 | **0.225** | 0.265 | 0.233 | **0.261** | 0.359 | 0.356 | 0.268 | 0.306 | 0.267 | 0.310 | 0.734 | 0.725 | 0.339 | 0.416 | 0.296 | 0.318 | 0.320 | 0.398 | 0.285 | 0.380 | 0.834 | 0.692 |
| | 336 | **0.243** | 0.283 | 0.248 | **0.273** | 0.397 | 0.369 | 0.272 | 0.294 | 0.290 | 0.315 | 0.750 | 0.735 | 0.368 | 0.430 | 0.319 | 0.330 | 0.353 | 0.415 | 0.282 | 0.376 | 0.941 | 0.723 |
| | 720 | 0.250 | 0.283 | 0.249 | **0.275** | 0.397 | 0.356 | 0.281 | 0.313 | 0.289 | 0.317 | 0.769 | 0.765 | 0.370 | 0.425 | 0.338 | 0.337 | 0.356 | 0.413 | 0.357 | 0.427 | 0.882 | 0.717 |
| | Avg | **0.230** | 0.270 | 0.233 | **0.262** | 0.369 | 0.356 | 0.260 | 0.297 | 0.270 | 0.307 | 0.641 | 0.639 | 0.347 | 0.417 | 0.301 | 0.319 | 0.330 | 0.401 | 0.291 | 0.381 | 0.885 | 0.711 |
| 1ˢᵗ Count | | **28** | **29** | 6 | 10 | 5 | 4 | 0 | 0 | 4 | 2 | 0 | 0 | 0 | 0 | 0 | 0 | 0 | 0 | 2 | 0 | 0 | 0 |

Table 16: Full results of the PEMS forecasting task. The input length is set to $T = 96$ for all baselines. *Avg* means the average results from all four prediction lengths.

| Models | | LiNo (Ours) | | iTransformer (2024b) | | Rlinear (2023a) | | TSMixer (2023) | | PatchTST (2023) | | Crossformer (2023) | | TiDE (2023) | | TimesNet (2023) | | DLinear (2023) | | FEDformer (2022b) | | Autoformer (2021) | |
|---|---|---|---|---|---|---|---|---|---|---|---|---|---|---|---|---|---|---|---|---|---|---|
| Metric | | MSE | MAE | MSE | MAE | MSE | MAE | MSE | MAE | MSE | MAE | MSE | MAE | MSE | MAE | MSE | MAE | MSE | MAE | MSE | MAE | MSE | MAE |
| PEMS03 | 12 | **0.061** | **0.163** | 0.071 | 0.174 | 0.126 | 0.236 | 0.075 | 0.186 | 0.099 | 0.216 | 0.090 | 0.203 | 0.178 | 0.305 | 0.085 | 0.192 | 0.122 | 0.243 | 0.126 | 0.251 | 0.272 | 0.385 |
| | 24 | **0.077** | **0.181** | 0.093 | 0.201 | 0.246 | 0.334 | 0.095 | 0.210 | 0.142 | 0.259 | 0.121 | 0.240 | 0.257 | 0.371 | 0.118 | 0.223 | 0.201 | 0.317 | 0.149 | 0.275 | 0.334 | 0.440 |
| | 48 | **0.113** | **0.217** | 0.125 | 0.236 | 0.551 | 0.529 | 0.121 | 0.240 | 0.211 | 0.319 | 0.202 | 0.317 | 0.379 | 0.463 | 0.155 | 0.260 | 0.333 | 0.425 | 0.227 | 0.348 | 1.032 | 0.782 |
| | 96 | **0.132** | **0.225** | 0.164 | 0.275 | 1.057 | 0.787 | 0.184 | 0.295 | 0.269 | 0.370 | 0.262 | 0.367 | 0.490 | 0.539 | 0.228 | 0.317 | 0.457 | 0.515 | 0.348 | 0.434 | 1.031 | 0.796 |
| | Avg | **0.096** | **0.197** | 0.113 | 0.221 | 0.495 | 0.472 | 0.119 | 0.233 | 0.180 | 0.291 | 0.169 | 0.281 | 0.326 | 0.419 | 0.147 | 0.248 | 0.278 | 0.375 | 0.213 | 0.327 | 0.667 | 0.601 |
| PEMS04 | 12 | **0.069** | **0.169** | 0.078 | 0.183 | 0.138 | 0.252 | 0.079 | 0.188 | 0.105 | 0.224 | 0.098 | 0.218 | 0.219 | 0.340 | 0.087 | 0.195 | 0.148 | 0.272 | 0.138 | 0.262 | 0.424 | 0.491 |
| | 24 | **0.081** | **0.184** | 0.095 | 0.205 | 0.258 | 0.348 | 0.089 | 0.201 | 0.153 | 0.275 | 0.131 | 0.256 | 0.292 | 0.398 | 0.103 | 0.215 | 0.224 | 0.340 | 0.177 | 0.293 | 0.459 | 0.509 |
| | 48 | **0.103** | **0.212** | 0.120 | 0.233 | 0.572 | 0.544 | 0.111 | 0.222 | 0.229 | 0.339 | 0.205 | 0.326 | 0.409 | 0.478 | 0.136 | 0.250 | 0.355 | 0.437 | 0.270 | 0.368 | 0.646 | 0.610 |
| | 96 | 0.137 | **0.247** | 0.150 | 0.262 | 1.137 | 0.820 | **0.133** | 0.247 | 0.291 | 0.389 | 0.402 | 0.457 | 0.492 | 0.532 | 0.190 | 0.303 | 0.452 | 0.504 | 0.341 | 0.427 | 0.912 | 0.748 |
| | Avg | **0.098** | **0.203** | 0.111 | 0.221 | 0.526 | 0.491 | 0.103 | 0.215 | 0.195 | 0.307 | 0.209 | 0.314 | 0.353 | 0.437 | 0.129 | 0.241 | 0.295 | 0.388 | 0.231 | 0.337 | 0.610 | 0.590 |
| PEMS07 | 12 | **0.055** | **0.146** | 0.067 | 0.165 | 0.118 | 0.235 | 0.073 | 0.181 | 0.095 | 0.207 | 0.094 | 0.200 | 0.173 | 0.304 | 0.082 | 0.181 | 0.115 | 0.242 | 0.109 | 0.225 | 0.199 | 0.336 |
| | 24 | **0.070** | **0.162** | 0.088 | 0.190 | 0.242 | 0.341 | 0.090 | 0.199 | 0.150 | 0.262 | 0.139 | 0.247 | 0.271 | 0.383 | 0.101 | 0.204 | 0.210 | 0.329 | 0.125 | 0.244 | 0.323 | 0.420 |
| | 48 | **0.095** | **0.189** | 0.110 | 0.215 | 0.562 | 0.541 | 0.124 | 0.231 | 0.253 | 0.340 | 0.311 | 0.369 | 0.446 | 0.495 | 0.134 | 0.238 | 0.398 | 0.458 | 0.165 | 0.288 | 0.390 | 0.470 |
| | 96 | **0.132** | **0.225** | 0.139 | 0.245 | 1.096 | 0.795 | 0.163 | 0.255 | 0.346 | 0.404 | 0.396 | 0.442 | 0.628 | 0.577 | 0.181 | 0.279 | 0.594 | 0.533 | 0.262 | 0.376 | 0.554 | 0.578 |
| | Avg | **0.088** | **0.181** | 0.101 | 0.204 | 0.504 | 0.478 | 0.112 | 0.217 | 0.211 | 0.303 | 0.235 | 0.315 | 0.380 | 0.440 | 0.124 | 0.225 | 0.329 | 0.395 | 0.165 | 0.283 | 0.367 | 0.451 |
| PEMS08 | 12 | **0.070** | **0.166** | 0.079 | 0.182 | 0.133 | 0.247 | 0.083 | 0.189 | 0.168 | 0.232 | 0.165 | 0.214 | 0.227 | 0.343 | 0.112 | 0.212 | 0.154 | 0.276 | 0.173 | 0.273 | 0.436 | 0.485 |
| | 24 | **0.093** | **0.190** | 0.115 | 0.219 | 0.249 | 0.343 | 0.117 | 0.226 | 0.224 | 0.281 | 0.215 | 0.260 | 0.318 | 0.409 | 0.141 | 0.238 | 0.248 | 0.353 | 0.210 | 0.301 | 0.467 | 0.502 |
| | 48 | **0.140** | **0.227** | 0.186 | 0.235 | 0.596 | 0.544 | 0.196 | 0.299 | 0.321 | 0.354 | 0.315 | 0.355 | 0.497 | 0.510 | 0.198 | 0.283 | 0.440 | 0.470 | 0.320 | 0.394 | 0.966 | 0.733 |
| | 96 | 0.247 | 0.283 | **0.221** | **0.267** | 1.166 | 0.814 | 0.266 | 0.331 | 0.408 | 0.417 | 0.377 | 0.397 | 0.721 | 0.592 | 0.320 | 0.351 | 0.674 | 0.565 | 0.442 | 0.465 | 1.385 | 0.915 |
| | Avg | **0.138** | **0.217** | 0.150 | 0.226 | 0.529 | 0.487 | 0.165 | 0.261 | 0.280 | 0.321 | 0.268 | 0.307 | 0.441 | 0.464 | 0.193 | 0.271 | 0.379 | 0.416 | 0.286 | 0.358 | 0.814 | 0.659 |
| 1st Count | | **18** | **19** | 1 | 1 | 0 | 0 | 1 | 0 | 0 | 0 | 0 | 0 | 0 | 0 | 0 | 0 | 0 | 0 | 0 | 0 | 0 | 0 |

Table 17: Full univariate forecasting results with prediction lengths $F \in \{96, 192, 336, 720\}$ and fixed lookback winodw $T = 96$ for all datasets.

| Models | | LiNo (Ours) | | MICN (2023) | | FEDformer (2022b) | | Autoformer (2021) | | Informer (2022a) | | LogTrans (2019) | |
|---|---|---|---|---|---|---|---|---|---|---|---|---|
| Metric | | MSE | MAE | MSE | MAE | MSE | MAE | MSE | MAE | MSE | MAE | MSE | MAE |
| ETTm1 | 96 | **0.029** | **0.126** | 0.033 | 0.134 | 0.033 | 0.140 | 0.056 | 0.183 | 0.109 | 0.277 | 0.049 | 0.171 |
| | 192 | **0.044** | **0.160** | 0.048 | 0.164 | 0.058 | 0.186 | 0.081 | 0.216 | 0.151 | 0.310 | 0.157 | 0.317 |
| | 336 | **0.058** | **0.185** | 0.079 | 0.210 | 0.084 | 0.231 | 0.076 | 0.218 | 0.427 | 0.591 | 0.289 | 0.459 |
| | 720 | **0.081** | **0.217** | 0.096 | 0.233 | 0.102 | 0.250 | 0.110 | 0.267 | 0.438 | 0.586 | 0.430 | 0.579 |
| | Avg | **0.053** | **0.172** | 0.064 | 0.185 | 0.069 | 0.202 | 0.081 | 0.221 | 0.281 | 0.441 | 0.231 | 0.382 |
| ETTm2 | 96 | 0.066 | 0.185 | **0.059** | **0.176** | 0.067 | 0.198 | 0.065 | 0.189 | 0.088 | 0.225 | 0.075 | 0.208 |
| | 192 | **0.100** | 0.235 | **0.100** | **0.234** | 0.102 | 0.245 | 0.118 | 0.256 | 0.132 | 0.283 | 0.129 | 0.275 |
| | 336 | **0.130** | **0.273** | 0.153 | 0.301 | 0.130 | 0.279 | 0.154 | 0.305 | 0.180 | 0.336 | 0.154 | 0.302 |
| | 720 | **0.176** | 0.328 | 0.210 | 0.354 | 0.178 | **0.325** | 0.182 | 0.335 | 0.300 | 0.435 | 0.160 | 0.321 |
| | Avg | **0.118** | **0.255** | 0.131 | 0.266 | 0.119 | 0.262 | 0.130 | 0.271 | 0.175 | 0.320 | 0.130 | 0.277 |
| ETTh1 | 96 | **0.056** | **0.180** | 0.058 | 0.186 | 0.079 | 0.215 | 0.071 | 0.206 | 0.193 | 0.377 | 0.283 | 0.468 |
| | 192 | **0.071** | **0.203** | 0.079 | 0.210 | 0.104 | 0.245 | 0.114 | 0.262 | 0.217 | 0.395 | 0.234 | 0.409 |
| | 336 | **0.085** | **0.228** | 0.092 | 0.237 | 0.119 | 0.270 | 0.107 | 0.258 | 0.202 | 0.381 | 0.386 | 0.546 |
| | 720 | **0.082** | **0.226** | 0.138 | 0.298 | 0.142 | 0.299 | 0.126 | 0.283 | 0.183 | 0.355 | 0.475 | 0.628 |
| | Avg | **0.074** | **0.209** | 0.092 | 0.233 | 0.111 | 0.257 | 0.105 | 0.252 | 0.199 | 0.377 | 0.345 | 0.513 |
| ETTh2 | 96 | **0.127** | 0.273 | 0.155 | 0.300 | 0.128 | **0.271** | 0.153 | 0.306 | 0.213 | 0.373 | 0.217 | 0.379 |
| | 192 | 0.176 | 0.326 | **0.169** | **0.316** | 0.185 | 0.330 | 0.204 | 0.351 | 0.227 | 0.387 | 0.281 | 0.429 |
| | 336 | **0.203** | **0.359** | 0.238 | 0.384 | 0.231 | 0.378 | 0.246 | 0.389 | 0.242 | 0.401 | 0.293 | 0.437 |
| | 720 | **0.214** | **0.371** | 0.447 | 0.561 | 0.278 | 0.420 | 0.268 | 0.409 | 0.291 | 0.439 | 0.218 | 0.387 |
| | Avg | **0.180** | **0.332** | 0.252 | 0.390 | 0.206 | 0.350 | 0.218 | 0.364 | 0.243 | 0.400 | 0.252 | 0.408 |
| Traffic | 96 | **0.138** | **0.214** | 0.158 | 0.241 | 0.207 | 0.312 | 0.246 | 0.346 | 0.257 | 0.353 | 0.226 | 0.317 |
| | 192 | **0.134** | **0.214** | 0.154 | 0.236 | 0.205 | 0.312 | 0.266 | 0.370 | 0.299 | 0.376 | 0.314 | 0.408 |
| | 336 | **0.142** | **0.223** | 0.165 | 0.243 | 0.219 | 0.323 | 0.263 | 0.371 | 0.312 | 0.387 | 0.387 | 0.453 |
| | 720 | **0.160** | **0.238** | 0.182 | 0.264 | 0.244 | 0.344 | 0.269 | 0.372 | 0.366 | 0.436 | 0.491 | 0.437 |
| | Avg | **0.143** | **0.222** | 0.165 | 0.246 | 0.219 | 0.323 | 0.261 | 0.365 | 0.309 | 0.388 | 0.355 | 0.404 |
| Weather | 96 | **0.0012** | **0.026** | 0.0029 | 0.039 | 0.0062 | 0.062 | 0.0110 | 0.081 | 0.0038 | 0.044 | 0.0046 | 0.052 |
| | 192 | **0.0015** | **0.029** | 0.0021 | 0.034 | 0.0060 | 0.062 | 0.0075 | 0.067 | 0.0023 | 0.040 | 0.0056 | 0.060 |
| | 336 | **0.0016** | **0.029** | 0.0023 | 0.034 | 0.0041 | 0.050 | 0.0063 | 0.062 | 0.0041 | 0.049 | 0.0060 | 0.054 |
| | 720 | **0.0021** | **0.035** | 0.0048 | 0.054 | 0.0055 | 0.059 | 0.0085 | 0.070 | 0.0031 | 0.042 | 0.0071 | 0.063 |
| | Avg | **0.0016** | **0.030** | 0.0030 | 0.040 | 0.0055 | 0.058 | 0.0083 | 0.070 | 0.0033 | 0.044 | 0.0058 | 0.057 |
| 1st Count | | **28** | **25** | 2 | 3 | 0 | 2 | 0 | 0 | 0 | 0 | 0 | 0 |