# OpenReview forum: "LiNo: Advancing Recursive Residual Decomposition of Linear and Nonlinear Patterns for Robust Time Series Forecasting"
_ICLR.cc/2025/Conference — ICLR 2025 Conference Withdrawn Submission_

### Official Review · Reviewer_Cccn · 2024-10-16

**Soundness:** 3
**Presentation:** 2
**Contribution:** 2
**Rating:** 5
**Confidence:** 3

**Summary:**

It's a complex time series forecasting model incorporating recursive residual, frequency, and linear/non-linear designs

**Strengths:**

The numerical performance is relative good.

**Weaknesses:**

I'm concerned that the work resembles a LEGO model: it combines various components into a high-level narrative without detailed mapping. The motivation and underlying problem are weak. The ablation should focus on the concept of Recursive Residual Decomposition instead of individual components. Removing entire modules that result in extremely poor outcomes doesn't provide meaningful insights. It would be more effective to compare it with similar residual-based designs or variations to evaluate the overarching idea.

**Questions:**

Could you elaborate on the design choices for the complex components? Beyond being a complicated module featuring recursive, residual, frequency, and linear elements, what insights can we gain from your design?

---

> ### Author Response · Authors · 2024-11-21
> **Response to Reviewer Cccn**
>
> We sincerely thank Reviewer Cccn for providing thorough and insightful comments. Here are our responses to your concerns and questions.
>
> # Weaknesses
> Thank you for your insightful comments and for raising concerns about the clarity of our paper's core contributions.
>
> We understand your concern that the work may appear to be a collection of components without a strong underlying problem statement.
>
> We appreciate the opportunity to clarify our motivations and the significance of our approach, particularly focusing on the **Recursive Residual Decomposition (RRD) concept**.
>
> ## Motivation and Solution
>
> The motivation behind our work stems from the observation that **existing time series forecasting models often struggle to disentangle the complex interplay between linear and nonlinear patterns within data**.
>
> This limitation hinders their ability to generalize and perform accurately across diverse real-world scenarios.
>
> Our goal is to address this challenge by introducing **a more nuanced decomposition strategy** that can recursively separate and model these patterns more effectively.
>
> Our RRD framework is designed to iteratively refine the decomposition process, allowing for a deeper extraction of **both linear and nonlinear patterns**.
>
> This recursive approach is distinct from **traditional methods that perform a single-level decomposition**, and we believe it is a significant advancement in time series analysis.
>
> ## Ablation Design
>
> In fact, beyond conducting ablations on individual components of the proposed LiNo, our paper mainly focus on the capacity of our general LiNo framework.
>
> Specifically, through **Table 6(a)** ("Ablation study of different No block choices"), we demonstrate that the proposed No block exhibits superior nonlinear pattern extraction capabilities compared to current state-of-the-art nonlinear models, such as iTransformer and TSMixer.
>
> Additionally, **Table 6(b)** and **Figure 3** provide evidence that our LiNo framework helps the baseline model to deliver more accurate and robust predictions.
>
> That said, your suggestion is indeed insightful, and we have incorporated it into our work.
>
> In Appendix F, we **“compare LiNo with similar residual-based designs or variations to evaluate the overarching idea.”**
>
> Experimental results show that LiNo significantly outperforms N-BEATS and its successor, N-HiTS, across the ETTm2, ECL, and Weather datasets.
>
> This highlights that, compared to leveraging Recursive Residual Decomposition (RRD) to refine final predictions based on residual errors from previous predictions (as in N-BEATS), **employing RRD to capture more nuanced linear and nonlinear patterns, as proposed in this work, is a more effective approach**.
>
> # Questions
>
> ## Model design
> The design of the LiNo framework aims to offer a deeper and more nuanced decomposition of time series signals to effectively handle real-world linear and nonlinear complexities by advancing the **Recursive Residual Decomposition (RRD)**.
>
> Specifically, the Li block extends classical techniques like moving average kernel or learnable convolution kernel, encompassing broader linear modes.
>
> The No block offers better nonlinear pattern extraction by capturing diverse nonlinear features, including temporal variations, frequency information, and inter-series dependencies.
>
> ## Take home message
> Our recursive design ensures that each pattern—linear or nonlinear—is extracted iteratively and independently, minimizing interference and fully leveraging residual information.
>
> This systematic separation not only enhances **robustness** but also boosts **interpretability** by isolating meaningful patterns at different scales.
>
> **Such a framework provides actionable insights into the intrinsic structure of time series, empowering precise and resilient forecasting.**

---

> > ### Author Response · Authors · 2024-11-23
> > **Request of Reviewer's attention and feedback**
> >
> > Dear Reviewer,
> >
> > We kindly remind you that it has been 2 days since we posted our rebuttal. Please let us know if our response has addressed your concerns.
> >
> > Following your suggestion, we have answered your concerns and improved the paper in the following aspects:
> > 1. A detailed and thorough discussion and comparison with **[N-BEATS](https://openreview.net/forum?id=r1ecqn4YwB)** in the **Sec.2** and **Appendix. F**.
> >
> > 2. A clearer and more precise model diagram, **Fig. 2**.
> >
> > 3. All images are in **PDF** format to increase readability and accessibility.
> >
> > 4. Modified all inappropriate **representation**, **typo** errors, and **reference** format.
> >
> > 5. Additional **ablation study** about the choice of model design in **Sec.4.3.2, Table.5**.
> >
> > In total, we have added **more than 50 new experiments**. All of these results have been included in the **revised paper**. Thanks again for your valuable review. We are looking forward to your reply and are happy to answer any future questions.

---

> > > ### Comment · Reviewer_Cccn · 2024-11-25
> > >
> > > From a scientific and philosophical standpoint, the concepts presented in this work do not fully align with my preferences. I question whether they will provide any additional valuable insights to the community, as the recursive, residual, frequency, and linear elements seem to form a predetermined combination. Even after rereading this work, it offers limited avenues for further exploration. However, I do appreciate the author's effort and dedication in conveying their concept. Ultimately, few works stand the test of time for their merit,  so I have adjusted my assessment accordingly

---

> ### Author Response · Authors · 2024-11-25
> **Deep thanks for your acknowledging our efforts and raising score**
>
> Dear Reviewer Cccn,
>
> **We greatly appreciate your thoughtful reflections on the scientific and philosophical aspects of our work.**
>
> We understand that the combination of recursive, residual, frequency, and linear elements may seem pre-determined, but we would like to clarify that:
>
> ***Our main contribution has never been the combination of complex models for extracting temporal, frequency, and inter-series dependencies.***
>
> ***Instead, it's the advanced **Recursive Residual Decomposition (RRD)** framework, **a generic design**, which enables a deeper and more nuanced extraction of both linear and nonlinear patterns.***
>
> As discussed and demonstrated in the paper, the **Li Block** (for extracting linear features) can be implemented using any linear model, such as [**moving average filters**](https://openreview.net/forum?id=J4gRj6d5Qm), [**exponential smoothing functions**](https://arxiv.org/abs/2202.01381), or [**learnable 1D convolution kernels**](https://openreview.net/forum?id=87CYNyCGOo). Similarly, the **No Block** (for extracting nonlinear features) can adopt any nonlinear model, such as [**iTransformer**](https://openreview.net/forum?id=JePfAI8fah) or [**TSMixer**](https://openreview.net/forum?id=wbpxTuXgm0), as shown in ***Table 6 (a)***.
>
> This method is designed to capture complex, multi-level dynamics that are often overlooked in traditional approaches. It further allows us to obtain more robust and interpretable time series forecasting results, as shown in ***Table 6 (b) and Fig.3***.
>
> We agree that this work may not resonate with all perspectives. However, this advanced extraction of both linear and nonlinear features, lays the foundation for future exploration in the area of robust and interpretable forecasting models. This contribution, we hope, will encourage further developments in both theory and application, helping the community advance towards more effective, robust, and interpretable time series modeling techniques.
>
> Finally, please allow us to quote your words: *"Ultimately, few works stand the test of time for their merit."*
>
> This is absolutely true, especially in the field of deep learning.
>
> Interestingly, in the community of Time Series Forecasting (TSF), the standout works that are ultimately remembered tend to because of their overall ideas, instead of their specific structures.
>
> For instance, [**Autoformer**](https://openreview.net/forum?id=J4gRj6d5Qm), while introducing an Auto-Correlation mechanism to replace traditional self-attention, became prominent primarily due to its **Seasonal-Trend Decomposition** concept.
>
> [**PatchTST**](https://openreview.net/forum?id=Jbdc0vTOcol) gained recognition for its key ideas of **Channel-Independent** and **Patching**.
>
> [**iTransformer**](https://openreview.net/forum?id=JePfAI8fah) is well-known for its **Whole Series Embedding** and **Channel-Wise Attention** mechanism.
>
> So we believe the **RRD** presented in this paper can provide meaningful sight to the field.
>
> **Again, we are truly grateful for the reviewer’s recognition of our efforts and dedication!**
>
> Please reconsider our contributions and insights from our work. If you have any further questions, we are looking forward to discussing with you!

---

> ### Author Response · Authors · 2024-11-25
> **A further analysis to validate the effectiveness of our proposed LiNo.**
>
> Dear Reviewer Cccn,
>
> We designed the following experiments to further validate the specific effectiveness of Recursive Residual Decomposition (RRD) in LiNo, which we hope will help provide a better understanding of our innovations and contributions.
>
> Here are some essential definitions:
>
> 1. The linear feature extractor is denoted as $Li\\_Block$, the extracted pattern $L_i$, and linear prediction $P^{li}_i$.
>
> 2. The nonlinear feature extractor is denoted as $No\\_Block$, the extracted pattern $N_i$, and nonlinear prediction $P^{no}_i$.
>
> 3. The projection to get the prediction is denoted as $FC$
>
> 4. We assume the input to the current layer in deep neural network as $H_i$ and the next layer $H_{i+1}$, where $H_0= X\\_embed=XW+b$, $X$ is the input raw time series
>
> 5. Prediction of current layer $P_{i}$
>
> 6. Final prediction $\hat{Y}$
>
> The learnable autoregressive model (AR) mentioned in the paper is employed as the linear feature extractor ($Li\\_Block$).
>
> The Temporal + Frequency projection, combined with the Tanh activation function, is adopted as the nonlinear feature extractor ($No\\_Block$).
>
> Following four schemes are designed :
>
> '**RAW**': The input features pass through the Li Block and No Block sequentially, but no feature decomposition is performed. The model's final output features are directly used for prediction (**Traditional Design**).
>
> - $L_i = Li\\_Block(H_i)$
> - $N_i = No\\_Block(L_i)$
> - $H_{i+1} = N_i$
> - $\hat{Y}=FC(H_{N})$
>
> '**LN**': The input features pass through the Li Block and No Block sequentially, with each block generating its own prediction based on the features extracted, but no residual decomposition is applied (**Common Linear-Nonlinear Decomposition Design**).
>
> - $L_i = Li\\_Block(H_i)$
> - $P^{li}_{i}=FC(L_i)$
> - $N_i = No\\_Block(L_i)$
> - $P^{no}_{i}=FC(N_i)$
> - $H_{i+1} = N_i$
> - $P^{i}=P^{li}_i+P^{no}_i$
> - $\hat{Y}=\sum_{i=1}^{N} P^{i}$
>
> 'RRD': The input features pass through the Li Block and No Block sequentially, and the output from the No Block is used for residual decomposition and prediction. This does not perform explicit residual decomposition of the linear and nonlinear features separately, but instead uses the overall features for decomposition (**Like did in N-BEATS**).
>
> - $L_i = Li\\_Block(H_i)$
> - $N_i = No\\_Block(L_i)$
> - $P_{i}=FC(N_i)$
> - $H_{i+1} = H_i-N_i$
> - $\hat{Y}=\sum_{i=1}^{N} P^{i}$
>
> 'LiNo': The input features pass through the Li Block, followed by prediction and residual decomposition. Then, the features go through the No Block for prediction and residual decomposition again (**Ours**).
>
> - $L_i = Li\\_Block(H_i)$
> - $P^{li}_{i}=FC(L_i)$
> - $R^{L}_{i} = H_i - L_i$
> - $N_i = No\\_Block(R^{L}_i)$
> - $P^{no}_{i}=FC(N_i)$
> - $R^{N}_{i} = R^{L}_i - N_i$
> - $H_{i+1} =R^{N}_{i}$
> - $P^{i}=P^{li}_i+P^{no}_i$
> - $\hat{Y}=\sum_{i=1}^{N} P^{i}$
>
> Below are the experimental results:
>
> |Models|length|RAW|RAW|LN|LN|RRD|RRD|LiNo|LiNo|
> |-|-|-|-|-|-|-|-|-|-|
> |Metric||MSE|MAE|MSE|MAE|MSE|MAE|MSE|MAE|
> |ETTh2|96|0.133|0.282|0.128|0.274|0.126|0.271|**0.126**|**0.271**|
> |     |192|0.183|0.335|0.177|0.327|0.179|0.328|**0.174**|**0.324**|
> |     |336|0.216|0.372|0.213|0.367|0.213|0.368|**0.209**|**0.364**|
> |     |720|0.227|0.384|0.229|0.385|0.226|0.381|**0.223**|**0.379**|
> |ETTm2|96|0.067|0.186|0.068|0.188|0.067|0.187|**0.066**|**0.186**|
> |     |192|0.102|0.239|0.102|0.237|0.102|0.238|**0.099**|**0.234**|
> |     |336|0.132|0.276|0.133|0.274|0.132|0.276|**0.130**|**0.272**|
> |     |720|0.185|0.334|0.185|0.334|**0.185**|**0.333**|0.187|0.335|
> |Traffic|96|0.167|0.247|0.163|0.255|0.168|0.253|**0.159**|**0.248**|
> |     |192|0.164|0.245|0.158|0.244|0.159|0.244|**0.155**|**0.241**|
> |     |336|0.163|0.247|0.158|0.245|0.156|0.242|**0.153**|**0.237**|
> |     |720|0.183|0.263|0.176|0.261|0.177|0.262|**0.174**|**0.260**|
> |Avg|   |0.160|0.284|0.158|0.283|0.158|0.282|**0.155**|**0.279**|
>
> It is clear that, although both RRD and LN show slight improvements over RAW, ***our LiNo design significantly outperforms the other designs***. This demonstrates:
>
> 1. The effectiveness of explicit linear and nonlinear modeling.
> 2. The effectiveness of RRD for representation decomposition.
> 3. The potential for further improvement by combining both approaches (**LiNo**).
>
> *We hope that these experimental results and analyses will help you gain a broad view of the contributions of this work.*
>
> BTW

---

> > ### Author Response · Authors · 2024-11-27
> > **The discussion period will be end soon**
> >
> > Dear Reviewer Cccn,
> >
> > Deep thanks for your acknowledgement of our efforts.
> >
> > ***We kindly remind you that the reviewer-author discussion phase will end soon. After that, we may not be able to respond to your comments.***
> >
> > We've tried our every effort to convey our main contribution, namely **the advanced Recursive Residual Decomposition
> >  of Linear and Nonlinear patterns**. This is a generic and model-agnostic framework, with Li Block (for linear pattern extraction) and No (for nonlinear pattern extraction) can take any form.
> >
> > Overall, we have made the following efforts:
> >
> > 1. Comparison and analysis of the overall model design across different configurations (especially LiNo and N-BEATS, in **Appendix. F**.).
> > 2. Comparison of predictive performance of baseline model under different model designs (LiNo achieves more accurate predictions, in **Table.6 (b), Table.12**.).
> > 3. Analysis of the predictive robustness of the baseline model under different designs (LiNo provides more robust predictions, in **Fig.3**.).
> > 4. Error bar analysis (LiNo exhibits stable predictions, in **Table 8**.).
> > 5. Visualization of the prediction results of different Blocks (the Li Block captures long-term trends and other linear patterns, while the No Block focuses more on nonlinear characteristics like short-term fluctuations, in **Fig 4**).
> > 6. Visualization of the Block weights (differences in weight patterns between Li Block and No Block, as well as among different Li Blocks, and different No Blocks. These underscore **the necessity of disentangling linear and nonlinear components, as well as further subdividing them**, in **Fig 19-31**).
> > 7. Comparison on both **Univariate and Multivariate** time series forecasting benchmarks (**LiNo achieves SOTA on both feild**).
> > 8. Further investigation into model efficiency, the impact of input length, loss functions, and model hyperparameters. (in **Table 9, Fig.5, Table 11, Table 4**)
> >
> > Our recursive design iteratively and independently extracts both linear and nonlinear patterns, minimizing interference and maximizing residual information. This separation enhances robustness and interpretability by isolating meaningful patterns at different scales, providing valuable insights into the intrinsic structure of time series for more precise and resilient forecasting.
> >
> > We hope these qualities of our work can come into your attention.
> >
> > BTW

---

> > > ### Author Response · Authors · 2024-11-29
> > > **Request for reconsideration of score based on additional experiments and insights**
> > >
> > > Dear Reviewer Cccn,
> > >
> > > Thank you once again for your thoughtful feedback and for acknowledging our efforts.
> > >
> > > We understand that the philosophical perspective of our work may not fully align with your preferences. ***However, I believe we both share the same view that diverse perspectives may bring new possibilities to the entire community.***
> > >
> > > We hope the additional analyses and experiments we've conducted, especially the validation of our Recursive Residual Decomposition (RRD) approach, clarify the significant impact of our framework (**with more than 200 new results, 7 new pages**).
> > >
> > > Our design focuses on disentangling linear and nonlinear patterns, enhancing both robustness and interpretability, as highlighted in our experiments and visualizations. **All these updates during rebuttal significantly strengthen our work and demonstrate the effectiveness of our approach.**
> > >
> > > With the discussion phase soon closing, we kindly ask that you reconsider your assessment in light of these additional insights. Your evaluation is of great importance to us.
> > >
> > > Thank you again for your time and consideration.

---

> > > > ### Author Response · Authors · 2024-12-02
> > > > **Kindly Request for Reviewer's Feedback**
> > > >
> > > > Dear Reviewer Cccn,
> > > >
> > > > Thank you for taking the time and effort in providing a valuable review of our work. As the discussion period is coming to a close, we hope that you have had the chance to review our rebuttal.
> > > >
> > > > Since the End of author/reviewer discussions is coming soon, **may we know if our response addresses your main concerns? If so, we kindly ask for your reconsideration of the score.** If you have any further concerns, please let us know, and we will be more than happy to engage in more discussion and paper improvements.
> > > >
> > > > Once again, thank you for your suggestion and time!
> > > >
> > > > Best regards,
> > > >
> > > > Authors

---

> > > > > ### Author Response · Authors · 2024-12-03
> > > > > **A Gentle Reminder of Feedback (Last Few Hours of Discussion Period)**
> > > > >
> > > > > Dear Reviewer Cccn,
> > > > >
> > > > > Thank you for taking the time and effort to provide a thoughtful review of our work. We have carefully addressed each of your insightful comments in our rebuttal.
> > > > >
> > > > > As the discussion period is nearing its end, we hope you’ve had the opportunity to review our responses. We remain eager to hear your feedback and are ready to address any additional concerns you might have. If our rebuttal has resolved your concerns or clarified any aspects of the paper, we would greatly appreciate it if you could reconsider your score. This would provide us with a valuable opportunity to present our work at the conference.
> > > > >
> > > > > Once again, thank you for your time and for offering valuable insights.
> > > > >
> > > > > Best regards,
> > > > >
> > > > > Authors

---

> > > > > > ### Author Response · Authors · 2024-12-03
> > > > > > **Summary of Response to Reviewer Cccn**
> > > > > >
> > > > > > Dear Reviewer Cccn,
> > > > > >
> > > > > > With **less than 3 hours** remaining before the discussion period concludes, may we know if our response addresses your main concerns?
> > > > > >
> > > > > > We have provided a more precise explanation of our contribution, addressed the concerns raised, and expanded our experimental analyses to further validate our approach.
> > > > > >
> > > > > > We hope you will reconsider the insights and contributions from our work in light of these improvements (**more than 200 new results, 7 new pages**).
> > > > > >
> > > > > > If you have any further questions, we would be glad to continue the discussion.
> > > > > >
> > > > > > Best regards,
> > > > > >
> > > > > > Authors

---

> ### Author Response · Authors · 2024-12-04
> **Kindly Request for Reviewer's Feedback**
>
> Dear Reviewer Cccn,
>
> Thank you once again for your thoughtful feedback and for acknowledging our efforts.
>
> We understand that the philosophical perspective of our work may not fully align with your preferences. However, I believe **we both agree that diverse perspectives are crucial for fostering new ideas and advancing the field.**
>
> We hope the additional analyses and experiments we’ve conducted, particularly the validation of our Recursive Residual Decomposition (RRD) approach, clarify the significant impact of our framework. With over **200 new results and 7 new pages** of content, we have strived to present a more robust and comprehensive argument for the effectiveness of our approach. We believe that these updates strengthen the work and demonstrate its potential to contribute meaningfully to the community.
>
> Given that our current score of **5.5** places us at significant risk of rejection, we would greatly appreciate it if you could consider your assessment in light of these updates and insights. Your evaluation is crucial to our chances of acceptance, and we would deeply appreciate your reconsideration of the score.
>
> We also wanted to highlight that, since the extended discussion period began, our score has **remained unchanged**. During these 6 additional days, ***many authors have engaged in fruitful discussions with their reviewers, receiving positive feedback and score increases***. We sincerely hope that our case is no exception, and that the added clarifications and updates will also lead to a more comprehensive reassessment.
>
> Thank you again for your time and thoughtful consideration.
>
> Best regards,
>
> Authors

---

### Official Review · Reviewer_dLTN · 2024-10-31

**Soundness:** 2
**Presentation:** 2
**Contribution:** 2
**Rating:** 5
**Confidence:** 3

**Summary:**

This paper introduces LiNo, a Recursive Residual Decomposition framework for time series forecasting. LiNo addresses the limitations of existing deep learning models by extracting both linear and nonlinear patterns. The framework employs a Li block to capture linear patterns and a No block to model nonlinear patterns. This recursive and deeper-level decomposition enhances the extraction of complex, multi-granular patterns in real-world time series data. Experimental results on thirteen real-world benchmarks show that LiNo outperforms state-of-the-art models in both univariate and multivariate forecasting scenarios, providing robust and precise predictions.

**Strengths:**

1. In experimental evaluations, the proposed method surpasses state-of-the-art models in both univariate and multivariate forecasting scenarios.
2. The method is easy to understand.
3. This paper proposes an efficient method to capture the linear and nonlinear patterns for time series forecasting.

**Weaknesses:**

1. The components of the method, while effective, are not entirely novel. Recursive residual decomposition has been demonstrated to be effective in N-BEATS [1], and the utility of frequency-domain MLPs is highlighted in FreMLP [2]. Additionally, the channel mixing technique used in LiNo is similar to that used in SOFTS [3], and it would be beneficial to include a more detailed comparison and discussion of these similarities.
2. The authors argue that the primary distribution involves linear and nonlinear decomposition, with ablation studies primarily focused on multivariate time series forecasting. In line 428, they state, "Extracting nonlinear patterns, such as inter-series dependencies, temporal variations, and frequency information, is crucial for accurate predictions." The classification of inter-series dependencies as nonlinear patterns may be debatable, as methods like the Multivariate Autoregressive Model, which are inherently linear, can also capture such dependencies. It would be valuable to separately study the effects of the channel mixing method and the linear-nonlinear decomposition. The claimed superiority of the No block ablation, using iTransformer and TSMixer, which mainly attributes its performance to the channel mixing part, could benefit from further evidence (Table 5, Figure 4).
3. The paper lacks an error bar analysis. To demonstrate the robustness of the proposed method, the experiments should be conducted multiple times using different random seeds.
4. The paper contains a few minor typographical errors. For instance, on line 103, "TSMxier" should be corrected to "TSMixer," and on line 252, "ReVIN" should be changed to "RevIN."

[1] Oreshkin B N, Carpov D, Chapados N, et al. N-BEATS: Neural basis expansion analysis for interpretable time series forecasting[J]. ICLR, 2020.

[2] Yi K, Zhang Q, Fan W, et al. Frequency-domain MLPs are more effective learners in time series forecasting[J]. NeurIPS, 2023.

[3] Lu H, XY Chen, et al. SOFTS: Efficient Multivariate Time Series Forecasting with Series-Core Fusion. ArXiv, 2024.

**Questions:**

Please see the weaknesses above.

---

> ### Author Response · Authors · 2024-11-21
> **Response to Reviewer dLTN**
>
> We sincerely thank Reviewer dLTN for acknowledging our strength and providing thorough and insightful comments. Here are our responses to your concerns and questions.
>
> # Weaknesses
>
> > **W.1 The components of the method, while effective, are not entirely novel....**
>
> **The similarities of LiNo and N-BEATS** has been discussed in the updated PDF, **Sec.2**, where we denoted the model designed in N-BEATS's fashion as **Mu**.
>
> When check experimental results in Table.5(b) and Fig.4, counterintuitively, the usage of RRD in N-BEATS (**Mu**) does not significantly outperform classic designs in terms of either forecasting accuracy or robustness to noise.
>
> In contrast, our LiNo framework excels in both aspects, which demonstrates that LiNo advanced RRD for more accurate and robust forecasting.
>
> Actually, utility of frequency-domain MLPs in this work is more related to **FITS**, instead of that in **FreMLP**. But we are more than happy to include and discuss the similarity.
>
> When we start this work, we do not notice that there is a work that so close to our channel mixing technique, thanks for mention it!!
>
> I think our channel mixing technique and **SOFTS** both stems from the idea of learning channel-dependence, while maintain channel-independence information.
>
> Here are some major differnce:
>
> 1. LiNo using weighted sum of all channels with weight generated by softmax function, while SOFTS used a stochastic pooling technique, which is more time consuming.
> 2. We directly perform softmax to input feature, while SOFTS first go through a Feedforward Network.
>
> > **W.2 The authors argue that the primary distribution involves linear and nonlinear decomposition, with ablation studies primarily focused on multivariate time series forecasting...**
>
> Thank you for your suggestion!
>
> We have **redesigned an ablation experiment without the interference of inter-series dependencies**.
>
> Specifically, we designed a simplified No Block with the following variants: using only Temporal projection, only Frequency projection, Temporal + Frequency projection (**TF**), and introducing nonlinearity (Tanh activation function) on top of TF.
>
> This simplified ablation study effectively demonstrates that extracting both temporal variations and frequency information, and the introducing of nonlinearity, is crucial for accurate predictions.
>
> |&nbsp;Models &nbsp; | Length &nbsp;| w/o&nbsp;No&nbsp;Block MSE | w/o&nbsp;No&nbsp;Block MAE | Temporal MSE | Temporal MAE | Frequency MSE | Frequency MAE | TF MSE | TF MAE | TF+Tanh MSE | TF+Tanh MAE |
> |----------|--------|------------------|------------------|--------------|--------------|---------------|---------------|--------|--------|-------------|-------------|
> | **ETTh1**| 96     | 0.395            | 0.407            | 0.385        | 0.398        | 0.383         | 0.393         | 0.383  | 0.392  | 0.375       | 0.394       |
> |          | 720    | 0.483            | 0.468            | 0.487        | 0.475        | 0.482         | 0.469         | 0.478  | 0.467  | 0.464       | 0.458       |
> | **ETTm1**| 96     | 0.344            | 0.371            | 0.341        | 0.369        | 0.342         | 0.369         | 0.338  | 0.366  | 0.322       | 0.359       |
> |          | 720    | 0.495            | 0.459            | 0.481        | 0.447        | 0.479         | 0.445         | 0.474  | 0.439  | 0.465       | 0.442       |
> | **ECL**  | 96     | 0.186            | 0.267            | 0.186        | 0.268        | 0.187         | 0.267         | 0.183  | 0.263  | 0.150       | 0.243       |
> |          | 720    | 0.245            | 0.320            | 0.245        | 0.320        | 0.245         | 0.320         | 0.241  | 0.317  | 0.221       | 0.308       |
> | **Weather**| 96    | 0.162            | 0.206            | 0.165        | 0.209        | 0.165         | 0.208         | 0.162  | 0.206  | 0.163       | 0.209       |
> |          | 720    | 0.342            | 0.338            | 0.344        | 0.344        | 0.345         | 0.343         | 0.342  | 0.338  | 0.343       | 0.339       |
>
> **Regarding the ablation of the No Block which proved its superiority, please forgive us for having a different perspective**.
>
> Our argument lies in the fact that iTransformer and TSMixer primarily attribute their performance to the channel mixing part, which makes them less capable of extracting more complex nonlinear patterns in real-world time series.
>
> In contrast, our No block is designed to extract nonlinear features other than inter-series dependencies, such as nonlinear temporal variations and frequency information, which leads to better performance.
>
> > **W.3 The paper lacks an error bar analysis.**
>
> We conducted an error bar analysis (using five different random seeds), but to save space in the main text, we included it in **Appendix C.1**.
>
> > **W.4 The paper contains a few minor typo errors...**
>
> Deep thanks for all the invaluable suggestions! We've revised the paper according to your invaluable comments.

---

> > ### Author Response · Authors · 2024-11-23
> > **Request of Reviewer's attention and feedback**
> >
> > Dear Reviewer,
> >
> > We kindly remind you that it has been 2 days since we posted our rebuttal. Please let us know if our response has addressed your concerns.
> >
> > Following your suggestion, we have answered your concerns and improved the paper in the following aspects:
> > 1. A detailed and thorough discussion and comparison with **[N-BEATS](https://openreview.net/forum?id=r1ecqn4YwB)** in the **Sec.2** and **Appendix. F**.
> >
> > 2. A clearer and more precise model diagram, **Fig. 2**.
> >
> > 3. All images are in **PDF** format to increase readability and accessibility.
> >
> > 4. Modified all inappropriate **representation**, **typo** errors, and **reference** format.
> >
> > 5. Additional **ablation study** about the choice of model design in **Sec.4.3.2, Table.5**.
> >
> > In total, we have added **more than 50 new experiments**. All of these results have been included in the **revised paper**. Thanks again for your valuable review. We are looking forward to your reply and are happy to answer any future questions.

---

> ### Comment · Reviewer_dLTN · 2024-11-25
> **Response to the rebuttal**
>
> Thank you for your response.
>
> **W1.** Thank you for your clarification. I appreciate that the revised paper includes comparisons with N-BEATS and FITS, which are helpful and well-placed. However, while the rebuttal acknowledges the similarity between your method and SOFTS, this comparison has not been explicitly discussed or incorporated into the paper itself.
>
> **W2.** Thank you for your effort and for redesigning the ablation experiments to provide additional insights. However, my concern remains that it is still unclear whether the performance gains primarily stem from the separation of channel-independent and channel-dependent components, or from the extraction of nonlinear features (beyond channel dependence). My question is less about the individual contributions of each part of the No-block and more about pinpointing the specific factor that makes your decomposition method genuinely effective.
>
> **W3.** Thank you, this has been resolved.
>
> **W4.** Thank you, this has been resolved.

---

> ### Author Response · Authors · 2024-11-25
> **The Second Response to Reviewer dLTN (Part 1)**
>
> Dear Reviewer dLTN,
>
> We are glad that our response has helped clarify some of your concerns. Regarding your confusion about **W.1&2**, we will provide further clarification here.
>
> ## W.1 While the rebuttal acknowledges the similarity between your method and **SOFTS**, this comparison has not been explicitly discussed or incorporated into the paper itself.
>
> Due to time constraints, we had originally planned to include this discussion in the final camera-ready version, which is why we were unable to update our PDF in time.
>
> Now, please refer to the latest version of the PDF, where we have presented this content. Kindly check ***Sec. 3.2, Lines 254-255***, as well as ***Appendix G***.
>
> ## W.2 My question is more about pinpointing the specific factor that makes your decomposition method genuinely effective.
>
> Thank you again for acknowledging our efforts and for articulating your questions and concerns so precisely and patiently.
>
> We will further address your remain concerns, specifically regarding the overall effectiveness of the decomposition proposed in LiNo, beyond the validity of the individual components.
>
> We designed the following experiments to investigate the specific effectiveness of Recursive Residual Decomposition (RRD) in LiNo under a ***Univariate Scenario***, in order to exclude any interference from channel-independent or channel-dependent information.
>
> Here are some essential definitions:
>
> 1. The linear feature extractor is denoted as $Li\\_Block$, the extracted pattern $L_i$, and linear prediction $P^{li}_i$.
>
> 2. The nonlinear feature extractor is denoted as $No\\_Block$, the extracted pattern $N_i$, and nonlinear prediction $P^{no}_i$.
>
> 3. The projection to get the prediction is denoted as $FC$
>
> 4. We assume the input to the current layer in deep neural network as $H_i$ and the next layer $H_{i+1}$, where $H_0= X\\_embed=XW+b$, $X$ is the input raw time series
>
> 5. Prediction of current layer $P_{i}$
>
> 6. Final prediction $\hat{Y}$
>
> The learnable autoregressive model (AR) mentioned in the paper is employed as the linear feature extractor ($Li\\_Block$).
>
> The Temporal + Frequency projection, combined with the Tanh activation function, is adopted as the nonlinear feature extractor ($No\\_Block$).
>
> Following four schemes are designed :
>
> '**RAW**': The input features pass through the Li Block and No Block sequentially, but no feature decomposition is performed. The model's final output features are directly used for prediction (**Traditional Design**).
>
> - $L_i = Li\\_Block(H_i)$
> - $N_i = No\\_Block(L_i)$
> - $H_{i+1} = N_i$
> - $\hat{Y}=FC(H_{N})$
>
> '**LN**': The input features pass through the Li Block and No Block sequentially, with each block generating its own prediction based on the features extracted, but no residual decomposition is applied (**Common Linear-Nonlinear Decomposition Design**).
>
> - $L_i = Li\\_Block(H_i)$
> - $P^{li}_{i}=FC(L_i)$
> - $N_i = No\\_Block(L_i)$
> - $P^{no}_{i}=FC(N_i)$
> - $H_{i+1} = N_i$
> - $P^{i}=P^{li}_i+P^{no}_i$
> - $\hat{Y}=\sum_{i=1}^{N} P^{i}$
>
> 'RRD': The input features pass through the Li Block and No Block sequentially, and the output from the No Block is used for residual decomposition and prediction. This does not perform explicit residual decomposition of the linear and nonlinear features separately, but instead uses the overall features for decomposition (**Like did in N-BEATS**).
>
> - $L_i = Li\\_Block(H_i)$
> - $N_i = No\\_Block(L_i)$
> - $P_{i}=FC(N_i)$
> - $H_{i+1} = H_i-N_i$
> - $\hat{Y}=\sum_{i=1}^{N} P^{i}$
>
> 'LiNo': The input features pass through the Li Block, followed by prediction and residual decomposition. Then, the features go through the No Block for prediction and residual decomposition again (**Ours**).
>
> - $L_i = Li\\_Block(H_i)$
> - $P^{li}_{i}=FC(L_i)$
> - $R^{L}_{i} = H_i - L_i$
> - $N_i = No\\_Block(R^{L}_i)$
> - $P^{no}_{i}=FC(N_i)$
> - $R^{N}_{i} = R^{L}_i - N_i$
> - $H_{i+1} =R^{N}_{i}$
> - $P^{i}=P^{li}_i+P^{no}_i$
> - $\hat{Y}=\sum_{i=1}^{N} P^{i}$

---

> ### Author Response · Authors · 2024-11-25
> **The Second Response to Reviewer dLTN (Part 2)**
>
> **Here are the experimental results:**
>
> |Models|length|RAW|RAW|LN|LN|RRD|RRD|LiNo|LiNo|
> |-|-|-|-|-|-|-|-|-|-|
> |Metric||MSE|MAE|MSE|MAE|MSE|MAE|MSE|MAE|
> |ETTh2|96|0.133|0.282|0.128|0.274|0.126|0.271|**0.126**|**0.271**|
> ||192|0.183|0.335|0.177|0.327|0.179|0.328|**0.174**|**0.324**|
> ||336|0.216|0.372|0.213|0.367|0.213|0.368|**0.209**|**0.364**|
> ||720|0.227|0.384|0.229|0.385|0.226|0.381|**0.223**|**0.379**|
> |ETTm2|96|0.067|0.186|0.068|0.188|0.067|0.187|**0.066**|**0.186**|
> ||192|0.102|0.239|0.102|0.237|0.102|0.238|**0.099**|**0.234**|
> ||336|0.132|0.276|0.133|0.274|0.132|0.276|**0.130**|**0.272**|
> ||720|0.185|0.334|0.185|0.334|**0.185**|**0.333**|0.187|0.335|
> |Traffic|96|0.167|0.247|0.163|0.255|0.168|0.253|**0.159**|**0.248**|
> ||192|0.164|0.245|0.158|0.244|0.159|0.244|**0.155**|**0.241**|
> ||336|0.163|0.247|0.158|0.245|0.156|0.242|**0.153**|**0.237**|
> ||720|0.183|0.263|0.176|0.261|0.177|0.262|**0.174**|**0.260**|
> |Avg||0.160|0.284|0.158|0.283|0.158|0.282|**0.155**|**0.279**|
>
> It is clear that, although both RRD and LN show slight improvements over RAW, ***our LiNo design significantly outperforms the other designs***. This demonstrates:
>
> 1. The effectiveness of explicit linear and nonlinear modeling.
> 2. The effectiveness of RRD for representation decomposition.
> 3. The potential for further improvement by combining both approaches (**LiNo**).
>
> *This result and analysis will be included in the camera-ready version, too*.
>
> BTW

---

> > ### Author Response · Authors · 2024-11-27
> > **The discussion period will be end soon**
> >
> > Dear Reviewer dLTN,
> >
> > We deeply appreciate your valuable and constructive review, which helps us clarify our work to a clearer stage.  And we are dedicated to solve all your concerns.
> >
> > ***We kindly remind you that the reviewer-author discussion phase will end soon. After that, we may not have a chance to respond to your comments.***
> >
> > During the rebuttal period, following your suggestion, we've included through discussion about **N-BEATS**, **FITS**, and especially **SOFTS** both in the main text and Appendix.
> >
> > Besides, we also additionally conduct further ablation study regarding **the overall effectiveness of the decomposition** proposed in LiNo, **beyond the validity of the individual components.**
> >
> > All mentioned information can be found in the updated PDF (**in purple**).
> >
> > We hope these information could be helpful to further clarify your concerns.
> >
> > Thanks again for your dedication in reviewing our paper. It helps us deeply.
> >
> > We are looking forward to your feedback!
> >
> > BTW

---

> > > ### Comment · Reviewer_dLTN · 2024-11-27
> > > **Maintain my score**
> > >
> > > I share reviewer Cccn's concerns about the limited novelty of this work. Each component of the proposed method appears to lack originality, as similar ideas have already been explored in time-series tasks. Furthermore, the experiments and analysis presented in both the paper and the rebuttal fail to offer sufficiently meaningful insights.
> > >
> > > Overall, the analysis and justifications provided do not meet the standards expected for ICLR, and I am inclined to maintain my original score.

---

> > > > ### Author Response · Authors · 2024-11-27
> > > > **Response to Reviewer dLTN**
> > > >
> > > > Dear Reviewer dLTN,
> > > >
> > > > Firstly, we would like to know if your concerns regarding **W.1&2** have been addressed.
> > > >
> > > > Regarding your latest two concerns, let me denote them as C1&2, and clarify them one by one.
> > > >
> > > > > C.1 Each component of the proposed method appears to lack originality, as similar ideas have already been explored in time-series tasks.
> > > >
> > > > Considering the shared concern with Reviewer Cccn that "each component of the proposed method appears to lack originality," we would like to reiterate that, as mentioned in the **Abstract and TL;DR**, the focus of this work is not on the novelty or effectiveness of each individual component. Instead, our main contribution lies in the advanced Recursive Residual Decomposition of Linear and Nonlinear patterns.  **Your valuable suggestion also highlights that our ablation experiments should place greater emphasis on evaluating the effectiveness of the overall decomposition design (which we have already demonstrated through our experiments).**
> > > >
> > > > ***This is a generic, **model-agnostic** framework, where both the Li Block (for linear pattern extraction) and No Block (for nonlinear pattern extraction) can be adapted to any form.*** Such as [**moving average filters**](https://openreview.net/forum?id=J4gRj6d5Qm), [**exponential smoothing functions**](https://arxiv.org/abs/2202.01381), or [**learnable 1D convolution kernels**](https://openreview.net/forum?id=87CYNyCGOo) for Li Block, [**iTransformer**](https://openreview.net/forum?id=JePfAI8fah) or [**TSMixer**](https://openreview.net/forum?id=wbpxTuXgm0) for No Block.
> > > >
> > > > As we shared with Reviewer Cccn, in the community of Deep Learning for Time Series Forecasting, **the standout works are distinguished by their overall ideas, rather than their specific structures.**
> > > >
> > > > For instance, [**N-BEATS**](https://openreview.net/forum?id=J4gRj6d5Qm) is not renowned for its intricate model design, but rather for its approach of **refining predictions through residuals**.
> > > >
> > > > [**PatchTST**](https://openreview.net/forum?id=Jbdc0vTOcol) gained recognition for its key ideas of **Channel-Independent** and **Patching**.
> > > >
> > > > [**iTransformer**](https://openreview.net/forum?id=JePfAI8fah) is well-known for its **Whole Series Embedding** and **Channel-Wise Attention** mechanism.
> > > >
> > > > All of them are outstanding works at ICLR, **known for their overall ideas rather than relying on the design of individual components.**
> > > >
> > > > From this perspective, our LiNo, with its core idea of iteratively and independently extracting both linear and nonlinear patterns, which haven't been explored in previous literature, as acknowledged by you, reviewer Bbht and 236E, is fully aligned with the standards expected for ICLR.
> > > >
> > > > > C.2 The experiments and analysis presented in both the paper and the rebuttal fail to offer sufficiently meaningful insights
> > > >
> > > > We believe that our extensive experiments do, in fact, offer meaningful insights into the design choices of our model, particularly regarding the disentangling of linear and nonlinear components and the efficacy of deeper recursive residual decomposition.
> > > >
> > > > Let's get into details:
> > > > 1. In Appendix F, we provide through comparison of LiNo with N-BEATS, including their distinct design principles. We show how LiNo’s recursive approach allows it to capture both linear and nonlinear components in a deeper-level manner, whereas N-BEATS primarily refines predictions based on residuals. And the experimental superiority of LiNo demonstrate the necessity of the separation of linear and nonlinear patterns.
> > > >
> > > > 2. In Tables 6(b) and 12, we demonstrate that baseline models designed in LiNo-fashion delivers more precise forecasting.
> > > >
> > > > 3. In Fig.3 and Table.8 (The error bar analysis), the robustness of LiNo framework is well-illustrated.
> > > >
> > > > 4. Visualization of the prediction results of different Blocks in Fig.4 and Visualization of the Block weights (in Fig 19-31) effectively demonstrate the different preferences of linear and nonlinear models for time series forecasting, as well as the distinction between linear and nonlinear patterns. These underscore the necessity of disentangling linear and nonlinear components, as well as further subdividing them.
> > > > 5. Remarkable performance of the proposed LiNo on **both Univariate and Multivariate time series forecasting benchmarks (Table 1&2)** suggests the mightness of LiNo. (LiNo not only achieves SOTA but also outperform other baselines by a lager scale, such as a decrease of **11.89%** in average MSE on the four PEMS-relevant benchmarks, a notoriously difficult benchmark, or a **7.87%** reduction of MSE on the ECL dataset.)
> > > >
> > > > We hope these aspects of our work can come into your attention.
> > > >
> > > > BTW

---

> > > > > ### Author Response · Authors · 2024-11-28
> > > > > **A very significant and urgent supplementary experiment.**
> > > > >
> > > > > Dear Reviewer dLTN,
> > > > >
> > > > > Many thanks for your valuable suggestions, instructive responses, and detailed descriptions of your concerns, which have inspired us to improve our paper substantially.
> > > > >
> > > > > Considering your second concern, which is that our experiments may not be sufficient, after careful reconsideration and inspection, we realize that **we have indeed overlooked a very important experiment**, which we believe you would also consider crucial.
> > > > >
> > > > > **It's the direct comparison of forecasting performance between the proposed No Block and SOFTS.**
> > > > >
> > > > > We apologize for the delay in updating the document. Due to the large number of additional experiments added during the rebuttal period, we were unable to make the updates in time. Fortunately, we managed to complete the experiments before today’s final PDF submission deadline, and have incorporated them into the latest and final version of our PDF, **Table 14**.
> > > > >
> > > > > Below are the experiment result, with number of SOFTS directly referenced from [the original paper](https://openreview.net/forum?id=89AUi5L1uA&referrer=%5Bthe%20profile%20of%20Han-Jia%20Ye%5D(%2Fprofile%3Fid%3D~Han-Jia_Ye1)), and number of No Block are obtained by replace the backbone of SOFTS using No Block under the same pipeline.
> > > > >
> > > > > |Models|length|ETTh1|ETTh1|ETTh2|ETTh2|ETTm1|ETTm1|ETTm2|ETTm2|Weather|Weather|
> > > > > |-|-|-|-|-|-|-|-|-|-|-|-|
> > > > > |Metric||MSE|MAE|MSE|MAE|MSE|MAE|MSE|MAE|MSE|MAE|
> > > > > |SOFTS|96|0.381|0.399|0.297|0.347|**0.325**|**0.361**|0.180|0.261|**0.166**|**0.208**|
> > > > > |No Block|96|**0.377**|**0.394**|**0.293**|**0.342**|0.333|0.365|**0.175**|**0.257**|0.172|0.211|
> > > > > |SOFTS|192|0.435|0.431|0.373|0.394|0.375|0.389|0.246|0.306|0.217|0.253|
> > > > > |No Block|192|**0.423**|**0.420**|**0.372**|**0.390**|**0.369**|**0.385**|**0.241**|**0.300**|**0.214**|**0.249**|
> > > > > |SOFTS|336|0.480|0.452|**0.410**|0.426|**0.405**|0.412|0.319|0.352|0.282|0.300|
> > > > > |No Block|336|**0.457**|**0.440**|0.414|**0.425**|0.407|**0.407**|**0.302**|**0.340**|**0.276**|**0.296**|
> > > > > |SOFTS|720|0.499|0.488|**0.411**|**0.433**|**0.466**|0.447|0.405|0.401|0.356|0.351|
> > > > > |No Block|720|**0.454**|**0.457**|0.417|0.438|0.470|**0.440**|**0.399**|**0.396**|**0.353**|**0.347**|
> > > > >
> > > > > It's crystal clear that our No Block outperforms SOFTS by a significant margin.
> > > > >
> > > > > Since the extended period for modifying the PDF has ended, we hope that our efforts in the rebuttal period (**more than 200 new results, 7 new pages**) have addressed your concerns to your satisfaction.
> > > > >
> > > > > Overall, we will try our best to ensure every question of every reviewer have been solved.
> > > > >
> > > > > We eagerly await your reply and are happy to answer any further questions.

---

> > > > > > ### Author Response · Authors · 2024-11-29
> > > > > > **Your response carries significant weight**
> > > > > >
> > > > > > Dear Reviewer dLTN,
> > > > > >
> > > > > > Thank you again for your thoughtful feedback.
> > > > > >
> > > > > > We sincerely hope the additional experiments and the in-depth analysis we've provided, particularly the comparison between No Block and SOFTS (Table 14), further address your concerns. **We believe these updates significantly strengthen our work and demonstrate the effectiveness of our approach.**
> > > > > >
> > > > > > Given the importance of your feedback at this stage, we respectfully request that you reconsider the score based on the comprehensive revisions we've made.
> > > > > >
> > > > > > We appreciate your time and consideration.

---

> > > > > > > ### Author Response · Authors · 2024-12-02
> > > > > > > **Kindly Request for Reviewer's Feedback**
> > > > > > >
> > > > > > > Dear reviewer dLTN,
> > > > > > >
> > > > > > > Thank you for taking the time and effort in providing a valuable review of our work. As the discussion period is coming to a close, we hope that you have had the chance to review our rebuttal.
> > > > > > >
> > > > > > > If our rebuttal has resolved your concerns or improved your understanding of the paper, **we would greatly appreciate it if you could reconsider your assessment and update the score accordingly.** Your feedback has been incredibly helpful, and we value the opportunity to further improve the work based on your insights.
> > > > > > >
> > > > > > > Thank you again for your thoughtful review and for considering our responses. Please feel free to reach out if you have any additional questions or require further clarifications.
> > > > > > >
> > > > > > > Best regards,
> > > > > > >
> > > > > > > Authors

---

> ### Author Response · Authors · 2024-12-03
> **Discussion Period Ending Soon. Eagerly Await Your Response.**
>
> Dear reviewer dLTN,
>
> We sincerely thank you for your insightful and detailed review. Your valuable suggestions have been instrumental in improving our paper.
>
> With only a few hours remaining before the deadline for the reviewer-author discussion phase, we kindly ask if our responses have adequately addressed your concerns.
>
> We look forward to your feedback and would happily answer any further questions. We also hope you will reconsider your original rating.
>
> Best regards,
>
> Authors

---

> ### Author Response · Authors · 2024-12-03
> **Summary of Response to Reviewer dLTN**
>
> Dear reviewer dLTN,
>
> With **less than 3 hours** remaining before the discussion period concludes, we are eager to receive your feedback. Please allow us to briefly summarize the key points of our rebuttal phase for your reference.
>
> In the rebuttal, we emphasized the uniqueness of our Recursive Residual Decomposition (RRD) framework, which iteratively and independently extracts both **linear and nonlinear** patterns, addressing gaps in prior methods like N-BEATS.
>
> We clarified that **LiNo’s core innovation lies in its holistic decomposition approach, not just component designs.**
>
> Experiments, including ablation studies, error bar analysis, and visualizations, demonstrated LiNo’s superior forecasting performance, robustness, and interpretability.
>
> Additionally, we incorporated new comparisons, such as with SOFTS, and improved the paper with clearer diagrams, fixed typos, and extended analyses (e.g., Tables 5, 6(b)).
>
> Our efforts in the rebuttal period include **more than 200 new results, 7 new pages**. These updates solidify our contributions, and we hope them have addressed all your concerns comprehensively.
>
> If our rebuttal has resolved your concerns or improved your understanding of the paper, **we would greatly appreciate it if you could reconsider your assessment and update the score accordingly.**
>
> Best regards,
>
> Authors

---

> ### Author Response · Authors · 2024-12-04
> **Kindly Request for Reviewer's Feedback**
>
> Dear Reviewer dLTN,
>
> We would like to express our heartfelt gratitude for the time and thoughtful effort you dedicated to reviewing our paper. We are also thankful for your recognition of its strengths.
>
> As the author/reviewer discussion phase has now concluded, we believe our rebuttal has fully addressed all of your comments and concerns. However, given the scoring standards of previous ICLR reviews, we notice that our current score of **5.5** is at the borderline level. We would greatly appreciate it if you could reconsider and potentially raise your score, allowing us the opportunity to present our work at the conference.
>
> Throughout the rebuttal period, we have made significant efforts to strengthen our paper, including the addition of over **200 new experiments and 7 new pages** of content. These updates provide a more comprehensive and robust presentation of our approach and its potential contributions.
>
> Additionally, we would like to highlight that **since the extended discussion period began, many authors have engaged in fruitful discussions with their reviewers, which has led to positive feedback and score increases**. We sincerely hope that our case is no exception and that the improvements made during the rebuttal period will result in a more comprehensive reassessment of our work.
>
> Thank you once again for your insightful review and constructive feedback.
>
> Best regards,
>
> Authors

---

### Official Review · Reviewer_Bbht · 2024-11-04

**Soundness:** 3
**Presentation:** 2
**Contribution:** 3
**Rating:** 6
**Confidence:** 4

**Summary:**

- The paper identifies explicitly and iteratively extracting linear and nonlinear patterns from time series as an opportunity to learn highly effective forecasting models.
- The extracted patterns are combined via simple linear layers to form the forecast.
- The method is then evaluated in a variety of experiments.

I did not check the entire appendix.

**Strengths:**

- I am not aware of these ideas being proposed before. However, I am not sufficiently familiar with the topic to rule it out definitively.
- The quality of the presentation ranges from poor (method section) to good (experiments).
- Time series forecasting is relevant in many domains and is far from being solved. The proposed method investigates a method that is interesting due to its interpretability and fair performance.
- The experiments go beyond mere performance metrics, performing an ablation study, other variations, depth sensitivity analysis, and lookback length investigations.

**Weaknesses:**

1. The method presentation in (mainly) Sec. 3.2 is poor. The notation is more complicated than necessary (e.g., it includes a batch dimension) while not being sufficiently precise to fully specify the computation (e.g., in l. 239 to 243). The accompanying Figure 2 further complicates things by its inconsistent colors, the red cutout not showing what it points at (on the left, the projections were independent), the MLPs only being linear layers, the "Cat" being a Hadamard product, arrows leading into the nothingness, and inconsistent formatting. See also the Questions section.
2. It is unclear to me why the Li block is called Linear. Going by Fig. 2 or Fig. 5, neither the pattern nor the prediction is linear in the time index, and the learned function is neither (e.g., there is a Dropout layer).
3. The learned representations are somewhat redundant. See, e.g., LP2 and LP3 in Fig. 5 (ETTh1, center) or NP1 and NP2 in Fig. 5 (ETTh1, right). These representations are not used for model interpretability.

#### Minor Comments
- The text size for the figure captions was very likely reduced from what is specified in the template.
- The reference's formatting is very sloppy (missing venues, capitalization of titles and names, broken math notation, and inconsistency).
- Section 3 could be simplified by omitting the batch dimension, as every operation only operates on a single time series. Given the deep learning context, a batch dimension can be conceptually added simply in one sentence without complicating notation.
- Language: "TSMxier" typo in l. 103. Reference in l. 118. Used citet instead of citep in l. 252. ...
- The convention of typesetting vectors and matrices in bold might ease readability (starting in Sec. 3.1).
- The images are not included as vector graphics, limiting accessibility, the ability to zoom in (cf. Fig. 3), and increasing file size.
#### References
- Oreshkin, Boris N., Dmitri Carpov, Nicolas Chapados, and Yoshua Bengio. “N-BEATS: Neural Basis Expansion Analysis for Interpretable Time Series Forecasting.” In International Conference on Learning Representations, 2019.
- Lin, Shengsheng, Weiwei Lin, Wentai Wu, Feiyu Zhao, Ruichao Mo, and Haotong Zhang. “SegRNN: Segment Recurrent Neural Network for Long-Term Time Series Forecasting.” arXiv, August 22, 2023. https://doi.org/10.48550/arXiv.2308.11200.
- Wang, Shiyu, Haixu Wu, Xiaoming Shi, Tengge Hu, Huakun Luo, Lintao Ma, James Y. Zhang, and Jun Zhou. “TimeMixer: Decomposable Multiscale Mixing for Time Series Forecasting.” In The Twelfth International Conference on Learning Representations, 2024.

**Questions:**

Note: The most important questions are listed first.

1. What is meant by "multi-level characteristics of real-world time series" (Sec. 1)? How do the time series in Fig. 1 relate to each other? How are they combined (additively?)?
2. L. 299f states, "These models represent the latest advancements in multivariate time series forecasting and encompass all mainstream prediction model types". However, recurrent models such as SegRNN (Lin et al. 2023) and the empirically strong TimeMixer model (Wang et al. 2024) are missing. Was this a deliberate choice?
3. Regarding the Li Block (Sec. 3.2): Why is padding performed? Is the linear prediction in l. 221 the same as the (not very linear) MLP in Fig. 2? Which of the model components are learned via gradient descent?
4. Regarding Fig. 2: What is the "backbone"?
5. Regarding the No Block (Sec. 3.2): Again, MLP blocks are suddenly linear projections (matrix multiplications). If the addition in l. 238 is element-wise, isn't the overall $N_i^{TF}$  entirely linear up to the point of activation. How can it be beneficial over a simple linear layer? Why is Tanh chosen, given that ReLU is a more common choice today? I do not understand the motivation behind the construction in l. 239-250.
6. The iterative pattern extraction reminds me of N-BEATS (Oreshkin et al., 2019). Are they that closely related? If so, this might be relevant for Sec. 2.
7. What are the diagonal stripes in, for instance, Fig. 19 (bottom, left)?

---

> ### Author Response · Authors · 2024-11-21
> **Response to Reviewer Bbht(Part 1)**
>
> We sincerely appreciate Reviewer Bbht's valuable suggestions regarding the **representation**, **figures**, and **citation formatting** in our manuscript. We also highly value your concerns about our proposed method and the weaknesses you pointed out.
>
> Here are our responses to your concerns and questions.
>
> We hope they will help address your confusion and provide clarity.
>
>
> # Minor Comments
> We have addressed all minor comments, including formatting, typos, notation clarity, vector graphics for figures (Now, all figures are in PDF to  ease readability), and improved references. We appreciate your thorough suggestions!!
>
>
> # Weakness
>
>
> > **W1. The method presentation in (mainly) Sec. 3.2...**
>
> Thank you for the detailed feedback!!!!
>
> We have revised Sec. 3.2 for clarity with simplified notation, addressed computational ambiguities. We also updated Figure 2 with consistent formatting and accurate representation of operations.
>
> Please check **Figure.2** and **Sec.3.2** in the updated PDF.
>
> > **W.2 It is unclear to me why the Li block is called Linear.**
>
> The linear pattern extraction in the Li block relies on an **Autoregressive (AR) model**. Therefore, the extraction of $\hat{L}_i$ is completely **Linear** with respect to the input $H_i$.
>
> **Dropout** randomly setting some values in $\hat{L}_i$ to zero during $training$. During *inference (after training)*, **Dropout** is typically turned off, and the model behaves deterministically. Thus, the application of Dropout does not change the linearity.
>
> The **Linear prediction** is obtained by:
> $P^{li}_{i}=L_iW+b$
>
> Thus, the overall process in the **Li block** — from feature extraction to the final prediction — is linear in nature.
>
> And since we employ an Autoregressive model to capture linear patterns in raw time series, we can see in **Figure.1** that the Autocorrelation Function (ACF) of the decomposed linear patterns (Linear 1 and Linear 2) shows clear pattern of AR(1) time series, which means current point is predominantly influenced by its previous point.
> > **W.3 The learned representations are somewhat redundant.**
>
>
> Revisiting our assumptions, the composition of linear and nonlinear patterns in each time series is undoubtedly diverse.
>
> We argue that such redundancy is both natural and reasonable, as it is likely that the time series contains only **a single linear pattern**.
>
> Once this pattern is fully captured by one Li Block, the patterns extracted by other Li Blocks become insignificant. The same logic applies to No Blocks.
>
> In Fig. 5, we observe a dominant linear pattern (LP1) on **ETTh1**, while the other two (LP2&3) appear less significant.
>
> In contrast, the nonlinear patterns NP1&2&3 have almost identical weights.
>
> In Table 4, we can also observe that the optimal LiNo blocks for each dataset are diverse, further highlighting the differences in the linear and nonlinear compositions of each dataset.
>
> So, such prediction visualization of LiNo provides us with the possibility to explore the unknown linear and nonlinear components of time series data.

---

> ### Author Response · Authors · 2024-11-21
> **Response to Reviewer Bbht(Part 2)**
>
> # Questions
>
> > **Q.1 What is meant by "multi-level characteristics of real-world time series" (Sec. 1)? How do the time series in Fig. 1 relate to each other? How are they combined (additively?)?**
>
> The *multi-level characteristics of real-world time series* suggests that real-world time series are the result of the additive combination of various linear and nonlinear components (as illustrated in **Fig.1** and **Sec.3.1**).
>
> **Fig.1** is the illustration of this assumption, where a raw time series (**red**) is decomposed into a series of subsequences (2 linear and 2 nonlinear). In other words, the raw time series (red) is the sum of the four time series (**additively**).
>
> > **Q.2 Why SegRNN and TimeMixer are not included as baselines?**
>
> We did not include SegRNN and TimeMixer mainly based on the following considerations:
> 1. SegRNN uses **MAE** as its loss function, which is widely regarded as an unfair comparison against baselines (including our LiNo, Informer, Autoformer, iTransformer, etc.) that use **MSE** as the loss function.
> 2. SegRNN uses an input length of **720**, and TimeMixer uses **512**, whereas in our paper, to follow more influential works like Informer, Autoformer, and iTransformer, the input length for all baselines is uniformly set to **96**.
> 3. Both SegRNN and TimeMixer have an uncorrected critical bug, which first identified in [FITS](https://arxiv.org/abs/2307.03756). Details of the bug can be found [here](https://github.com/VEWOXIC/FITS). This bug causes the models to skip a significant portion of the test set, resulting in unfair comparisons.
>
> > **Q.3 Why is padding performed? Is the linear prediction in l. 221 the same as the (not very linear) MLP in Fig. 2? Which of the model components are learned via gradient descent?**
>
> Padding before convolution ensures proper alignment for the proposed convolution operation (A concise and efficient design for the **Autoregressive model**). This makes sure the input $H_i\in \mathbb{R}^{C \times D}$ and output $\hat{L}_i\in \mathbb{R}^{C \times D}$ share the same feature dimension as in **Autoformer**.
>
> The MLP (which, thanks to your suggestion, should be more appropriately called FC) of the Li Blcok in Fig.2 refers to the linear prediction in l. 221.
>
> All parameters in **Li Block**, including the autoregressive coefficients, biases, and weights in FCs, are learned via gradient descent during training.
>
> > **Q.4 Regarding Fig. 2: What is the "backbone"?**
>
> The **"backbone"** to refer to the LiNo Block, which includes a Li Block and a No Block. However, upon careful consideration, we realized that the original diagram might cause confusion.
>
> **Therefore, we decided to abandon this concept.**
>
> Please refer to the updated model diagram in **Figure 2 (middle)** for the structure of the proposed LiNo Block.
>
> We sincerely apologize for any confusion caused by the initial version of the diagram!

---

> ### Author Response · Authors · 2024-11-21
> **Response to Reviewer Bbht(Part 3)**
>
> > **Q.5  If the addition in l. 238 is element-wise, isn't the overall $N_i^{TF}$ entirely linear up to the point of activation. How can it be beneficial over a simple linear layer? Why is Tanh chosen, given that ReLU is a more common choice today? I do not understand the motivation behind the**
>
> Separately using temporal and frequency linear projections allows the model to specialize in capturing distinct aspects of time series data. Combining these projections using **$N_i^{TF}$** enhances expressiveness, improving the model's ability to learn complex patterns and leading to better performance than a single projection.
>
> **Tanh** was chosen over **ReLU** to provide a smooth, symmetric activation function that better captures the complex dependencies in both time and frequency domains.
>
> These are proved in the following experiment result.
>
>  1. **Temporal** means only extracting temporal variations (*without introducing nonlinearity*)
>
>  2. **Frequency** strands for frequency information extraction (*without introducing nonlinearity*)
>
>  3. **TF** means these two patterns are captured simultaneously (*without introducing nonlinearity*)
>
>  4. The nonlinearity of the No Block is introduced by adding **ReLU** or **Tanh** activation function (*based on **TF***).
>
> The combination of **Temporal and Frequency** domain information (**TF**) consistently outperform single domain, and **Tanh consistently outperforms ReLU** across all datasets.
>
> | Models   | Length | Temporal MSE | Temporal MAE | Frequency MSE | Frequency MAE | TF MSE | TF MAE | ReLU MSE | ReLU MAE | Tanh MSE | Tanh MAE |
> |----------|--------|--------------|--------------|---------------|---------------|--------|--------|----------|----------|----------|----------|
> | **ETTh1**| 96     | 0.385        | 0.398        | 0.383         | 0.393         | 0.383  | **0.392**  | 0.384    | 0.398    | **0.375**    | 0.394    |
> |          | 720    | 0.487        | 0.475        | 0.482         | 0.469         | 0.478  | 0.467  | 0.488    | 0.477    | **0.464**    | **0.458**    |
> | **ETTm1**| 96     | 0.341        | 0.369        | 0.342         | 0.369         | 0.338  | 0.366  | 0.328    | 0.365    | **0.322**    | **0.359**    |
> |          | 720    | 0.481        | 0.447        | 0.479         | 0.445         | 0.474  | **0.439**  | 0.478    | 0.445    | **0.465**    | 0.442    |
> | **ECL**  | 96     | 0.186        | 0.268        | 0.187         | 0.267         | 0.183  | 0.263  | 0.154    | 0.248    | **0.150**    | **0.243**    |
> |          | 720    | 0.245        | 0.320        | 0.245         | 0.320         | 0.241  | 0.317  | 0.223    | 0.311    | **0.221**    | **0.308**    |
> | **Weather** | 96  | 0.165        | 0.209        | 0.165         | 0.208         | **0.162**  | **0.206**  | 0.162    | 0.208    | 0.163    | 0.209    |
> |           | 720   | 0.344        | 0.344        | 0.345         | 0.343         | **0.342**  | **0.338**  | 0.345    | 0.345    | 0.343    | 0.339    |
>
> > **Q.6 The iterative pattern extraction reminds me of N-BEATS (Oreshkin et al., 2019). Are they that closely related? If so, this might be relevant for Sec. 2.**
>
> Though **LiNo and N-BEATS** share some similarities, such as both employing the concept of **Recursive Residual Decomposition (RRD)**, they are fundamentally different:
> 1. N-BEATS does not explicitly capture linear and nonlinear separately.
> 2. As **a univariate** model, N-BEATS cannot handle more complex multivariate time series.
> 3. The role of RRD differs in each model: while N-BEATS uses RRD to refine the final prediction based on residual errors from previous predictions, LiNo employs RRD to capture more nuanced linear and nonlinear patterns.
>
> We added these comparisons in the updated PDF, **Section 2**. We included further analysis on N-BEATS and LiNo in **Appendix F**.
>
> > **Q.7 What are the diagonal stripes in, for instance, Fig. 19 (bottom, left)?**
>
> Fig. 19-31 are visualizations of the trained weights of Li Blocks and No Blocks on each dataset.
>
> We employed the visualization techniques provided in the [An Analysis of Linear Time Series Forecasting Models](https://openreview.net/forum?id=xl82CcbYaT) to explore the patterns learned by each Li Block and No Block.
>
> More details are included in Appendix D.2.

---

> > ### Author Response · Authors · 2024-11-23
> > **Request of Reviewer's attention and feedback**
> >
> > Dear Reviewer,
> >
> > We kindly remind you that it has been 2 days since we posted our rebuttal. Please let us know if our response has addressed your concerns.
> >
> > Following your suggestion, we have answered your concerns and improved the paper in the following aspects:
> > 1. A detailed and thorough discussion and comparison with **[N-BEATS](https://openreview.net/forum?id=r1ecqn4YwB)** in the **Sec.2** and **Appendix. F**.
> >
> > 2. A clearer and more precise model diagram, **Fig. 2**.
> >
> > 3. All images are in **PDF** format to increase readability and accessibility.
> >
> > 4. Modified all inappropriate **representation**, **typo** errors, and **reference** format.
> >
> > 5. Additional **ablation study** about the choice of model design in **Sec.4.3.2, Table.5**.
> >
> > In total, we have added **more than 50 new experiments**. All of these results have been included in the **revised paper**. Thanks again for your valuable review. We are looking forward to your reply and are happy to answer any future questions.

---

> ### Comment · Reviewer_Bbht · 2024-11-24
> **Convincing, except for the choice of benchmarks**
>
> Oddly, Figures 3 & 4 are still rasterized despite your comment. However, this is not relevant for assessing whether the work should be accepted and can also be fixed in a camera-ready version.
>
> - W.1: The presentation (e.g., Figure 2) has indeed improved. Although generally, highlighting changes by color changes is helpful. It would make it easier to check this properly.
> - W.2: Makes sense; thank you for clarifying.
> - W.3: I'll assume that you meant Figure 4. I agree. This makes sense.
> - Q.1: This could be clarified in the manuscript, too.
> - **Q.2**: **I strongly disagree with them being impossible to compare against for several reasons.** Firstly, SegRNN could very likely be trained by MSE, too. Secondly, training with different optimization criteria does not diminish comparability as long as the evaluation is done the same way (see, e.g., language model benchmarks). Thirdly, the claim, at least about TimeMixer, is plainly wrong. They explicitly discuss that different input lengths are not really comparable (see paragraph "Unified experiment settings" in p. 6). Subsequently, they "fix the input length as 96 for all experiments" for their main results in Table 2 (see caption). Fourthly, even if so, this is just an implementation detail and could be made comparable by you. Fifthly and lastly, the error you mentioned is just an implementation detail and not a flaw in the methods. You could simply provide a fair comparison yourself by changing one line and re-running the models.
> - Q.3: Thank you.
> - Q.4: Thank you for the cleanup.
> - Q.5: While I do not understand why exactly this is happening (I lack a theory), you provide very convincing results. This work does not require an even deeper analysis.
> - Q.6: Thank you for elaborating. This makes sense.
> - Q.7: Alright.
>
> **While I am willing to raise scores (rating from 3 to 5, presentation by one, confidence by one), I am still concerned by the incomplete evaluation as discussed in Q.2.**

---

> > ### Author Response · Authors · 2024-11-25
> > **The Second Response to Reviewer Bbht [Part 1]**
> >
> > Deep thanks for your insightful response and raising score!!
> >
> > Following your advice, We have included the clarification of Q1 into the manuscript, please check **L.49-52 and and the last sentence of the caption of Fig.1** in the latest version PDF.
> >
> > Again, we sincerely appreciate your constructive suggestions. After careful and thorough consideration, we have implemented comparative experiments with **SegRNN** and **TimeMixer** as per your instructions.
> >
> > For SegRNN, upon investigation, we noticed that their [**Official Implementation Repository**](https://github.com/lss-1138/SegRNN) had successfully resolved the bug mentioned. Additionally, they added their experimental results with input length of **$𝐿=96$**. Therefore, we directly referenced those results. **It is worth noting that their training loss function remains MAE**.
> >
> > For TimeMixer, we utilized the well-known [**Time Series Library**](https://github.com/thuml/Time-Series-Library), authored by the same team as TimeMixer. Importantly, this library addresses the bug we previously mentioned. We re-ran the models using this updated implementation.
> >
> > The results are as follows: LiNo still consistently outperforms overall, achieving leading results in **15/28 cases for MSE** and **19/28 cases for MAE**.
> >
> > Thank you once again for your valuable suggestions.
> >
> > |Dataset|Length|LiNo|LiNo|TimeMixer|TimeMixer|SegRNN(MAE)|SegRNN(MAE)|
> > |-|-|-|-|-|-|-|-|
> > |Metric||MSE|MAE|MSE|MAE|MSE|MAE|
> > |ETTm1  |96    |**0.322**|**0.361**|0.328|0.367|0.330|0.369|
> > |       |192   |**0.365**|**0.383**|0.369|0.389|0.369|0.392|
> > |       |336   |0.401|**0.408**|0.404|0.411|**0.399**|0.412|
> > |       |720   |0.469|0.447|0.473|0.451|**0.454**|**0.443**|
> > |ETTm2  |96    |**0.171**|**0.254**|0.176|0.259|0.173|0.255|
> > |       |192   |**0.237**|**0.298**|0.242|0.303|0.237|0.298|
> > |       |336   |**0.296**|**0.336**|0.303|0.339|0.296|0.336|
> > |       |720   |0.395|**0.393**|0.396|0.399|**0.389**|0.407|
> > |ETTh1  |96    |0.378|0.395|0.384|0.400|**0.368**|**0.395**|
> > |       |192   |0.423|0.423|0.437|0.429|**0.408**|**0.419**|
> > |       |336   |0.455|**0.438**|0.472|0.446|**0.444**|0.440|
> > |       |720   |0.459|**0.456**|0.508|0.489|**0.446**|0.457|
> > |ETTh2  |96    |0.292|0.340|0.297|0.348|**0.278**|**0.335**|
> > |       |192   |0.375|0.391|0.369|0.392|**0.359**|**0.389**|
> > |       |336   |**0.418**|**0.426**|0.427|0.435|0.421|0.436|
> > |       |720   |**0.422**|**0.441**|0.442|0.461|0.432|0.455|
> > |ECL    |96    |**0.138**|**0.233**|0.153|0.244|0.151|0.245|
> > |       |192   |**0.155**|**0.250**|0.168|0.259|0.164|0.258|
> > |       |336   |**0.171**|**0.267**|0.185|0.275|0.180|0.277|
> > |       |720   |**0.191**|**0.290**|0.227|0.312|0.218|0.313|
> > |Traffic|96    |0.429|0.276|0.473|0.287|**0.419**|**0.269**|
> > |       |192   |0.450|0.289|0.486|0.294|**0.434**|**0.276**|
> > |       |336   |0.468|0.297|0.488|0.298|**0.450**|**0.284**|
> > |       |720   |0.514|0.320|0.536|0.314|**0.483**|**0.302**|
> > |Weather|96    |**0.154**|**0.199**|0.162|0.208|0.165|0.227|
> > |       |192   |**0.205**|**0.248**|0.208|0.252|0.211|0.273|
> > |       |336   |**0.262**|**0.290**|0.263|0.293|0.270|0.318|
> > |       |720   |**0.343**|**0.342**|0.345|0.345|0.357|0.376|
> > |**1st Count**| |**15**|**19**|0|0|13|9|

---

> > > ### Author Response · Authors · 2024-11-25
> > > **The Second Response to Reviewer Bbht [Part 2]**
> > >
> > > During our experiments, we also identified several important insights, which we would like to share with you.
> > >
> > > **These findings, along with all the results mentioned, will be incorporated into our final camera-ready version**.
> > >
> > > ## 1. First, regarding the bug mentioned earlier:
> > > ```
> > > if flag == 'test':
> > >     shuffle_flag = False
> > >     drop_last = True
> > >     batch_size = args.batch_size
> > > ```
> > > Due to the setting ```drop_last = True```, a significant portion of the test samples may be excluded from evaluation when using a large batch size (e.g., **256** in [SegRNN](https://github.com/lss-1138/SegRNN/blob/main/scripts/SegRNN/etth2.sh)). For instance, given that the test set of ETTh1&2 contains **only 2785 samples**, this results in many samples being left untested.
> > >
> > > Moreover, because ```shuffle_flag = False```, the untested samples are always those at the end of the test set. These samples are typically the most challenging to predict, as they are temporally farthest from the training set and thus most prone to distributional shifts, characterized by noticeable differences in **mean and variance** compared to the training set.
> > >
> > > **Thus, We believe this issue is critical and should be addressed seriously by the community as a whole**.
> > >
> > > ## 2. Second, we argue that differing optimization criteria indeed diminish the comparability between baselines.
> > >
> > > The following experimental results present the performance of LiNo and SegRNN trained with MAE or MSE, respectively. It is evident that the results obtained using MAE are almost always superior to those obtained with MSE. And the difference is not subtle; it is a significant and unmistakable discrepancy. **Therefore, comparing baselines trained with MAE to those trained with MSE would certainly lead to an unfair comparison!**
> > >
> > > |Models|length|LiNo(MSE)|LiNo(MSE)|SegRNN(MSE)|SegRNN(MSE)|LiNo(MAE)|LiNo(MAE)|SegRNN(MAE)|SegRNN(MAE)|
> > > |-|-|-|-|-|-|-|-|-|-|
> > > |Metric||MSE|MAE|MSE|MAE|MSE|MAE|MSE|MAE|
> > > |ETTm1|96|0.322|0.361|0.342|0.379|**0.310**|**0.339**|0.330|0.369|
> > > |     |192|0.365|0.383|0.383|0.402|**0.363**|**0.368**|0.369|0.392|
> > > |     |336|0.401|0.408|0.407|0.420|**0.396**|**0.388**|0.399|0.412|
> > > |     |720|0.469|0.447|0.471|0.455|**0.451**|**0.437**|0.454|0.443|
> > > |ETTm2|96|0.171|0.254|0.176|0.259|**0.170**|**0.248**|0.173|0.255|
> > > |     |192|0.237|0.298|0.241|0.305|**0.233**|**0.291**|0.237|0.298|
> > > |     |336|0.296|0.336|0.301|0.346|**0.291**|**0.329**|0.296|0.336|
> > > |     |720|0.395|0.393|0.425|0.436|**0.386**|**0.392**|0.389|0.407|
> > > |ETTh1|96|0.378|0.395|0.385|0.411|0.369|**0.388**|**0.368**|0.395|
> > > |     |192|0.423|0.423|0.434|0.441|0.419|**0.417**|**0.408**|0.419|
> > > |     |336|0.455|0.438|0.462|0.463|0.449|**0.436**|**0.444**|0.440|
> > > |     |720|0.459|0.456|0.497|0.488|0.453|**0.451**|**0.446**|0.457|
> > > |ETTh2|96|0.292|0.340|0.284|0.340|0.281|**0.333**|**0.278**|0.335|
> > > |     |192|0.375|0.391|0.375|0.396|0.366|**0.382**|**0.359**|0.389|
> > > |     |336|0.418|0.426|0.425|0.437|**0.408**|**0.421**|0.421|0.436|
> > > |     |720|0.422|0.441|0.431|0.454|**0.409**|**0.429**|0.432|0.455|
> > > |Weather|96|0.154|0.199|0.165|0.230|**0.150**|**0.187**|0.165|0.227|
> > > |       |192|0.205|0.248|0.213|0.277|**0.199**|**0.236**|0.211|0.273|
> > > |       |336|0.262|0.290|0.274|0.323|**0.256**|**0.282**|0.270|0.318|
> > > |       |720|0.343|0.342|0.354|0.372|**0.338**|**0.334**|0.357|0.376|
> > > |Avg||0.342|0.363|0.352|0.382|**0.335**|**0.354**|0.340|0.372|
> > >
> > > When we combine the results of the two loss functions (MSE and MAE) and take the average, our LiNo still significantly outperforms SegRNN, as shown below:
> > >
> > > |Models|length|LiNo(AVG)|LiNo(AVG)|SegRNN(AVG)|SegRNN(AVG)|
> > > |------|------|---------|---------|-----------|-----------|
> > > |Metric|      |MSE|MAE|MSE|MAE|
> > > |ETTm1|96|**0.316**|**0.350**|0.336|0.374|
> > > |     |192|**0.364**|**0.376**|0.376|0.397|
> > > |     |336|**0.399**|**0.398**|0.403|0.416|
> > > |     |720|**0.460**|**0.442**|0.463|0.449|
> > > |ETTm2|96|**0.171**|**0.251**|0.175|0.257|
> > > |     |192|**0.235**|**0.295**|0.239|0.302|
> > > |     |336|**0.294**|**0.333**|0.299|0.341|
> > > |     |720|**0.391**|**0.393**|0.407|0.422|
> > > |ETTh1|96|**0.374**|**0.392**|0.377|0.403|
> > > |     |192|**0.421**|**0.420**|0.421|0.430|
> > > |     |336|**0.452**|**0.437**|0.453|0.452|
> > > |     |720|**0.456**|**0.454**|0.472|0.473|
> > > |ETTh2|96|0.287|**0.337**|**0.281**|0.338|
> > > |     |192|0.371|**0.387**|**0.367**|0.393|
> > > |     |336|**0.413**|**0.424**|0.423|0.437|
> > > |     |720|**0.416**|**0.435**|0.432|0.455|
> > > |Weather|96|**0.152**|**0.193**|0.165|0.229|
> > > |       |192|**0.202**|**0.242**|0.212|0.275|
> > > |       |336|**0.259**|**0.286**|0.272|0.321|
> > > |       |720|**0.341**|**0.338**|0.356|0.374|
> > > |Avg|   |**0.338**|**0.359**|0.346|0.377|

---

> ### Comment · Reviewer_Bbht · 2024-11-25
> **Convincing response**
>
> Thank you for expanding the paper in this regard. It is a very significant improvement, and **resolves my remaining main concern** beside the initially confusing presentation. **I thus raise my score from 5 to 6** and the soundness by one. I will not change it further.
>
> Closing, some remarks you may consider for the camera-ready version:
> - Regarding the implementation of the test size, you are right. This, indeed, has the potential to skew results significantly and should receive a bit more attention than it already did.
> - The results you reported do not quite match the ones in Table 13 of Wang et al. (2024), making TimeMixer appear slightly worse than in the original work. I expect you for the final paper to double-check that nothing went wrong.
> - While different training objectives yield different results, I still do not see the problem of no comparability. From my point of view, this is merely another hyperparameter to tune.

---

> > ### Author Response · Authors · 2024-11-25
> > **Deep thanks for your acknowledgement of our efforts and raising the score**
> >
> > Dear Reviewer Bbht,
> >
> > Thank you for your thoughtful feedback and for raising the score.
> >
> > All points raised are discussed during the rebuttal phase, including experimental results and clarifications, will be thoroughly reviewed and incorporated to enhance the paper’s clarity and accuracy.
> >
> > Deep thank again!
> >
> > BTW

---

> ### Author Response · Authors · 2024-12-04
> **Kindly Request for Reviewer's Feedback**
>
> Dear Reviewer Bbht,
>
> We sincerely appreciate the time and effort you dedicated to reviewing our paper during this busy period, as well as your recognition of its strengths and our efforts in the rebuttal phase (**with more than 200 new results and 7new pages**) that eventually clarified all of your concerns.
>
> With the author/reviewer discussion phase now concluded, and no further concerns raised, we believe our rebuttal has adequately addressed all of your comments. However, based on the scoring standards of previous ICLRs, we are concerned that our current score of **5.5** places us at risk of rejection. We would be sincerely grateful if you could reconsider your score and provide us with the opportunity to present our work at the conference.
>
> Thank you once again for your thoughtful review and valuable feedback.
>
> Best regards,
>
> Authors

---

### Official Review · Reviewer_236E · 2024-11-07

**Soundness:** 3
**Presentation:** 3
**Contribution:** 3
**Rating:** 6
**Confidence:** 3

**Summary:**

This paper presents a new neural network architecture for multi-variate time series forecasting. The main idea is to iteratively extract linear and non-linear patterns from the time series which are easier to learn and forecast. The linear patterns are extracted using 1D convolutions (part of the Li Block) where as the non linear features (part of the No Block) are extracted using linear projections on the frequency (FFT) and time domain (temporal variation features). The inter time-series dependencies are modeled using a weighted mean (akin to a pooling function) where the weights are computed using a softmax function.

The LiNo blocks are iteratively applied to the residuals to build predictors for each residual. All the predictors are finally added to produce the final prediction. The authors have presented extensive experimentation showing the effectiveness of the approach yielding state of the art results on several benchmark datasets.

**Strengths:**

Strengths:
- The author presented a novel time series forecasting method based on the idea of iteratively extracting linear and non-linear features from the residuals. While the idea has been introduced earlier in the literature (RRD), this paper extends the idea using neural networks.
- The experiments performed in this paper are extensive and the baselines used are state of the art. Comparison with several state of the art baselines on benchmark performance shows improved prediction accuracy.
- Additionally, the analysis presented in the paper including the ablation studies are extensive furthering strengthening the proposed method.

**Weaknesses:**

Weaknesses
- Several parts of the paper are unclear. In particular the design of the model (see questions below)
- The paper builds on the ideas of residual modeling and non-linear feature extraction using FFT, both which have been explored earlier in the literature [1, 2, 3] ([1] has not been cited), weakening the novelty of the approach slightly.

[1] Oreshkin, Boris N., et al. "N-BEATS: Neural basis expansion analysis for interpretable time series forecasting." arXiv preprint arXiv:1905.10437 (2019).

[2] Challu, Cristian, et al. "Nhits: Neural hierarchical interpolation for time series forecasting." Proceedings of the AAAI conference on artificial intelligence. Vol. 37. No. 6. 2023.

[3] Xu, Zhijian, Ailing Zeng, and Qiang Xu. "FITS: Modeling time series with $10 k $ parameters." arXiv preprint arXiv:2307.03756 (2023).

**Questions:**

- What is the advantage of using frequency domain operations? How do they benefit the predictions? Is it only useful if there are cyclical seasonal patterns? Will it be useful in cases where there is no seasonality or the frequency of the seasonal pattern can shift?
- Line 239 to 243: Can you elaborate on how the softmax is applied and the weighted mean is computed? It would be helpful to include a figure showing this operation in detail.
- How is the FFT and IFFT calculated? In particular, is this operation differentiable is is back-propagated through?
- Where does H_i sit in figure 2?
- Figure 2 No Block: Why is the MLP applied to the final prediction (right before adding the prediction from the Li Block) but not when computing the residual for the next block?
- Figure 2: What does the backbone consist of?
- What is the purpose of dropout in Equation 2? Does it help with spreading out the coefficients across the whole input window rather than have it concentrated on a few time steps?

Some suggested clarifications in the text:
- Line 078: "Another problem is the design of current nonlinear models, which mainly focus on
one or two types of nonlinear patterns."
Which types of non-linear patterns are being talked about here.
- Line 090: What is "traditional RRD"?
- Line 143: Typo "closely close"
- Eq 3: did you mean T<D?

---

> ### Author Response · Authors · 2024-11-21
> **Response to Reviewer 236E**
>
> We sincerely thank Reviewer 236Efor providing thorough and insightful comments. Here are our responses to your concerns and questions.
>
> *Tips: All the valuable suggested clarifications you provided have been carefully considered and incorporated into our revisions. Thank you once again.*
>
> > **Q1. Advantage of frequency domain operations**
>
> Frequency domain operations capture periodic and high-frequency components, aiding predictions even with shifting or weak seasonality, as shown in [FITS](https://openreview.net/forum?id=bWcnvZ3qMb). They provide complementary insights alongside time-domain features, as shown in the experiment results below.
>
> We denote the use of only the temporal projection as **Temporal**, the use of only the frequency projection as **Frequency**, and the combination of both as **TF**.
>
> | Datasets| Input| Temporal | Temporal | Frequency | Frequency | TF | TF |
> |-|-|-|-|-|-|-|-|
> || |MSE|MAE|MSE|MAE|MSE|MAE|
> | ETTh1    | 96     | 0.385        | 0.398        | 0.383         | 0.393         | **0.383**   | **0.392**  |
> |          | 720    | 0.487        | 0.475        | 0.482         | 0.469         | **0.478**   | **0.467**  |
> | ETTm1    | 96     | 0.341        | 0.369        | 0.342         | 0.369         | **0.338**   | **0.366**  |
> |          | 720    | 0.481        | 0.447        | 0.479         | 0.445         | **0.474**   | **0.439**  |
> | ECL      | 96     | 0.186        | 0.268        | 0.187         | 0.267         | **0.183**   | **0.263**  |
> |          | 720    | 0.245        | 0.320        | 0.245         | 0.320         | **0.241**   | **0.317**  |
> | Weather  | 96     | 0.165        | 0.209        | 0.165         | 0.208         | **0.162**   | **0.206**  |
> |          | 720    | 0.344        | 0.344        | 0.345         | 0.343         | **0.342**   | **0.338**  |
>
> > **Q2. Line 239 to 243: Can you elaborate on how the softmax is applied and the weighted mean is computed? It would be helpful to include a figure showing this operation in detail.**
>
> > **Q.4 Where does H_i sit in figure 2?**
>
> We have redrawn the model diagram based on your invaluable suggestion, and additionally, we included an extra figure to provide a detailed explanation of the proposed channel mixing technique in detail.
>
> $H_i$ is the input feature of each LiNo Block with $H_0=X_{embed}$ and $H_i=R^N_{i-1}$, so **$H_i$** sit at the beginning of each LiNo block.
>
> Please check **Figure.2 (upper left)**  for **channel mixing**, and **Figure.2 (left)**  for **where **$H_i$** sit**.
>
> > **Q.3 How is the FFT and IFFT calculated? In particular, is this operation differentiable is is back-propagated through?**
>
> Following [FITS](https://openreview.net/forum?id=bWcnvZ3qMb), the FFT/IFFT operations are implemented using PyTorch’s differentiable operations, ensuring seamless gradient flow for back-propagation and enabling the model to learn frequency-based features effectively.
>
> Give an input $R^L_i\in \mathbb{R}^{C \times D}$, the FFT, FC linear projection, IFFT are sequentially applyed to the $D$ dimension.
>
> > **Q.5 Why the MLP is'nt applied when computing the residual for the next block**
>
> Applying the MLP directly to the obtained residual risks altering critical features needed for subsequent blocks. Keeping residuals untouched ensures each block extracts complementary features effectively without redundant interference.
>
> Besides, This helps ensure that all residuals can be reconstructed back to the original input features, as emphasized in the preliminary analysis of Sec.3.1, preserving information integrity.
>
> > **Q.6 Figure 2: What does the backbone consist of?**
>
> Initially, we considered using **"backbone"** to refer to the LiNo Block, which includes a Li Block and a No Block. However, upon careful consideration, we realized that the original diagram might cause confusion.
>
> Therefore, we decided to abandon this concept. Please refer to the updated model diagram in **Figure 2 (middle)** for the details of the proposed LiNo Block.
>
> We sincerely apologize for any confusion caused by the initial version of the diagram!
>
> > **Q.7 What is the purpose of dropout in Equation 2?**
>
> Dropout in Equation 2 is used for regularization, helping to prevent overfitting.
>
> It doesn't directly affect coefficient distribution, as it will be turned off during inference. But it ensures the model generalizes better across the input window.

---

> > ### Author Response · Authors · 2024-11-23
> > **Request of Reviewer's attention and feedback**
> >
> > Dear Reviewer,
> >
> > We kindly remind you that it has been 2 days since we posted our rebuttal. Please let us know if our response has addressed your concerns.
> >
> > Following your suggestion, we have answered your concerns and improved the paper in the following aspects:
> > 1. A detailed and thorough discussion and comparison with **[N-BEATS](https://openreview.net/forum?id=r1ecqn4YwB)** in the **Sec.2** and **Appendix. F**.
> >
> > 2. A clearer and more precise model diagram, **Fig. 2**.
> >
> > 3. All images are in **PDF** format to increase readability and accessibility.
> >
> > 4. Modified all inappropriate **representation**, **typo** errors, and **reference** format.
> >
> > 5. Additional **ablation study** about the choice of model design in **Sec.4.3.2, Table.5**.
> >
> > In total, we have added **more than 50 new experiments**. All of these results have been included in the **revised paper**. Thanks again for your valuable review. We are looking forward to your reply and are happy to answer any future questions.

---

> ### Author Response · Authors · 2024-11-25
> **The discussion period ending soon**
>
> Dear Reviewer 236E,
>
> Thanks again for your valuable and constructive review, which helps us revise our work to a clearer stage.
>
> **We kindly remind you that the reviewer-author discussion phase will end soon. After that, we may not have a chance to respond to your comments.**
>
> Besides, during the rebuttal period, following your suggestion, we have answered your concerns and **improved the paper**, which may be helpful to you in further justifying our contribution.
>
> ***All points raised are discussed during the rebuttal phase, including experimental results and clarifications, will be incorporated into the camera-ready version to enhance the paper’s clarity and accuracy.***
>
> Thanks again for your dedication in reviewing our paper. It helps us a lot.
>
> We are looking forward to your feedback.

---

> > ### Comment · Reviewer_236E · 2024-11-25
> > **Maintaining my score**
> >
> > Thanks for the updates. I will maintain my score due to the strong similarities with N-BEATS as also pointed by other reviewers. The new figure unfortunately is still not as informative (this does not affect my score).

---

> > > ### Author Response · Authors · 2024-11-25
> > > **A further detailed comparision of LiNo and N-BEATS.**
> > >
> > > Dear Reviewer 236E,
> > >
> > > To better assist you in understanding the similarities and differences between our work and **N-BEATS**, and to help clarify our contributions, we have provided the following analysis.
> > >
> > > Here are some essential definitions:
> > >
> > > 1. The linear feature extractor is denoted as $Li\\_Block$, the extracted pattern $L_i$, and linear prediction $P^{li}_i$.
> > >
> > > 2. The nonlinear feature extractor is denoted as $No\\_Block$, the extracted pattern $N_i$, and nonlinear prediction $P^{no}_i$.
> > >
> > > 3. The projection to get the prediction is denoted as $FC$
> > >
> > > 4. We assume the input to the current layer in deep neural network as $H_i$ and the next layer $H_{i+1}$, where $H_0= X\\_embed=XW+b$, $X$ is the input raw time series
> > >
> > > 5. Prediction of current layer $P_{i}$
> > >
> > > 6. Final prediction $\hat{Y}$
> > >
> > > Then, **N-BEATS** can be summarized as:
> > >
> > > - $N_i = No\\_Block(H_i)$
> > > - $P_{i}=FC(N_i)$
> > > - $H_{i+1} = H_i-N_i$
> > > - $\hat{Y}=\sum_{i=1}^{N} P^{i}$
> > >
> > > While our **LiNo** can be formulated as:
> > >
> > > - $L_i = Li\\_Block(H_i)$
> > > - $P^{li}_{i}=FC(L_i)$
> > > - $R^{L}_{i} = H_i - L_i$
> > > - $N_i = No\\_Block(R^{L}_i)$
> > > - $P^{no}_{i}=FC(N_i)$
> > > - $R^{N}_{i} = R^{L}_i - N_i$
> > > - $H_{i+1} =R^{N}_{i}$
> > > - $P^{i}=P^{li}_i+P^{no}_i$
> > > - $\hat{Y}=\sum_{i=1}^{N} P^{i}$
> > >
> > > Below are some difference:
> > > 1. **LiNo** introduces a two-stage extraction process: linear (Li_Block) and nonlinear (No_Block).
> > > 2. **LiNo** computes predictions from both linear and nonlinear components, while **N-BEATS** only uses the nonlinear output.
> > > 3. **LiNo** refines intermediate representations after each extraction stage, whereas **N-BEATS** directly modifies hidden states.
> > > 4. **LiNo** combines both linear and nonlinear predictions, while **N-BEATS** only sums nonlinear predictions.
> > >
> > > We designed the following experiments to validate ***the superiority of LiNo over N-BEATS-style design***.
> > >
> > > The learnable autoregressive model (AR) mentioned in the paper is employed as the linear feature extractor ($Li\\_Block$).
> > >
> > > The Temporal + Frequency projection, combined with the Tanh activation function, is adopted as the nonlinear feature extractor ($No\\_Block$).
> > >
> > > To ensure a fair comparison with the **N-BEATS-style** design (since completely removing the $Li\\_Block$ may result in potential loss of parameters), the $No\\_Block$ used in N-BEATS is actually a combination of $Li\\_Block$ and $No\\_Block$, which is still nonlinear:
> > >
> > > - $L_i = Li\\_Block(H_i)$
> > > - $N_i = No\\_Block(L_i)$
> > > - $P_{i}=FC(N_i)$
> > > - $H_{i+1} = H_i-N_i$
> > > - $\hat{Y}=\sum_{i=1}^{N} P^{i}$
> > >
> > > Below are the experimental results of two design:
> > >
> > > | Models | length | N-BEATS MSE | N-BEATS MAE | LiNo MSE | LiNo MAE |
> > > |--------|--------|---------|---------|----------|----------|
> > > | ETTh2  | 96     | 0.126   | 0.271   | **0.126** | **0.271** |
> > > |        | 192    | 0.179   | 0.328   | **0.174** | **0.324** |
> > > |        | 336    | 0.213   | 0.368   | **0.209** | **0.364** |
> > > |        | 720    | 0.226   | 0.381   | **0.223** | **0.379** |
> > > | ETTm2  | 96     | 0.067   | 0.187   | **0.066** | **0.186** |
> > > |        | 192    | 0.102   | 0.238   | **0.099** | **0.234** |
> > > |        | 336    | 0.132   | 0.276   | **0.130** | **0.272** |
> > > |        | 720    | **0.185**   | **0.333**   | 0.187    | 0.335    |
> > > | Traffic| 96     | 0.168   | 0.253   | **0.159** | **0.248** |
> > > |        | 192    | 0.159   | 0.244   | **0.155** | **0.241** |
> > > |        | 336    | 0.156   | 0.242   | **0.153** | **0.237** |
> > > |        | 720    | 0.177   | 0.262   | **0.174** | **0.260** |
> > > | Avg    |        | 0.158   | 0.282   | **0.155** | **0.279** |
> > >
> > > ***We hope that these experimental results and analyses will help you gain a broad view of the contributions of this work.***
> > >
> > > BTW

---

> ### Author Response · Authors · 2024-12-02
> **Kindly Request for Reviewer's Feedback**
>
> Dear Reviewer 236E,
>
> Thank you for your time and effort in providing a thoughtful review of our work. As the discussion period is drawing to a close, we hope that you have had the opportunity to review our rebuttal.
>
> If our response has addressed your concerns or clarified any aspects of the paper, we would be grateful if you could reconsider your assessment and adjust the score accordingly. Your feedback has been invaluable, and we appreciate the chance to further refine our work based on your insights.
>
> Once again, thank you for your constructive review and for considering our responses. If you have any further questions or require additional clarifications, please do not hesitate to reach out.
>
> Best regards,
> Authors

---

> ### Author Response · Authors · 2024-12-04
> **Kindly Request for Reviewer's Feedback**
>
> Dear Reviewer 236E,
>
> We sincerely appreciate the effort you have put into reviewing our paper during this busy period, as well as the recognition of the strengths of our work. We are truly grateful for your assessment.
>
> We fully respect your current evaluation of our work. However, based on the scoring standards of past ICLRs, we find that our current score of **5.5** is at the borderline level. Therefore, we kindly ask if you could consider raising the score once again, so that we would have the opportunity to present our work at the conference. We would be deeply grateful.
>
> During the rebuttal period, we added **over 200 new experiments and 7 pages** of content, strengthening the paper and providing a more robust presentation of our approach.
>
> We also note that **many authors have received positive feedback and score increases during the extended discussion period.** We hope that the improvements made to our work will also lead to a more comprehensive reassessment.
>
> Thank you again for taking the time to review and comment on our paper.
>
> Best regards,
>
> Authors

---

### Author Response · Authors · 2024-11-25
**An updated version of our paper**

Dear reviewers:

As the rebuttal phase will close within **less than 37 hours**, we have promptly updated the latest version of the paper based on your valuable feedback during the rebuttal period, as well as the results of our fruitful discussions.

## ***All significant modifications and additions have been marked in purple.***

**The reviewers raised insightful and constructive concerns, and we have made every effort to address all of them by providing sufficient evidence and the requested results.** We hope these experimental results and analyses included in the latest version PDF will help clarify these concerns.

Here is the summary of the major revisions compare to the original paper:
1. **A more detailed and thorough** discussion and comparison with **[N-BEATS](https://openreview.net/forum?id=r1ecqn4YwB)** in the **Sec.2** and **Appendix. F** (***Both Experiment Results and Theory Analysis are included this time***).
2. A clearer and more precise model diagram, **Fig. 2**.
3. All images are in **PDF** format to increase readability and accessibility.
4. Modified all inappropriate **representation**, **typo** errors, and **reference** format.
5. Additional ablation study about **the choice of model design** in **Sec.4.3.2, Table.5** (individual component choice), and **the overall model design** in **Appendix. F.2, Table.12**.
6. Comparison with extensive baselines ([**SegRNN**](https://arxiv.org/abs/2308.11200) and [**TimeMixer**](https://openreview.net/forum?id=7oLshfEIC2)) in **Table.10**.
7. Comparison of the channel mixing technique used in proposed No Block and **SOFTS** in **Sec. 3.2, Lines 254-255, as well as Appendix G, Table 14.**.
8. Study about the influence of different loss function (MSE and MAE) to LiNo in **Table.11**.

In total, we have added **more than 200 new results, 7 pages**. All of these results have been included in the revised paper. Thanks again for your valuable review!

We are looking forward to your reply and are happy to answer any future questions.

---

### Note · Authors · 2025-01-22

I have read and agree with the venue's withdrawal policy on behalf of myself and my co-authors.